# The Dimension Strikes Back with Gradients: Generalization of Gradient Methods in Stochastic Convex Optimization

**Matan Schliserman**                                     SCHLISERMAN@MAIL.TAU.AC.IL
*Blavatnik School of Computer Science, Tel Aviv University*

**Uri Sherman**                                          URISHERMAN@MAIL.TAU.AC.IL
*Blavatnik School of Computer Science, Tel Aviv University*

**Tomer Koren**                                           TKOREN@TAUEX.TAU.AC.IL
*Blavatnik School of Computer Science, Tel Aviv University and Google Research*

**Editors:** Gautam Kamath and Po-Ling Loh

## Abstract

We study the generalization performance of gradient methods in the fundamental stochastic convex optimization setting, focusing on its dimension dependence. First, for full-batch gradient descent (GD) we give a construction of a learning problem in dimension $d = O(n^2)$, where the canonical version of GD (tuned for optimal performance on the empirical risk) trained with $n$ training examples converges, with constant probability, to an approximate empirical risk minimizer with $\Omega(1)$ population excess risk. Our bound translates to a lower bound of $\Omega(\sqrt{d})$ on the number of training examples required for standard GD to reach a non-trivial test error, answering an open question raised by Feldman (2016) and Amir, Koren and Livni (2021) and showing that a non-trivial dimension dependence is unavoidable. Furthermore, for standard one-pass stochastic gradient descent (SGD), we show that an application of the same construction technique provides a similar $\Omega(\sqrt{d})$ lower bound for the sample complexity of SGD to reach a non-trivial empirical error, despite achieving optimal test performance. This again provides for an exponential improvement in the dimension dependence compared to previous work (Koren et al., 2022), resolving an open question left therein.

## 1. Introduction

The study of generalization properties of stochastic optimization algorithms has been at the heart of contemporary machine learning research. While in the more classical frameworks studies largely focused on the learning problem (e.g., Alon et al., 1997; Blumer et al., 1989), in the past decade it has become clear that in modern scenarios the particular algorithm used to learn the model plays a vital role in its generalization performance. As a prominent example, heavily over-parameterized deep neural networks trained by first order methods output models that generalize well, despite the fact that an arbitrarily chosen Empirical Risk Minimizer (ERM) may perform poorly (Zhang et al., 2017; Neyshabur et al., 2014, 2017). The present paper aims at understanding the generalization behavior of gradient methods, specifically in connection with the problem dimension, in the fundamental Stochastic Convex Optimization (SCO) learning setup; a well studied, theoretical framework widely used to study stochastic optimization algorithms.

The seminal work of Shalev-Shwartz et al. (2010) was the first to show that uniform convergence, the canonical condition for generalization in statistical learning (e.g., Vapnik, 1971; Bartlett and Mendelson, 2002) may not hold in high-dimensional SCO: they demonstrated learning problems where there exist certain ERMs that overfit the training data (i.e., exhibit large population risk),

while models produced by e.g., Stochastic Gradient Descent (SGD) or regularized empirical risk minimization generalize well. The construction presented by Shalev-Shwartz et al. (2010), however, featured a learning problem with dimension exponential in the number of training examples, which only served to prove an $\Omega(\log d)$ lower bound on the sample complexity for reaching non-trivial population risk performance, where $d$ is the problem dimension. In a followup work, Feldman (2016) showed how to dramatically improve the dimension dependence and established an $\Omega(d)$ sample complexity lower bound, matching (in terms of $d$) the well-known upper bound obtained from standard covering number arguments (see e.g., Shalev-Shwartz and Ben-David, 2014).

Despite settling the dimension dependence of uniform convergence in SCO, it remained unclear from Shalev-Shwartz et al. (2010); Feldman (2016) whether the sample complexity lower bounds for uniform convergence actually transfer to natural learning algorithms in this framework, and in particular, to common gradient-based optimization methods. Indeed, it is well-known that in SCO there exist simple algorithms, such as SGD, that the models they produce actually generalize well with high probability (see e.g., Shalev-Shwartz and Ben-David, 2014), despite these lower bounds. More technically, the construction of Feldman (2016) relied heavily on the existence of a "peculiar" ERM which does not seem reachable by gradient steps from a data-independent initialization, and it was not at all clear (and in fact, stated as an open problem in Feldman, 2016) how to adapt the construction so as to pertain to ERMs that could be found by gradient methods.

In an attempt to address this issue, Amir et al. (2021b) recently studied the population performance of batch Gradient Descent (GD) in SCO, and demonstrated problem instances where it leads (with constant probability) to an approximate ERM that generalizes poorly, unless the number of training examples is dimension-dependent.[1] Subsequently, Amir et al. (2021a) generalized this result to the more general class of batch first-order algorithms. However, due to technical complications, the constructions in these papers were based in part on the earlier arguments of Shalev-Shwartz et al. (2010) rather than the developments by Feldman (2016), and therefore fell short of establishing their results in dimension polynomial in the number of training examples. As a consequence, their results are unable to rule out a sample complexity upper bound for GD that depends only (poly-)logarithmically on the problem dimension.

In this work, we resolve the open questions posed in both Feldman (2016) and Amir et al. (2021b): Our first main result demonstrates a convex learning problem where GD, unless trained with at least $\Omega(\sqrt{d})$ training examples, outputs a bad ERM with constant probability. This bridges the gap between the results of Feldman (2016) and actual, concrete learning algorithms (albeit with a slightly weaker rate of $\Omega(\sqrt{d})$, compared to the $\Omega(d)$ of the latter paper) and greatly improves on the previous $\Omega(\log d)$ lower bound of Amir et al. (2021b), establishing that the sample complexity of batch GD in SCO has a significant, polynomial dependence on the problem dimension.

Furthermore, in our second main result we show how an application of the same construction technique provides a similar improvement in the dimension dependence of the *empirical risk* lower bound presented in the recent work of Koren et al. (2022), thus also resolving the open question left in their work. This work demonstrated that in SCO, well-tuned SGD may *underfit* the training data despite achieving optimal population risk performance. At a deeper level, the overfitting of GD and underfitting of SGD both stem from a combination of two conditions: lack of algorithmic stability, and failure of uniform convergence; as it turns out, this combination allows for the output models to exhibit a large *generalization gap*, defined as the difference in absolute value between the empirical

---

1. Here we refer to GD as performing $T = n$ iterations with stepsize $\eta = \Theta(1/\sqrt{n})$, where $n$ denotes the size of the training set, but our results hold more generally; see below for a more detailed discussion of the various regimes.

and population risks. Our work presents a construction technique for such generalization gap lower bounds that achieves small polynomial dimension dependence, providing for an exponential improvement over previous works.

## 1.1. Our contributions

In some more detail, our main contributions are as follows:

(i) We present a construction of a learning problem in dimension $d = O(nT + n^2 + \eta^2 T^2)$ where running GD for $T$ iterations with step $\eta$ over a training set of $n$ i.i.d.-sampled examples leads, with constant probability, to a solution with population error $\Omega(\eta\sqrt{T} + 1/\eta T)$.[2] In particular, for the canonical configuration of $T = n$ and $\eta = \Theta(1/\sqrt{n})$, the lower bound becomes $\Omega(1)$ and demonstrates that GD suffers from catastrophic overfitting already in dimension $d = O(n^2)$. Put differently, this translates to an $\widetilde{\Omega}(\sqrt{d})$ lower bound the number of training examples required for GD to reach nontrivial test error. See Theorem 1 below for a formal statement and further implications of this result.

(ii) Furthermore, we give a construction of dimension $d = \widetilde{O}(n^2)$ where the empirical error of one-pass SGD trained over $T = n$ training examples is $\Omega(\eta\sqrt{n} + 1/\eta n)$. Assuming the standard setting of $\eta = \Theta(1/\sqrt{n})$, chosen for optimal test performance, the empirical error lower bound becomes $\Omega(1)$, showing that the "benign underfitting" phenomena of one-pass SGD is exhibited already in dimension polynomial in the number of training samples. Rephrasing this lower bound in terms of the number of training examples required to reach nontrivial empirical risk, we again obtain an $\widetilde{\Omega}(\sqrt{d})$ sample complexity lower bound. See Theorem 2 for the formal statement and further implications.

Both of the results above are tight (up to logarithmic factors) in view of existing matching upper bounds of Bassily et al. (2020). We remark that the constructions leading to the results feature *differentiable* Lipschitz and convex loss functions, whereas the lower bounds in previous works concerned with gradient methods (Amir et al., 2021b,a; Koren et al., 2022) crucially applied only to the class of non-differentiable loss functions. From the perspective of general non-smooth convex optimization, this implies that our lower bounds remain valid under *any* choice of a subgradient oracle of the loss function (as opposed to only claiming that there *exists* a subgradient oracle under which they apply, like prior results do).

## 1.2. Main ideas and techniques

Our work builds primarily on two basic ideas. The first is due to Feldman (2016), whereby an exponential number (in $n$) of approximately orthogonal directions, that represent the potential candidates for a "bad ERM," are embedded in a $\Theta(n)$-dimensional space. The second idea, underlying Bassily et al. (2020); Amir et al. (2021b,a); Koren et al. (2022), is to augment the loss function with a highly non-smooth component, that is capable of generating large (sub-)gradients around initialization directed at all candidate directions, that could steer GD towards a bad ERM that overfits the training set.

---

2. By population error (or test error) we mean the population excess risk, namely the gap in population risk between the returned solution and the optimal solution.

The major challenge is in making these two components play in tandem: since the candidate directions of Feldman (2016) are only *nearly* orthogonal, the progress of GD towards one specific direction gets hampered by its movement in other, irrelevant directions. And indeed, previous work in this context fell short of resolving this incompatibility and instead, opted for a simpler construction with a perfectly-orthogonal set of candidate directions, that was used in the earlier work of Shalev-Shwartz et al. (2010). Unfortunately though, this latter construction requires the ambient dimensionality to be exponential in the number of samples $n$, which is precisely what we aim to avoid.

Our solution for overcoming this obstacle, which we describe in length in Section 3, is based on several new ideas. Firstly, we employ multiple copies of the original construction of Feldman (2016) in orthogonal subspaces, in a way that it suffices for GD to make a *single* step within each copy so as to reach, across all copies, a bad ERM solution; this serves to circumvent the "collisions" between consecutive GD steps alluded to above. Secondly, we carefully design a convex loss term that, when augmented to the loss function, forces successive gradient steps to be taken in a round-robin fashion between the different copies, so that each subspace indeed sees a single update step through the GD execution. Lastly, we introduce a novel technique that memorizes the full training set by "encoding" it into the iterates in a convex and differentiable manner, so that the GD iterate itself (to which the subgradient oracle has access) contains the information required to "decode" the right movement direction towards a bad ERM. We further show how all of these added loss components can be made differentiable, so as to allow for a differentiable construction overall. A detailed overview of these construction techniques and a virtually complete description of our construction are provided in Section 3.

## 1.3. Additional related work

**Learnability and generalization in the SCO framework.** Our work belongs to the body of literature on stability and generalization in modern statistical learning theory, pioneered by Shalev-Shwartz et al. (2010) and the earlier foundational work of Bousquet and Elisseeff (2002). In this line of research, Hardt et al. (2016); Bassily et al. (2020) study algorithmic stability of SGD and GD in the smooth and non-smooth (convex) cases, respectively. In the general non-smooth case which we study here, Bassily et al. (2020) gave an iteration complexity upper bound of $O(\eta\sqrt{T} + 1/\eta T + \eta T/n)$ test error for $T$ iterations with step size $\eta$ over a training set of size $n$. The more recent work of Amir et al. (2021b) showed this to be tight up to log-factors in the dimension independent regime, and Amir et al. (2021a) further extends this result to any optimization algorithm making use of only batch gradients (i.e., gradients of the empirical risk).

Even more recently, Kale et al. (2021) consider a variation on the SCO model where individual losses may be non-convex (yet the expected loss is still convex). In this model, they prove that *regularized* ERMs may fail to learn (as opposed to plain ERMs, studied in Shalev-Shwartz et al. (2010); Feldman (2016)), and also present a sample complexity separation result between SGD and GD. As commented in their Section 3, the result relating to regularized ERMs can be extended to hold with $d = \Theta(n)$, by a direct application of the technique of Feldman (2016). Importantly, this is not the case for their sample complexity result for gradient methods, which inherits the exponential dimension requirement from Amir et al. (2021b) which they build upon.

**Sample complexity of ERMs.** With relation to the sample complexity of an (arbitrary) ERM in SCO, Feldman (2016) showed that reaching $\epsilon$-test error requires $\Omega(d/\epsilon + 1/\epsilon^2)$ training samples,

but did not establish optimality of this bound. In a recent work, Carmon et al. (2023) show this to be nearly tight and presents a $\widetilde{O}(d/\epsilon + 1/\epsilon^2)$ upper bound for any ERM, improving over the $O(d/\epsilon^2)$ upper bound that can be derived from standard covering number arguments. Another recent work related to ours is that of Magen and Shamir (2023), who provided another example for a setting in which learnability can be achieved without uniform convergence, showing that uniform convergence may not hold in the class of vector-valued linear (multi-class) predictors. However, the dimension of their problem instance was exponential in the number of training examples.

**Implicit regularization and benign overfitting.** Another relevant body of research focuses on understanding the effective generalization of over-parameterized models trained to achieve zero training error through gradient methods (see e.g., Bartlett et al., 2020, 2021; Belkin, 2021). This phenomenon appears to challenge conventional statistical wisdom, which emphasizes the importance of balancing data fit and model complexity, and motivated the study of implicit regularization (or bias) as a notion for explaining generalization in the over-parameterized regimes. Our findings in this paper could be viewed as an indication that, at least in SCO, generalization does not stem from some form of an implicit bias or regularization; see Amir et al. (2021b); Koren et al. (2022) for a more detailed discussion.

**Follow-up work.** Inspired by our techniques, Livni (2024) recently provided an $\widetilde{\Omega}(d)$ lower bound on the sample complexity of GD (in the same canonical setup we consider), which improves upon the $\widetilde{\Omega}(\sqrt{d})$ that follows from Theorem 1. We remark however that their construction does not seem to extend to a (generalization gap) lower bound construction for SGD.

## 2. Problem setup and main results

We consider the standard setting of Stochastic Convex Optimization (SCO). The problem is characterized by a population distribution $\mathcal{D}$ over an instance set $Z$, and loss function $f : W \times Z \to \mathbb{R}$ defined over convex domain $W \subseteq \mathbb{R}^d$ in $d$-dimensional Euclidean space. We assume that, for any fixed instance $z \in Z$, the function $f(w, z)$ is both convex and $L$-Lipschitz with respect to its first argument $w$. In this setting, the learner is interested in minimizing the *population loss* (or *risk*) which corresponds to the expected value of the loss function over $\mathcal{D}$, defined as

$$F(w) = \mathbb{E}_{z \sim \mathcal{D}}[f(w, z)], \qquad \text{(population risk/loss)}$$

namely, finding a model $w \in W$ that achieves an $\varepsilon$-optimal population loss, namely such that $F(w) \le F(w_*) + \varepsilon$, where $w_* \in \arg\min_{w \in W} F(w)$ is a population minimizer.

To find such a model $w$, the learner uses a set of $n$ training examples $S = \{z_1, \ldots, z_n\}$, drawn i.i.d. from the unknown distribution $\mathcal{D}$. Given the sample $S$, the corresponding *empirical loss* (or *risk*), denoted $\widehat{F}(w)$, is defined as the average loss over samples in $S$:

$$\widehat{F}(w) = \frac{1}{n} \sum_{i=1}^{n} f(w, z_i). \qquad \text{(empirical risk/loss)}$$

We let $\widehat{w}_* \in \arg\min_{w \in W} \widehat{F}(w)$ denote a minimizer of the empirical risk, refered to as an empirical risk minimizer (ERM). Moreover, for every $w \in W$, we define the *generalization gap* at $w$ as the absolute value of the difference between the population loss and the empirical loss, i.e., $|F(w) - \widehat{F}(w)|$.

**Optimization algorithms.** First-order algorithms make use of a (deterministic) subgradient oracle that takes as input a pair $(w, z)$ and returns a subgradient $g(w, z) \in \partial_w f(w, z)$ of the convex loss function $f(w, z)$ with respect to $w$. If $|\partial_w f(w, z)| = 1$, the loss $f(\cdot, z)$ is differentiable at $w$ and the subgradient oracle simply returns the gradient at $w$; otherwise, the subgradient oracle is allowed to emit any subgradient in the subdifferential set $\partial_w f(w, z)$.

First, we consider standard gradient descent (GD) with a fixed step size $\eta > 0$ applied to the empirical risk $\widehat{F}$. We allow for a potentially projected, $m$-suffix averaged version of the algorithm that takes the following form: Given an initialization $w_1 \in W$,

$$
\begin{aligned}
\text{update} \quad & w_{t+1} = \Pi_W\left[ w_t - \frac{\eta}{n} \sum_{i=1}^{n} g(w_t, z_i) \right], \qquad \forall\, 1 \leq t < T; \\
\text{return} \quad & w_{T,m} := \frac{1}{m} \sum_{i=1}^{m} w_{T-i+1}.
\end{aligned}
\tag{1}
$$

Here $\Pi_W : \mathbb{R}^d \to W$ denotes the Euclidean projection onto the set $W$; when $W$ is the entire space $\mathbb{R}^d$, this becomes simply unprojected GD. The algorithm returns either the final iterate, the average of the iterates, or more generally, any $m$-suffix average ($1 \leq m \leq T$) of iterates.

The second method that we analyze is Stochastic Gradient Descent (SGD), which is again potentially projected and/or suffix averaged. This method uses a fixed stepsize $\eta > 0$ and takes the following form: : Given an initialization $w_1 \in W$,

$$
\begin{aligned}
\text{update} \quad & w_{t+1} = \Pi_W\left[ w_t - \eta g(w_t, z_t) \right], \qquad \forall\, 1 \leq t < T; \\
\text{return} \quad & w_{T,m} := \frac{1}{m} \sum_{i=1}^{m} w_{T-i+1}.
\end{aligned}
\tag{2}
$$

**Main results.** Our main contributions in the context of SCO are tight lower bounds for the population loss of GD and for the empirical loss of SGD, where the problem dimension is polynomial in the number of samples $n$ and steps $T$. First, for the population risk performance of GD, we prove the following:

**Theorem 1** *Fix $n > 0, T > 3200^2$ and $0 \leq \eta \leq 1/(5\sqrt{T})$ and let $d = 178nT + 2n^2 + \max\{1, 25\eta^2 T^2\}$. There exists a distribution $\mathcal{D}$ over instance set $Z$ and a convex, differentiable and 1-Lipschitz loss function $f : \mathbb{R}^d \times Z \to \mathbb{R}$ such that for GD (either projected or unprojected; cf. Eq. (1) with $W = \mathbb{B}^d$ or $W = \mathbb{R}^d$ respectively) initialized at $w_1 = 0$ with step size $\eta$, for all $m = 1, \ldots, T$, the $m$-suffix averaged iterate has, with probability at least $\frac{1}{6}$ over the choice of the training sample,*

$$
F(w_{T,m}) - F(w_*) = \Omega\left( \min\left\{ \eta\sqrt{T} + 1/\eta T, 1 \right\} \right).
\tag{3}
$$

For SGD, we prove the following theorem concerning its empirical risk and generalization gap:

**Theorem 2** *Fix $n > 2048$ and $0 \leq \eta \leq 1/(5\sqrt{n})$ and let $d = 712n \log n + 2n^2 + \max\{1, 25\eta^2 n^2\}$. There exists a distribution $\mathcal{D}$ over instance set $Z$ and a convex, 1-Lipschitz and differentiable loss function $f : \mathbb{R}^d \times Z \to \mathbb{R}$ such that for one-pass SGD (either projected or unprojected; cf. Eq. (2) with $W = \mathbb{B}^d$ or $W = \mathbb{R}^d$ respectively) over $T = n$ steps initialized at $w_1 = 0$ with step size $\eta$, for*

*all $m = 1, \ldots, T$, the m-suffix averaged iterate has, with probability at least $\frac{1}{2}$ over the choice of the training sample,*

$$\widehat{F}(w_{T,m}) - \widehat{F}(\widehat{w}_*) = \Omega\left(\min\left\{\eta\sqrt{T} + 1/\eta T, 1\right\}\right), \tag{4}$$

$$and \quad \left|F(w_{T,m}) - \widehat{F}(w_{T,m})\right| = \Omega\left(\min\left\{\eta\sqrt{T} + 1/\eta T, 1\right\}\right). \tag{5}$$

**Discussion.** As noted in the introduction, both of the bounds above are tight up to logarithmic factors in view of matching upper bounds due to Bassily et al. (2020). For GD tuned for optimal convergence on the empirical risk, where $T = n$ and $\eta = \Theta(1/\sqrt{n})$, Theorem 1 gives an $\Omega(1)$ lower bound for the population error, which precludes any sample complexity upper bound for this algorithm of the form $O(d^p/\epsilon^q)$ unless $p \geq \frac{1}{2}$. In particular, this implies an $\Omega(\sqrt{d})$ lower bound the number of training examples required for GD to reach a nontrivial population risk. In contrast, lower bounds in previous work (Amir et al., 2021b) only implies an exponentially weaker $\Omega(\log d)$ dimension dependence in the sample complexity. We note however that there is still a small polynomial gap between our sample complexity lower bounds to the known (nearly tight) bounds for generic ERMs (Feldman, 2016; Carmon et al., 2023); we leave narrowing this gap as an open problem for future investigation.

More generally, with GD fixed to perform $T = n^\alpha, \alpha > 0$ steps, and setting $\eta$ so as to optimize the lower bound, the right-hand side in Eq. (3) becomes $\Theta(n^{-\alpha/4})$, which rules out any sample complexity upper bound of the form $O(d^p/\epsilon^q)$ unless it satisfies $\max\{2, \alpha + 1\}p + \frac{1}{4}\alpha q \geq 1$.[3] Specifically, we see that any dimension-free upper bound with $T = n$ must have at least an $1/\epsilon^4$ dependence on $\epsilon$; and that for matching the statistically optimal sample complexity rate of $1/\epsilon^2$, one must either run GD for $T = n^2$ steps or suffer a polynomial dimension dependence in the rate (e.g., for $T = n$ this dependence is at least $d^{1/4}$).

Similar lower bounds (up to a logarithmic factor) are obtained for SGD through Theorem 2, but for the empirical risk of the algorithm when tuned for optimal performance on the population risk with $T = n$. In this case, the bounds provide an exponential improvement in the dimension dependence over the recent results of Koren et al. (2022), showing that the "benign underfitting" phenomena they revealed for one-pass SGD is exhibited already in dimension polynomial in the number of training samples.

Finally, we remark that our restriction on $\eta$ is only meant for placing focus on the more common and interesting range of stepsizes in the context of stochastic optimization. It is not hard to extend the result of Theorems 1 and 2 to larger values of $\eta$ (in this case the lower bounds are $\Omega(1)$, the same rate the theorems give for $\eta = \Theta(1/\sqrt{T})$), in the same way this is done in previous work (e.g., Amir et al., 2021b; Koren et al., 2022).

## 3. Overview of constructions and proof ideas

In this section we outline the principal ideas leading to our main results and give an overview of the lower bound constructions; due to space constraints, all formal proofs are deferred to the supplementary material. As discussed above, the main technical contribution of this paper is in

---

3. To see this, let $r = \max\{2, \alpha + 1\}$ and note that for our construction $d = O(nT + n^2) = O(n^r)$ and $\epsilon = \Omega(n^{-\alpha/4})$; the sample complexity upper bound $O(d^p/\epsilon^q)$ can be therefore rewritten in terms of $n$ as $O(n^{rp+\alpha q/4})$, and since this should asymptotically upper bound the number of samples $n$, one must have that $rp + \frac{1}{4}\alpha q \geq 1$.

establishing the first $\Omega(\eta\sqrt{T})$ term in Eqs. (3) and (4) using a loss function in dimension polynomial in $n$ and $T$, and this is also the focus of our presentation in this section. In Sections 3.1 to 3.4 we focus on GD and describe the main ingredients and technical steps towards proving our first main result Theorem 1. The additional steps and adjustments needed to obtain our second main result concerning SGD (Theorem 2) are then discussed in Section 3.6.

Starting with GD and Theorem 1, recall that our goal is to establish a learning scenario where GD is likely to converge to a "bad ERM", namely a minimizer of the empirical risk whose population loss is large. We will do that in four steps: we will first establish that such a "bad ERM" actually exists; then, we will show how to make such a solution reachable by gradient steps from the origin; we next describe how the information required to identify this solution can be "memorized" by GD into its iterates; and finally, we show how to combine these components and actually drive GD towards a bad ERM.

## 3.1. A preliminary: existence of bad ERMs

Our starting point is the work of Feldman (2016) that demonstrated that in SCO, an empirical risk minimizer might fail to generalize, already in dimension linear in the number of training samples. Their approach was based on a construction of a set of unit vectors $U \subset \mathbb{R}^{d'}$ (for $d' = \Theta(n)$) of size $2^{\Omega(n)}$, that are "nearly orthogonal": the dot product between any two distinct $u, v \in U$ satisfies $|\langle u, v \rangle| \leq \frac{1}{8}$.[4] Then, they take the power set $Z = P(U)$ of $U$ as the sample space (namely, identifying samples with subsets of $U$), the distribution $\mathcal{D}$ to be uniform over $Z$, and the (convex, Lipschitz) loss $h_{\mathrm{F16}} : \mathbb{R}^{d'} \times Z \to \mathbb{R}$ to be defined as follows:

$$h_{\mathrm{F16}}(w, V) = \max\left\{\tfrac{1}{2}, \max_{u \in V}\langle u, w \rangle\right\}. \tag{6}$$

For this problem instance, they show that with constant probability over the training set sample $S = \{V_1, \ldots, V_n\} \overset{\mathrm{iid}}{\sim} \mathcal{D}^n$, there exists $u_0 \in U \setminus \bigcup_{i=1}^n V_i$ that is in fact an $\Omega(1)$-bad ERM (for which the generalization gap is $\Omega(1)$). To see why this is the case, note that, since every vector $u \in U$ is in every training example $V_i$ with probability $\frac{1}{2}$, the set $U$ (whose size is exponential in $n$) is large enough to guarantee the existence of a vector $u_0 \notin \bigcup_{i=1}^n V_i$ with constant probability. Consequently, the empirical loss of such $u_0$ equals $\frac{1}{2}$ (since $\langle u_0, v \rangle \leq \frac{1}{8} < \frac{1}{2}$ for any $v \in V_i$). However, for a fresh example $V \sim \mathcal{D}$, with probability $\frac{1}{2}$ it holds that $u_0 \in V$ in which case $h_{\mathrm{F16}}(u_0, V) = 1$, and thus the population risk of $u_0$ is at least $= \frac{1}{2} \cdot 1 + \frac{1}{2} \cdot \frac{1}{2} = \frac{3}{4}$.

## 3.2. Ensuring that bad ERMs are reachable by GD

As Feldman (2016) explain in their work, although there exists an ERM with a large generalization gap, it is not guaranteed that such a minimizer is at all reachable by gradient methods. This is because in their construction, the loss function $h_{\mathrm{F16}}$ remains flat (and equals $\frac{1}{2}$, see Eq. (6)) inside a ball of radius $\Omega(1)$ around the origin (where GD is initialized); hence it remains unclear how to steer GD with stepsize of order $\eta = O(1/\sqrt{T})$ away from this flat region. To address this challenge, we increase dimensionality and replicate Feldman's construction in $T$ orthogonal subspaces; which allows us to decrease, in each of the subspaces, the distance required to travel towards a bad ERM to only $O(\eta)$. Then, while each of these subspace ERMs is only $\Omega(\eta)$-bad, when taken together

---

4. The original construction Feldman (2016) satisfied slightly different conditions, which we adjust here for our analysis.

they constitute an $\Omega(1)$-bad ERM in the lifted space. Concretely, we introduce a loss function $h : \mathbb{R}^{d'} \times P(U) \to \mathbb{R}$ that resembles Feldman's function from Eq. (6) up to a minor adjustment:

$$h(w', V) = \max \left\{ \tfrac{3}{32}\eta, \max_{u \in V} \langle u, w' \rangle \right\}. \tag{7}$$

As in the original construction by Feldman, $V$ here ranges over subsets of a set $U \subseteq \mathbb{R}^{d'}$ of size $2^{\Omega(n)}$, the elements of which are nearly-orthogonal unit vectors. Then, we construct a loss function in dimension $d = Td'$ by applying $h$ in $T$ orthogonal subspaces of dimension $d'$, denoted $W^{(1)}, \ldots, W^{(T)}$, as follows:[5]

$$\ell_1(w, V) = \sqrt{\sum_{k=2}^{T} \left( h(w^{(k)}, V) \right)^2}. \tag{8}$$

Here and throughout, $w^{(k)}$ refers to the $k$'th orthogonal component of the vector $w$, that resides in the subspace $W^{(k)}$. Finally, the distribution $\mathcal{D}$ is again taken to be uniform over $Z = P(U)$, and a training set is formed by sampling $S = \{V_1, \ldots, V_n\} \sim \mathcal{D}^n$. As before, we know that with probability at least $\frac{1}{2}$, there exists a vector $u_0$ such that $u_0 \in U \setminus \bigcup_{i=1}^{n} V_i$. In addition (and unlike before), it can be shown that any vector $w$ satisfying $w^{(k)} = c\eta u_0$ for a sufficiently large constant $c > 0$ and $\Omega(T)$-many components $k$, is an $\Omega(1)$-bad ERM with respect to $\ell_1$. Further, the important point is that such bad ERMs are potentially reachable by GD.

### 3.3. Memorizing the dataset in the iterate

There is one notable obstacle to the plan we just described: the vector $u_0$ is determined by the full description of the training set, and it is unclear how to find such a vector through subgradients of the empirical loss, which is the only input GD has that carries information about the training set. To cope with this difficulty, we augment the domain $W$ with an "encoding subspace" denoted $W^{(0)}$ which is orthogonal to $W^{(1)}, \ldots, W^{(T)}$, and employ a mechanism that effectively *memorizes the full training set in the iterates* $w_t^{(0)}$ in a manner that can be decoded by the subgradient oracle.

We take $W^{(0)}$ to be of dimension $2n^2$, and augment samples with a number $j \in [n^2]$, drawn uniformly at random; namely, each sample in the training set is now a pair $(V_i, j_i) \in P(U) \times [n^2]$, for $i = 1, \ldots, n$. We then let $\phi : P(U) \times [n^2] \to W^{(0)}$ denote an encoding function such that $\phi(V, j)$ maps the set $V$ into the $j$'th (2-dimensional) subspace of the encoding space $W^{(0)}$. The role of $j$ is to ensure that with constant probability, different training examples are mapped to distinct subspaces of $W^{(0)}$. The encoding is then implemented within the optimization process through the following loss term:

$$\ell_2(w, (V, j)) := \langle -\phi(V, j), w^{(0)} \rangle. \tag{9}$$

Now, following the first step of GD, we have $w_2^{(0)} = \frac{\eta}{n} \sum_{i=1}^{n} \phi(V_i, j_i)$, and with constant probability each vector $\phi(V_i, j_i)$ is non-zero in distinct components of $w_2^{(0)}$. Proceeding, we represent every potential training set using a vector $\psi \in \Psi \subseteq \mathbb{B}^{2n^2}$, and define a mapping $\alpha : \mathbb{R}^{2n^2} \to U$ such that for all $\psi \in \Psi$, $\alpha(\psi)$ is some vector in $U$ that is not contained in the training set represented by $\psi$ (if such a vector exists). The desired gradient step is then induced by the additional loss term:

$$\ell_3(w) := \max \left\{ \delta_1, \max_{\psi \in \Psi} \left\{ \langle \psi, w^{(0)} \rangle - \beta \langle \alpha(\psi), w^{(1)} \rangle \right\} \right\}, \tag{10}$$

---

5. The summation starts at $k = 2$ due to technical reasons that will become apparent later in this proof sketch.

where $\beta, \delta_1 > 0$ are small predefined constants. For suitable choices of $\phi$ and $\alpha$ it can be shown $\psi^* \in \Psi$ that represents the actual training set $V_1, \ldots, V_n$ is realized as a unique maximizer in Eq. (10), which in turn triggers a gradient step along $u_0 := \alpha(\psi^*)$ in the subspace $W^{(1)}$. We note that while there exist rather straightforward approaches to construct $\phi, \alpha$, the challenge lies in designing such functions so that $u_0$ is realized as the *unique* subgradient (i.e., the gradient) of the loss function. Such a mechanism, established formally in Theorem 4, will serve to show that our lower bound is valid for *any* subgradient oracle (and not only for an adversarially chosen one), as well as for making the construction differentiable; we discuss this later on, in Section 3.5.

### 3.4. Making GD converge to a bad ERM

Our final task is to make GD converge to a "bad ERM" $w \in \mathbb{R}^d$ such that $w^{(k)} = c\eta u_0$ for a sufficiently large constant $c > 0$ and $\Omega(T)$-many values of $k$ (assuming $w_2^{(1)} = c_1 u_0$ as we detailed in the previous section). To this end, we employ a variation of a technique used in previous lower bound constructions (Bassily et al., 2020; Amir et al., 2021b; Koren et al., 2022) to induce gradient instability around the origin. Notably, in these prior instances, the potential directions of progress—analogous to vectors in our set $U$—were perfectly orthogonal (and thus, the dimension of space was required to be exponential in $n$). By contrast, in our scenario the vectors in $U$ are only approximately orthogonal, and directly applying previous approaches could lead to situations where gradient steps from consecutive iterations may interfere with progress made in correlated directions in previous iterations. Accordingly, our approach differs from previous works and incorporates a chain-like gradient step sequence, induced by the following additional loss term:

$$\ell_4(w) = \max \left\{ \delta_2, \max_{u \in U, \, 1 \le k < T} \left\{ \tfrac{3}{8} \langle u, w^{(k)} \rangle - \tfrac{1}{2} \langle u, w^{(k+1)} \rangle \right\} \right\}, \tag{11}$$

where $\delta_2 > 0$ is a small constant. The key idea here is that following the initialization stage, the inner maximization above is always attained at the same vector $u = u_0$, and for values of $k$ that increase by 1 in every iteration of GD. Consequently, subgradient steps with respect to this term will result in making a step towards $u_0$ in each of the components $w^{(1)}, w^{(2)}, \ldots$ one by one, avoiding interference between consecutive steps. At the end of this process, there are $\Omega(T)$ values of $k$ such that $w^{(k)} = \tfrac{1}{8} \eta u_0$, which is what we set to achieve.

In some more detail, assuming GD successfully reaches a vector $w_2 = w$ with $w^{(1)} = c_1 u_0$ and $w^{(k)} = 0$ for $k > 1$ ($c_1 > 0$ is a small constant), we have that the maximum in Eq. (11) is uniquely attained at $k = 1$ and $u = u_0$. Consequently, the subgradient of $\ell_4$ is a vector $g$ such that $g^{(1)} = \tfrac{3}{8} u_0$, $g^{(2)} = -\tfrac{1}{2} u_0$ (and $g^{(k)} = 0$ for $k \ne 1, 2$), and taking a subgradient step with stepsize $\eta$ results in $w^{(1)} = (\eta\beta - \tfrac{3\eta}{8}) u_0$ and $w^{(2)} = \tfrac{\eta}{2} u_0$ (for $k \ne 1, 2$, $w^{(k)}$ remains as is). In each subsequent iteration, the maximization in Eq. (11) is attained at an index $k$ for which $w^{(k)} = \tfrac{\eta}{2} u_0$ and at $u = u_0$ [6]. Subsequently, every gradient step adds $-\tfrac{3\eta}{8} u_0$ to $w^{(k)}$ and $\tfrac{\eta}{2} u_0$ to $w^{(k+1)}$ and results in $w^{(2)} = w^{(3)} = \ldots = w^{(k)} = \tfrac{\eta}{8} u_0$ and $w^{(k+1)} = \tfrac{\eta}{2} u_0$ (whereas for all $s > k + 1$, $w^{(s)}$ remains zero).

Finally, we note that the GD dynamics we described ensure that the iterates $w_1, \ldots, w_T$ remain strictly within the unit ball $\mathbb{B}^d$, even when the algorithm does not employ any projections. As a consequence, the construction we described applies equally to a projected version of GD, with projections to the unit ball, and the resulting lower bound will apply to both versions of the algorithm.

---

6. This follows from direct computation and considering the near-orthogonality of vectors in $U$.

### 3.5. Putting things together

We can now integrate the ideas described in Sections 3.1 to 3.4 into a construction of a learning problem where GD overfits the training data (with constant probability), that would serve to prove our lower bound. To summarize this construction:

- The examples in the learning problem are parameterized by pairs $(V, j) \in Z := P(U) \times [n^2]$, where $U$ is the set of nearly-orthogonal vectors described in Section 3.1, and $P(U)$ is its power set;

- The population distribution $\mathcal{D}$ is uniform over pairs $(V, j) \in Z$, namely $V \sim \text{Unif}(P(U))$ (i.e., $V$ is formed by including every element $u \in U$ independently with probability $\frac{1}{2}$) and $j \sim \text{Unif}([n^2])$;

- The loss function in this construction, $f : W \times (P(U) \times [n^2]) \to \mathbb{R}$, is then given by:

$$\forall (V, j) \in Z, \qquad f(w, (V, j)) := \ell_1(w, V) + \ell_2(w, (V, j)) + \ell_3(w) + \ell_4(w), \qquad (12)$$

with the terms $\ell_1, \ell_2, \ell_3, \ell_4$ as defined in Eqs. (8) to (11) respectively.

With a suitable choice of parameters, this construction serves to prove Theorem 1. We remark that, while $f$ in this construction is convex and $O(1)$-Lipschitz, it is evidently non-differentiable. For obtaining a construction with a differentiable objective that maintains the same lower bound and establishes the Theorem 1 fully, we add one final step of randomized smoothing of the objective. This argument hinges on the fact that the subgradients of $f$ are unique *along any possible trajectory of GD*, so that smoothing in a sufficiently small neighborhood would preserve gradients along any such trajectory (and thus does not affect the the dynamics of GD), while making the objective differentiable everywhere. The full proof of Theorem 1 is given in Appendix A.

### 3.6. Adapting the construction for SGD

Moving on to discuss our second main result for SGD, we provide here a brief overview of the necessary modifications upon the construction for GD to establish the lower bound for SGD in Theorem 2; further details can be found in Appendix B. In the case of SGD, our goal is to establish underfitting: namely, to show that the algorithm may converge to a solution with an excessively large empirical risk despite successfully converging on the population risk.

The main ideas leading to our construction for SGD are similar to what we discussed above, but there are several necessary modifications that arise from the fact that, whereas in GD the entire training set is revealed already in the first iteration, in SGD it is revealed sequentially, one training sample at a time. In particular, unlike in the case of GD where it is possible to identify a bad ERM $u_0$ after the first step of the algorithm and steer the algorithm in this direction in every subspace $W^{(1)}, W^{(2)}, \ldots$, for SGD the required progress direction in $W^{(t)}$, represented as a "bad solution" $u_t$, can be only determined in the $t$'th step based on the encoded training set up to that point, $V_1, \ldots, V_{t-1}$. As a result, it is crucial to modify the loss function such that the process of decoding such $u_t$ from $V_1, \ldots, V_{t-1}$ occurs in every iteration $t$.

Another essential adjustment involves identifying a solution with a large generalization gap (namely large empirical risk, low population risk) and guiding the SGD iterates to converge to such a solution. Considering the function $\ell_1$ defined in Eq. (8), such a solution is represented by a vector $u \in U$ that appears in all of the sets $V_1, \ldots, V_n$ in the training sample. However, since $u_t$

cannot depend on future examples, our goal within every subspace $W^{(t)}$ is to take a single gradient step towards a vector $u_t$ present only in sets up to that point, namely in $\bigcap_{i=1}^{t-1} V_i$ (note that such $u_t$ maximizes the corresponding loss functions $\ell_1(w, V_1) \ldots \ell_1(w, V_{t-1})$). Additionally, to ensure that gradients for future loss functions remain zero and do not affect the algorithm's dynamics, it is necessary to guarantee that $u_t \in \bigcap_{i=t}^{n} \overline{V}_i$; in other words, we are looking for a solution $u_t \in \bigcap_{i=1}^{t-1} V_i \cap \bigcap_{i=t}^{n} \overline{V}_i$. For ensuring that such a vector actually exists (with constant probability), we lift the dimension of the set $U$ and the subspaces $\{W^{(k)}\}_{k=1}^{n}$ to $d' = \Theta(n \log n)$ (instead of $\Theta(n)$ as before) and modify the distribution $\mathcal{D}$ so as to have that $V$ is sampled such that every element $u \in U$ is included in $V$ independently with probability $1/4n^2$.

With these adaptations in place, we can obtain Theorem 2; we include the full technical details in Appendix B.

## Acknowledgments

This project has received funding from the European Research Council (ERC) under the European Union's Horizon 2020 research and innovation program (grant agreements No. 101078075; 882396). Views and opinions expressed are however those of the author(s) only and do not necessarily reflect those of the European Union or the European Research Council. Neither the European Union nor the granting authority can be held responsible for them. This work received additional support from the Israel Science Foundation (ISF, grant numbers 2549/19 and 3174/23), from the Len Blavatnik and the Blavatnik Family foundation, and from the Adelis Foundation.

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

## Appendix A. Overfitting of GD: Proof of Theorem 1

In this section, we provide a formal proof of our main result for GD. We first establish a lower bound of $\Omega(\eta\sqrt{T})$ for the population loss of GD given a non-differentiable loss (Theorem 5). The proof of Theorem 1 given at the end of this section then extends the result to a differentiable loss and combines it with a standard optimization lower bound. Unless stated otherwise, proofs for lemmas below are deferred to Appendix A.4.

For the first step, for a dimension $d'$ that will be set later, we use a set of approximately orthogonal vectors in $\mathbb{R}^{d'}$ with size (at least) exponential in $d'$, the existence of which is given by the following lemma, adapted from Feldman (2016).

**Lemma 3** *For any $d' \geq 256$, there exists a set $U_{d'} \subseteq \mathbb{R}^{d'}$, with $|U_{d'}| \geq 2^{d'/178}$, such that all $u \in U_{d'}$ are of unit length $\|u\| = 1$, and for all $u, v \in U_{d'}, u \neq v$, it holds that $|\langle u, v\rangle| \leq \frac{1}{8}$.*

Now, let $n$ be the number of examples in the training set. We define the set $U := U_{d'}$ to be a set as specified by Theorem 3 for $d' = 178n$. Then, as outlined in Section 3, we define the sample space $Z := \{(V, j) : V \subseteq U, j \in [n^2]\}$ and the distribution $\mathcal{D}$ as the uniform distribution. Our next lemma asserts the existence of suitable encoding ($\phi$) and selection ($\alpha$) functions.

**Lemma 4** *Let $n, d \geq 1$ and a set $U \subseteq \mathbb{B}^d$. Let $P(U)$ be the power set of $U$. Then, there exist a set $\Psi \subseteq \mathbb{R}^{2n^2}$, a number $0 < \epsilon < \frac{1}{n}$ and two mappings $\phi : P(U) \times [n^2] \to \mathbb{R}^{2n^2}, \alpha : \mathbb{R}^{2n^2} \to U$ such that,*

(i) *For every $j \in [n^2]$ and $V \subseteq U$, it holds $\|\phi(V, j)\| \leq 1$;*

(ii) *For every $\psi \in \Psi$, it holds $\|\psi\| \leq 1$;*

(iii) *Let $V_1, \ldots, V_n$ be arbitrary subsets of $U$. If $j_1, \ldots, j_n$ hold that $j_i \neq j_k$ for $i \neq k$, $\psi^* = \max_{\psi \in \Psi} \langle \psi, \frac{1}{n} \sum_{i=1}^{n} \phi(V_i, j_i) \rangle$ is that,*

- 
$$\left\langle \psi^*, \frac{1}{n} \sum_{i=1}^{n} \phi(V_i, j_i) \right\rangle > \frac{7}{8n};$$

- *For every $\psi \in \Psi, \psi \neq \psi^*$:*

$$\left\langle \psi^*, \frac{1}{n} \sum_{i=1}^{n} \phi(V_i, j_i) \right\rangle \geq \left\langle \psi, \frac{1}{n} \sum_{i=1}^{n} \phi(V_i, j_i) \right\rangle + \epsilon;$$

- *If $\bigcup_{i=1}^{n} V_i \neq U$, then it holds that:*

$$\alpha(\psi^*) = v_{i^*} \in U \setminus \bigcup_{i=1}^{n} V_i, \quad \text{for } i^* = \min \left\{ i : v_i \in U \setminus \bigcup_{i=1}^{n} V_i \right\}.$$

We are now ready to state our main theorem for this section, providing the principal lower bound guarantees.

**Theorem 5** *Assume that $n > 0$, $T > 3200^2$ and $\eta \leq \frac{1}{\sqrt{T}}$. Consider the distribution $\mathcal{D}$ and the loss function $f$ defined in Section 3.5 instantiated with $\Psi, \phi, \alpha$ from Theorem 4, $d = 178nT + 2n^2$, $\varepsilon = \frac{1}{n^2}(1 - \cos(\frac{\pi}{2|P(U)|}))$, $\beta = \frac{\epsilon}{4T^2}$, $\delta_1 = \frac{\eta}{2n}$ and $\delta_2 = \frac{3\eta\beta}{16}$. Then $f$ is convex and 5-Lipschitz, and for Unprojected GD (cf. Eq. (1) with $W = \mathbb{R}^d$) on $\widehat{F}$, initialized at $w_1 = 0$ with step size $\eta$, we have, with probability at least $\frac{1}{6}$ over the choice of the training sample:*

*(i) The iterates of GD remain within the unit ball, namely $w_t \in \mathbb{B}^d$ for all $t = 1, \ldots, T$;*

*(ii) For all $m = 1, \ldots, T$, the m-suffix averaged iterate has:*

$$F(w_{T,m}) - F(w_*) = \Omega(\eta\sqrt{T}).$$

The lemmas given next will constitute the proof of Theorem 5. Before presenting them, for convenience, we repeat the loss function definition along with the constants specified in our theorem above:

$$f(w, V) = \sqrt{\sum_{k=2}^{T} \left(\max\left\{\tfrac{3}{32}\eta, \max_{u \in V}\langle u, w^{(k)}\rangle\right\}\right)^2} \tag{$\ell_1$}$$

$$+ \langle -\phi(V, j), w^{(0)}\rangle \tag{$\ell_2$}$$

$$+ \max\left\{\delta_1, \max_{\psi \in \Psi}\left\{\langle \psi, w^{(0)}\rangle - \beta\langle\alpha(\psi), w^{(1)}\rangle\right\}\right\} \tag{$\ell_3$}$$

$$+ \max\left\{\delta_2, \max_{u \in U, \, 1 \leq k < T}\left\{\tfrac{3}{8}\langle u, w^{(k)}\rangle - \tfrac{1}{2}\langle u, w^{(k+1)}\rangle\right\}\right\}, \tag{$\ell_4$}$$

where,

$$d = 178nT + 2n^2,$$

$$\varepsilon = \frac{1}{n^2}(1 - \cos(\frac{\pi}{2|P(U)|})),$$

$$\beta = \frac{\epsilon}{4T^2},$$

$$\delta_1 = \frac{\eta}{2n},$$

$$\delta_2 = \frac{3\eta\beta}{16}.$$

**Lemma 6** *For every $(V, j) \in Z$, the loss function $f(w, (V, j))$ as defined by Theorem 5 is convex and 5-Lipschitz over $\mathbb{R}^d$ with respect to its first argument.*

Next, we formulate the "good event" — conditioned on this event the optimization dynamics of GD will lead to a bad ERM:

$$\mathcal{E} := \left\{\bigsqcup_{i=1}^{n} V_i \neq U\right\} \cap \left\{j_k \neq j_l, \, \forall k \neq l\right\}. \tag{13}$$

In words, under the event $\mathcal{E}$ there exists at least one "bad direction" (which is a vector in the set $U \setminus \bigcup_{i=1}^{n} V_i$) and there is no collision between the indices $j_1, \ldots, j_n$. In the following lemma we show that $\mathcal{E}$ holds with constant probability.

**Lemma 7** *For the event $\mathcal{E}$ defined in Eq. (13), it holds that $\Pr(\mathcal{E}) \geq \frac{1}{6}$.*

The next lemma is the key component in the proof of Theorem 5, and characterizes the trajectory of GD when applied to the empirical risk $\widehat{F}$, conditioned on the good event. The proof is provided in Appendix A.1.

**Lemma 8** *Assume the conditions of Theorem 5, and consider the iterates of unprojected GD on $\widehat{F}$, with step size $\eta \leq 1/\sqrt{T}$ initialized at $w_1 = 0$. Under the event $\mathcal{E}$, we have for all $t \geq 5$ that*

$$w_t^{(k)} = \begin{cases} \frac{\eta}{n} \sum_{i=1}^{n} \phi(V_i, j_i) & k = 0; \\ \left(-\frac{3}{8} + \frac{t-2}{4} \frac{\epsilon}{T^2}\right) \eta u_0 & k = 1; \\ \frac{1}{8} \eta u_0 & 2 \leq k \leq t - 3; \\ \frac{1}{2} \eta u_0 & k = t - 2; \\ 0 & t - 1 \leq k \leq T, \end{cases} \tag{14}$$

*where $u_0$ is a vector such that $u_0 \in U \setminus \bigcup_{i=1}^{n} V_i$. Furthermore, $\|w_t\| \leq 1$ for all $t \in [T]$.*

Now we can turn to prove Theorem 5. Here we prove the lower bound for the case of suffix averaging with $m = 1$, namely, when the output solution is the final iterate $w_T$ of GD; the full proof for the more general case can be found in Appendix A.3.

**Proof** [of Theorem 5 ($m = 1$ case)] We prove the theorem under the condition that $\mathcal{E}$ occurs, which happens w.p. $\geq \frac{1}{6}$ by Theorem 7. First, note that $f$ is convex and 5-Lipschitz by Theorem 6, and that (i) is implied immediately by the fact that $\|w_t\| \leq 1$, as assured by Theorem 8.

Next, consider $w_T$ that takes the form stated in Eq. (14) given by Theorem 8. Note that if a vector $v \in U$ is in a set $V \subseteq U$, it holds that $\max_{u \in V} \langle u, v \rangle = 1$. However, if $v \notin V$, it holds that $\max_{u \in V} \langle u, v \rangle = \frac{1}{8}$. As a result, by the fact that every vector for a fresh pair $(V, j) \sim D$, $u_0 \in U$ is in $V$ with probability $\frac{1}{2}$, the following holds:

$$\mathbb{E}_V \sqrt{\sum_{k=2}^{T} \max\left\{\frac{3\eta}{32}, \max_{u \in V} \langle u, w_T^{(k)} \rangle\right\}^2} \geq \mathbb{E}_V \sqrt{\sum_{k=2}^{T-3} \max\left\{\frac{3\eta}{32}, \max_{u \in V} \langle u, w_T^{(k)} \rangle\right\}^2}$$

$$= \mathbb{E}_V \sqrt{(T - 4) \max\left\{\frac{3\eta}{32}, \max_{u \in V} \langle u, \frac{\eta}{8} u_0 \rangle\right\}^2}$$

$$= \frac{\eta \sqrt{T - 4}}{8} \mathbb{E}_V \max\left\{\frac{3}{4}, \max_{u \in V} \langle u, u_0 \rangle\right\}$$

$$\geq \frac{\eta \sqrt{T - 4}}{8} \left(\frac{3}{4} \Pr(u_0 \notin V) + \Pr(u_0 \in V)\right)$$

$$= \frac{7\eta}{64} \sqrt{T - 4}.$$

Moreover, again by Theorem 8, we notice that for every $t$, $V \subseteq U$ and $j \in [n^2]$,

$$\ell_2(w_t, (V, j)) \geq -\|w_t^{(0)}\| \geq -\eta, \ell_3(w_t) \geq \delta_1 \text{ and } \ell_4(w_t) \geq \delta_2,$$

thus, it holds that:

$$F(w_T) \geq \frac{7\eta}{64} \sqrt{T - 4} + \delta_1 + \delta_2 - \eta;$$

$$F(w_*) \leq F(0) \leq \frac{3\eta}{32}\sqrt{T} + \delta_1 + \delta_2 + \eta.$$

Then, since $T \geq 100$ by assumption, we have $\frac{6\sqrt{T}}{7\sqrt{T-4}} \leq 9/10$, thus:

$$F(w_T) - F(w_*) \geq \frac{7\eta}{64}\sqrt{T-4} - \frac{3\eta}{32}\sqrt{T} \geq \frac{7\eta\sqrt{T-4}}{640},$$

which completes the proof. ∎

The remaining elements for Theorem 1 are establishing Theorem 5 for a differentiable loss function, and merging the result with a standard optimization lower bound of $\Omega\left(\min\left(\frac{1}{\eta T}, 1\right)\right)$ given in Theorem 38 in Appendix D.

**Proof** [of Theorem 1] Let $\beta, \epsilon, \delta_1, \delta_2$, and $f, \mathcal{D}$ be those specified in Theorem 5, and define

$$\tilde{f}(w, (V, j)) := \mathbb{E}_{v \sim \text{Unif}(\mathbb{B}^d)} \left[ f(w + \delta v, (V, j)) \right], \tag{15}$$

where

$$\delta := \frac{\eta \beta}{32}. \tag{16}$$

Further, denote the empirical loss and the population loss with respect to the loss function $\tilde{f}$ as $\widehat{\tilde{F}}(w) = \frac{1}{n}\sum_{i=1}^{n}\tilde{f}(w, (V_i, j_i))$ and $\tilde{F}(w) = \mathbb{E}_{(V,j)\sim\mathcal{D}}\tilde{f}(w, (V, j))$, respectively. We have that this loss function is convex and Lipschitz, and that the GD iterates identify with those of the original, non-smoothed loss.

**Lemma 9** *For every $(V, j) \in Z$, the loss function $\tilde{f}$ is differentiable, convex and 1-Lipschitz with respect to its first argument and over $\mathbb{R}^d$.*

**Lemma 10** *Let $w_t, \tilde{w}_t$ be the iterates of Unprojected GD with step size $\eta \leq \frac{1}{\sqrt{T}}$ and $w_1 = 0$, on $\widehat{F}$ and $\widehat{\tilde{F}}$ respectively. Then, if $\mathcal{E}$ occurs, for every $t \in [T]$, it holds that $w_t = \tilde{w}_t$.*

Now let $\overline{w_{T,m}}$ be the $m$-suffix average of $GD$ when is applied on $\widehat{F}$. Let $\overline{w_*} = \arg\min_w F(w)$. By Theorem 10, we know that, with probability of at least $\frac{1}{6}$, $\mathcal{E}$ occurs and $w_{T,m} = \overline{w_{T,m}}$. Then, by Theorem 5 and the fact that $\|\tilde{f} - f\|_\infty \leq 5\delta$ (see Theorem 35),

$$\begin{aligned}
\frac{\eta}{3200}\sqrt{T} &\leq F(\overline{w_{T,m}}) - F(\overline{w_*}) \\
&= F(w_{T,m}) - F(\overline{w_*}) \\
&\leq \tilde{F}(w_{T,m}) + 5\delta - \tilde{F}(\overline{w_*}) + 5\delta \\
&\leq \tilde{F}(w_{T,m}) + 5\delta - \tilde{F}(w_*) + 5\delta,
\end{aligned}$$

hence,

$$\begin{aligned}
\tilde{F}(w_{T,m}) - \tilde{F}(w_*) &\geq \frac{\eta}{3200}\sqrt{T} - \frac{10\eta\epsilon}{128T^2} \\
&\geq \frac{\eta}{3200}\sqrt{T} - \frac{\eta}{10T^2}
\end{aligned}$$

$$\geq \frac{\eta}{6400}\sqrt{T} \qquad\qquad (T \geq 30)$$

$$=: C_1 \eta \sqrt{T}.$$

Further, by Theorem 38, we know that for Unprojected $GD$ and $d_2 = \max(25\eta^2 T^2, 1)$, there exist a constant $C_2$ and a deterministic loss function $\tilde{f}^{\mathrm{OPT}} : \mathbb{R}^{d_2} \to \mathbb{R}$ such that

$$\tilde{f}^{\mathrm{OPT}}(w_{T,m}) - \tilde{f}^{\mathrm{OPT}}(w_*) \geq C_2 \min\left(1, \frac{1}{\eta T}\right).$$

Now, let $C = \frac{1}{2}\min(C_1, C_2)$. If $\eta \geq T^{-\frac{3}{4}}$, then, $\eta\sqrt{T} \geq \min(1, \frac{1}{\eta T})$, and we get,

$$\tilde{F}(w_{T,m}) - \tilde{F}(w_*) \geq C\left(\eta\sqrt{T} + \min\left(1, \frac{1}{\eta T}\right)\right) \geq C\left(\min\left(1, \eta\sqrt{T} + \frac{1}{\eta T}\right)\right).$$

Otherwise, we get that,

$$\tilde{f}^{\mathrm{OPT}}(w_{T,m}) - \tilde{f}^{\mathrm{OPT}}(w_*) \geq C\left(\eta\sqrt{T} + \min\left(1, \frac{1}{\eta T}\right)\right) \geq C\left(\min\left(1, \eta\sqrt{T} + \frac{1}{\eta T}\right)\right).$$

Since in both cases, by Theorems 8 and 38, $w_t \in \mathbb{B}^d$ for every $t \in [T]$, the theorem is applicable also for Projected GD. Finally, we scale the loss by $1/5$, and note that the iterates of GD with step size $\eta$ on $\widetilde{F}$ behave identically to those with step size $\frac{1}{5}\eta$ on the scaled version. This completes the proof. ∎

### A.1. GD Dynamics: Proof of Theorem 8

To prove Theorem 8, we break down to the different components of the loss and analyze how the terms $\ell_1, \ell_3$ and $\ell_4$ affects the dynamics of $GD$ under the event $\mathcal{E}$. For each of these components, which involve maximum over linear functions, we show which term achieves the maximum value for each $w_t$ and derive the expressions for the gradients at those points by the maximizing terms. First, we show that under this event, the gradients of $\ell_1$ do not affect the dynamics since in any iteration $t$ the gradient of $\ell_1$ is zero, as stated in the following lemma.

**Lemma 11** *Assume the conditions of Theorem 5 and the event $\mathcal{E}$. Let $0 \leq c \leq \frac{1}{2}$, and let $w \in \mathbb{R}^d$ be such that for every $2 \leq k \leq T$, $w^{(k)} = c\eta u_0$, where $u_0 \in U \setminus \bigcup_{i=1}^n V_i$. Then, for every $i \in [n]$, it holds that*

(i) *for every $k \geq 2$, it holds $\max_{u \in V_i}\langle w^{(k)}, u\rangle \leq \frac{\eta}{16}$;*

(ii) *$\ell_1(\cdot, V_i)$ is differentiable at $w$ and $\nabla\ell_1(w, V_i) = 0$.*

**Proof** For the first part, we know that, for every $2 \leq k \leq T$, $w^{(k)} = c\eta u_0$, where $c \leq \frac{1}{2}$. In addition, by the facts that $u_0 \in U \setminus \bigcup_{i=1}^n V_i$ and that for every $u \neq v \in U$, it holds that $\langle u, v\rangle \leq \frac{1}{8}$ (see Theorem 3), we get for every $i$, $\max_{u \in V_i}\langle u_0, u\rangle \leq \frac{1}{8}$, thus, for every $i$ and $k \geq 2$,

$$\max_{u \in V_i}\langle u, w^{(k)}\rangle = \max_{u \in V_i}\langle u, c\eta u_0\rangle \leq \frac{1}{8} \cdot c\eta \leq \frac{\eta}{16}.$$

For the second part, for every sub-gradient $g(w, V_i) \in \partial \ell_1(w, V_i)$, there exists a sub gradient $g_h(w, V_i) \in \partial \left( \sum_{k=2}^{T} \max \left( \frac{3\eta}{32}, \max_{u \in V} \langle u, w^{(k)} \rangle \right)^2 \right)$ such that

$$g(w, V_i) = \frac{g_h(w, V_i)}{2\sqrt{\sum_{k=2}^{T} \max \left( \frac{3\eta}{32}, \max_{u \in V_i} \langle u w^{(k)} \rangle \right)^2}}.$$

Then, since for every $k$, it holds that $\max_{u \in U} \langle w^{(k)}, u_0 \rangle \leq \frac{\eta}{16} < \frac{3\eta}{32}$, we have $g_h(w, V_i) = 0$, hence $\nabla \ell_1(w, V_i) = 0$. ∎

Next, for the term $\ell_3$, as outlined in Section 3.3, it is used for identifying the actual training set $S = \{(V_i, j_i)\}_{i=1}^{n}$ given an encoding $\psi^* = \frac{1}{n} \sum_{i=1}^{n} \phi(V_i, j_i)$ in the iterate $w_t^{(0)}$ and ensuring a performance of gradient step in $W^{(1)}$ towards a corresponding vector $u_0 \in U \setminus \bigcup_{i=1}^{n} V_i$ in the following iteration. It is done by getting $\psi^*$ as a maximum of linear functions (with positive constant margin) over the set $\Psi$ which contains all possible encoded datasets. This idea is formalized in the following lemma.

**Lemma 12** *Assume the conditions of Theorem 5 and the event $\mathcal{E}$. Let $\psi^* = \frac{1}{n} \sum_{i=1}^{n} \phi(V_i, j_i)$ and $w \in \mathbb{R}^d$ be such $w^{(0)} = \eta \psi^*$, and let $w^{(1)} = c \eta u_0$ for $|c| \leq 1$ and $u_0 \in U \setminus \bigcup_{i=1}^{n} V_i$. Then*

*(i) For every $\psi \in \Psi$, $\psi \neq \psi^*$:*

$$\langle w^{(0)}, \psi^* \rangle - \frac{\epsilon}{4T^2} \langle \alpha(\psi^*), w^{(1)} \rangle > \langle w^{(0)}, \psi \rangle - \frac{\epsilon}{4T^2} \langle \alpha(\psi), w^{(1)} \rangle + \frac{\eta \epsilon}{4};$$

*(ii) For $\psi = \psi^*$, it holds that*

$$\langle w^{(0)}, \psi^* \rangle - \frac{\epsilon}{4T^2} \langle \alpha(\psi^*), w^{(1)} \rangle > \delta_1 + \frac{\eta}{16n};$$

*(iii) $\ell_3$ is differentiable at $w$ and the gradient is given by:*

$$(\nabla \ell_3(w))^{(k)} = \begin{cases} \psi^* & k = 0; \\ -\frac{\epsilon}{4T^2} u_0 & k = 1; \\ 0 & otherwise. \end{cases}$$

**Proof** For the first part, by Theorem 4, the fact that for every $\psi$, $\|\alpha(\psi)\| \leq 1$, and by $\|w^{(1)}\| \leq \eta$, for every $\psi \in \Psi$, $\psi^* = \frac{1}{n} \sum_{i=1}^{n} \phi(V_i, j_i)$ holds,

$$\begin{aligned}
\langle w^{(0)}, \psi^* \rangle - \frac{1}{4} \frac{\epsilon}{T^2} \langle \alpha(\psi^*), w^{(1)} \rangle &\geq \langle \frac{\eta}{n} \sum_{i=1}^{n} \phi(V_i, j_i), \psi^* \rangle - \frac{\eta \epsilon}{4} \\
&\geq \eta \langle \frac{1}{n} \sum_{i=1}^{n} \phi(V_i, j_i), \psi^* \rangle - \frac{\eta \epsilon}{4} \\
&\geq \eta \langle \frac{1}{n} \sum_{i=1}^{n} \phi(V_i, j_i), \psi \rangle + \eta \epsilon - \frac{\eta \epsilon}{4} \qquad \text{(Theorem 4)}
\end{aligned}$$

$$= \eta \langle \frac{1}{n} \sum_{i=1}^{n} \phi(V_i, j_i), \psi \rangle + \frac{\eta \epsilon}{2} + \frac{\eta \epsilon}{4}$$

$$> \langle w^{(0)}, \psi \rangle - \frac{1}{4} \frac{\epsilon}{T^2} \langle \alpha(\psi), w^{(1)} \rangle + \frac{\eta \epsilon}{4}.$$

Further (again by Theorem 4),

$$\arg \max_{\psi \in \Psi} \left( \langle w^{(0)}, \psi \rangle - \frac{1}{4} \frac{\epsilon}{T^2} \langle \alpha(\psi), w^{(1)} \rangle \right) = \psi^*.$$

For the second part, by the fact that $\epsilon < \frac{1}{n}$ and Theorem 4,

$$\langle w^{(0)}, \psi^* \rangle - \frac{1}{4} \frac{\epsilon}{T^2} \langle \alpha(\psi^*), w^{(1)} \rangle \geq \frac{7\eta}{8n} - \frac{\eta}{4n} > \frac{\eta}{2n} + \frac{\eta}{16n} = \delta_1 + \frac{\eta}{16n}.$$

Now, by $\mathcal{E}$, for $u_0$ which is the $u$ with the minimal index in $U \setminus \bigcup_{i=1}^{n} V_i$, we have:

$$\alpha(\psi^*) = u_0 \in U \setminus \bigcup_{i=1}^{n} V_i.$$

As a result, by the fact that the maximum is attained uniquely at $\psi^*$, we derive that,

$$\nabla \ell_3(w)^{(k)} = \begin{cases} \frac{1}{n} \sum_{i=1}^{n} \phi(V_i, j_i) & k = 0 \\ -\frac{1}{4} \frac{\epsilon}{T^2} u_0 & k = 1 \\ 0 & \text{otherwise.} \end{cases}$$

■

Finally, for $\ell_4$, as detailed in Section 3.4, the role of this term is to make the last iterate $w_T$ hold $w_T^{(k)} = \frac{\eta}{8} \eta u_0$ for $\Omega(T)$ many sub-spaces $W^{(k)}$. In the following lemma, we show that in every iteration $t$, every gradient step increases the amount of such $k$s by 1, namely, in every iteration $t$, the maximum of $\ell_4$ is attained at $u = u_0$ and index $k_t = \arg \max\{k : w_t^{(k)} \neq 0\}$, which increases by 1 in every iteration, making the $w_{t+1}^{(k_t)} = \frac{\eta}{8} \eta u_0$.

**Lemma 13** *Assume the conditions of Theorem 5 and the event $\mathcal{E}$. Let $w \in \mathbb{R}^d$, $u_0 \in U \setminus \bigcup_{i=1}^{n} V_i$ and $3 \leq m < T$ be such that $w^{(1)} = c\eta u_0$ for $-\frac{3}{8} \leq c \leq 0$, $w^{(k)} = \frac{\eta}{8} u_0$ for every $2 \leq k \leq m - 1$, $w^{(m)} = \frac{\eta}{2} u_0$ and $w^{(k)} = 0$ for every $k \geq m + 1$. Then, it holds that,*

*(i) For every pair $u \in U$ and $k < T$ such that $k \neq m$ or $u \neq u_0$,*

$$\tfrac{3}{8} \langle u_0, w^{(m)} \rangle - \tfrac{1}{2} \langle u_0, w^{(m+1)} \rangle > \tfrac{3}{8} \langle u, w^{(k)} \rangle - \tfrac{1}{2} \langle u, w^{(k+1)} \rangle + \tfrac{\eta}{64}$$

*(ii)*

$$\tfrac{3}{8} \langle u_0, w^{(m)} \rangle - \tfrac{1}{2} \langle u_0, w^{(m+1)} \rangle > \delta_2 + \tfrac{\eta}{64}.$$

*(iii) $\ell_4$ is differentiable at $w$ and the gradient is given as follows:*

$$\left( \nabla \ell_4(w) \right)^{(k)} = \begin{cases} \frac{3}{8} u_0 & k = m; \\ -\frac{1}{2} u_0 & k = m + 1; \\ 0 & \text{otherwise.} \end{cases}$$

**Proof** We show that the maximum is attained uniquely at $k = m$ and $u = u_0$. We repeatedly use that $|\langle u, u_0 \rangle| \leq 1/8$ for $u \in U, u \neq u_0$ (see Theorem 3). For $k = 1$ and every $u \in U$,

$$\frac{3}{8}\langle u, w_t{}^{(1)} \rangle - \frac{1}{2}\langle u, w_t{}^{(2)} \rangle = \frac{3}{8}\langle u, c\eta u_0 \rangle - \frac{1}{2}\langle u, \frac{\eta}{8}u_0 \rangle \leq \frac{9\eta}{512} + \frac{\eta}{128} = \frac{13\eta}{512}.$$

Moreover, for every $2 \leq k \leq m - 2$ and every $u \in U$,

$$\frac{3}{8}\langle u, w^{(k)} \rangle - \frac{1}{2}\langle u, w^{(k+1)} \rangle = \frac{3}{8}\langle u, \frac{\eta}{8}u_0 \rangle - \frac{1}{2}\langle u, \frac{\eta}{8}u_0 \rangle \leq \frac{3\eta}{64} + \frac{\eta}{128} = \frac{7\eta}{128}.$$

For $k = m - 1$ and every $u \in U$,

$$\frac{3}{8}\langle u, w^{(m-1)} \rangle - \frac{1}{2}\langle u, w^{(m)} \rangle = \frac{3}{8}\langle u, \frac{\eta}{8}u_0 \rangle - \frac{1}{2}\langle u, \frac{\eta}{2}u_0 \rangle \leq \frac{3\eta}{64} + \frac{\eta}{32} = \frac{5\eta}{64}.$$

For $k = m$ and $u = u_0$,

$$\frac{3}{8}\langle u, w^{(m)} \rangle - \frac{1}{2}\langle u, w^{(m+1)} \rangle = \frac{3}{8}\langle u_0, \frac{\eta}{2}u_0 \rangle - \frac{1}{2}\langle u_0, 0 \rangle = \frac{3\eta}{16}.$$

For $k = m$ and $u \neq u_0$,

$$\frac{3}{8}\langle u, w^{(m)} \rangle - \frac{1}{2}\langle u', w^{(m+1)} \rangle = \frac{3}{8}\langle u, \frac{\eta}{2}u_0 \rangle - \frac{1}{2}\langle u', 0 \rangle \leq \frac{3\eta}{128}.$$

For every $m + 1 \leq k < T - 1$ and every $u \in U$,

$$\frac{3}{8}\langle u, w^{(k)} \rangle - \frac{1}{2}\langle u', w^{(k+1)} \rangle = 0.$$

Moreover, since $T \geq 4, \eta < 1, \epsilon < 1, \delta_1 \leq \frac{3\eta}{1024}$, and

$$\frac{3}{8}\langle u, w^{(m)} \rangle - \frac{1}{2}\langle u, w^{(m+1)} \rangle = \frac{3\eta}{16} > \delta_2 + \frac{\eta}{64}.$$

We derive that,

$$\nabla \ell_4(w)^{(k)} = \begin{cases} \frac{3}{8}u_0 & k = m \\ -\frac{1}{2}u_0 & k = m + 1 \\ 0 & \text{otherwise.} \end{cases}$$

∎

The final components required for the proof of Theorem 8 are direct computations for the initial iterates, which are detailed in Appendix A.2. With these in place, we are now ready for the proof.

**Proof** [of Theorem 8] We prove the lemma by induction on $t$; the base case, for $t = 5$, is proved in Theorem 17 in Appendix A.2. For the induction step, fix any $t \geq 5$ and assume the that the lemma holds for $w_t$; we will prove the claim for $w_{t+1}$.

First, for $\ell_1$, note that, by the hypothesis of the induction, for every $2 \leq k \leq T, w_t{}^{(k)} = c\eta u_0$ for $c \leq \frac{1}{2}$, thus, by Theorem 11, for every $i$, $\nabla \ell_1(w_t, V_i) = 0$.

For $\ell_2$, we know that, for every $i$,

$$\left(\nabla \ell_2(w_t, (V_i, j_i))\right)^{(k)} = \begin{cases} -\phi(V_i, j_i) & k = 0; \\ 0 & \text{otherwise.} \end{cases}$$

For $\ell_3$, using the hypothesis of the induction, which implies that $w_t^{(1)} = c\eta u_0$ for $|c| \leq 1$ and $w_t^{(0)} = \frac{\eta}{n} \sum_{i=1}^{n} \phi(V_i, j_i)$, by Theorem 12, we get that,

$$\left(\nabla \ell_3(w_t)\right)^{(k)} = \begin{cases} \frac{1}{n} \sum_{i=1}^{n} \phi(V_i, j_i) & k = 0; \\ -\frac{\epsilon}{4T^2} u_0 & k = 1; \\ 0 & \text{otherwise.} \end{cases}$$

For $\ell_4$, again by the inductive hypothesis, we know that $w_t^{(1)} = \left(-\frac{3}{8} + \frac{(t-2)}{4} \frac{\epsilon}{T^2}\right)\eta u_0 = c\eta u_0$ for $-\frac{3}{8} \leq c \leq 0$. Then the conditions of Theorem 13 hold for $m = t - 2$, thus, it holds that,

$$\left(\nabla \ell_4(w_t)\right)^{(k)} = \begin{cases} \frac{3}{8} u_0 & k = t - 2; \\ -\frac{1}{2} u_0 & k = t - 1; \\ 0 & \text{otherwise.} \end{cases}$$

Combining all together, we get that,

$$\left(\nabla \widehat{F}(w_t)\right)^{(k)} = \begin{cases} -\frac{\epsilon}{4T^2} u_0 & k = 1; \\ \frac{3}{8} u_0 & k = t - 2; \\ -\frac{1}{2} u_0 & k = t - 1; \\ 0 & \text{otherwise,} \end{cases}$$

where $u_0 \in U \setminus \bigcup_{i=1}^{n} V_i$, which gives the desired for $w_{t+1} = w_t - \eta \nabla \widehat{F}(w_t)$ by direct computation. Lastly, we have $\|w_t\| \leq 1$ by Theorem 18, and the proof is complete. ∎

## A.2. GD Dynamics: The first iterates

**Lemma 14** *Under the conditions of Theorem 5, if $\mathcal{E}$ occurs and $w_t$ is the iterate of Unprojected GD on $\widehat{F}$, with step size $\eta \leq \frac{1}{\sqrt{T}}$ and $w_1 = 0$, then, for $t = 2$ it holds that,*

$$w_2^{(k)} = \begin{cases} \frac{\eta}{n} \sum_{i=1}^{n} \phi(V_i, j_i) & k = 0 \\ 0 & k > 0. \end{cases}$$

**Proof** For $t = 1$, $w_1 = 0$. By Theorem 11 we know that for every $i$, $\nabla \ell_1(w_1, V_i) = 0$. Moreover, by the fact that $\delta_1, \delta_2 > 0$ the maximum in $\ell_3$ and $\ell_4$ is attained in $\delta_1$ and $\delta_2$, respectively, thus we get that

$$\nabla \ell_3(w_1) = \nabla \ell_4(w_1) = 0$$

As a result,

$$\nabla \widehat{F}(w_1)^{(k)} = \frac{1}{n} \sum_{i=1}^{n} \nabla \ell_2(w_1, (V_i, j_i))^{(k)} = \begin{cases} -\frac{1}{n} \sum_{i=1}^{n} (V_i, j_i) & k = 0 \\ 0 & \text{otherwise,} \end{cases}$$

hence,

$$w_2^{(k)} = \begin{cases} \frac{\eta}{n} \sum_{i=1}^n \phi(V_i, j_i) & k = 0 \\ 0 & \text{otherwise.} \end{cases}$$

∎

**Lemma 15** *Under the conditions of Theorem 5, if $\mathcal{E}$ occurs and $w_t$ is the iterate of Unprojected GD on $\widehat{F}$, with step size $\eta \le \frac{1}{\sqrt{T}}$ and $w_1 = 0$, then, for $t = 3$ it holds that,*

$$w_3^{(k)} = \begin{cases} \frac{\eta}{n} \sum_{i=1}^n \phi(V_i, j_i) & k = 0 \\ \frac{\eta}{4} \frac{\epsilon}{T^2} u_0 & k = 1 \\ 0 & 2 \le k \le T, \end{cases}$$

*where $u_0 \in U \setminus \bigcup_{i=1}^n V_i$.*

**Proof** By Theorem 14, $w_2^{(1)}, ..., w_2^{(T)} = 0$, thus, by Theorem 11, we know that for every $i$, $\nabla \ell_1(w_1, V_i) = 0$. Moreover, by the fact that $\delta_2 > 0$, we get that $\nabla \ell_4(w_2) = 0$. For $\ell_3(w_2)$, by Theorem 12, using the fact that $w_2^{(1)} = 0$ and $w_2^{(0)} = \frac{\eta}{n} \sum_{i=1}^n \phi(V_i, j_i)$, we get that

$$\nabla \ell_3(w_2)^{(k)} = \begin{cases} \frac{1}{n} \sum_{i=1}^n \phi(V_i, j_i) & k = 0 \\ -\frac{1}{4} \frac{\epsilon}{T^2} u_0 & k = 1 \\ 0 & \text{otherwise.} \end{cases}$$

For $\ell_2(w_2)$, for every $i$, the gradient is

$$\nabla \ell_2(w_2, (V_i, j_i))^{(k)} = \begin{cases} -\phi(V_i, j_i) & k = 0 \\ 0 & \text{otherwise.} \end{cases}$$

(Note that this implies in particular, $\frac{1}{n} \sum_{i=1}^n \nabla \ell_2(w_2, (V_i, j_i))^{(0)} = -\nabla \ell_3(w_2)^{(0)}$.) Combining all together, we conclude that, for $u_0 \in U \setminus \bigcup_{i=1}^n V_i$, it holds that,

$$\nabla \widehat{F}(w_2)^{(k)} = \begin{cases} 0 & k = 0 \\ -\frac{1}{4} \frac{\epsilon}{T^2} u_0 & k = 1 \\ 0 & 2 \le k \le T, \end{cases}$$

and the result follows from $w_3 = w_2 - \eta \nabla \widehat{F}(w_2)$. ∎

**Lemma 16** *Under the conditions of Theorem 5, if $\mathcal{E}$ occurs and $w_t$ is the iterate Unprojected GD on $\widehat{F}$, with step size $\eta \le \frac{1}{\sqrt{T}}$ and $w_1 = 0$, then, for $t = 4$ it holds that,*

$$w_4^{(k)} = \begin{cases} \frac{\eta}{n} \sum_{i=1}^n \phi(V_i, j_i) & k = 0 \\ -\frac{3\eta}{8} u_0 + \frac{\eta}{2} \frac{\epsilon}{T^2} u_0 & k = 1 \\ \frac{\eta}{2} u_0 & k = 2 \\ 0 & 3 \le k \le T, \end{cases}$$

*where $u_0 \in U \setminus \bigcup_{i=1}^n V_i$.*

**Proof** We start with $\ell_1, \ell_2, \ell_3$. For $\ell_1$, by Theorem 15, for every $2 \leq k \leq T$, $w_3^{(k)} = 0$, thus, by Theorem 11, we know that for every $i$, $\nabla \ell_1(w_1, V_i) = 0$. For $\ell_2$, we know that, for every $i$,

$$\nabla \ell_2(w_3, (V_i, j_i))^{(k)} = \begin{cases} -\phi(V_i, j_i) & k = 0 \\ 0 & \text{otherwise.} \end{cases}$$

For $\ell_3$, by Theorem 12, using the fact that $w_3^{(1)} = c\eta u_0$ for $|c| \leq 1$ and $w_3^{(0)} = \frac{\eta}{n} \sum_{i=1}^n \phi(V_i, j_i)$, we get that,

$$\nabla \ell_3(w_3)^{(k)} = \begin{cases} \frac{1}{n} \sum_{i=1}^n \phi(V_i, j_i) & k = 0 \\ -\frac{1}{4} \frac{\epsilon}{T^2} u_0 & k = 1 \\ 0 & \text{otherwise.} \end{cases}$$

Now, for $\ell_4$, we show that the maximum is attained uniquely in $k = 1$ and $u = u_0$: For $k \neq 1$, for every $u \in U$

$$\frac{3}{8} \langle u, w_3^{(k)} \rangle - \frac{1}{2} \langle u, w_3^{(k+1)} \rangle = 0.$$

For $k = 1$ and $u \neq u_0$,

$$\begin{aligned} \frac{3}{8} \langle u, w_3^{(k)} \rangle - \frac{1}{2} \langle u, w_3^{(k+1)} \rangle &= \frac{3}{8} \langle u, w_3^{(1)} \rangle - \frac{1}{2} \langle u, w_3^{(2)} \rangle \\ &= \frac{3}{8} \langle u, \frac{\eta}{4} \frac{\epsilon}{T^2} u_0 \rangle \\ &\leq \frac{3\eta}{256} \frac{\epsilon}{T^2} \\ &< \delta_2. \end{aligned} \tag{17}$$

For $k = 1$ and $u = u_0$,

$$\begin{aligned} \frac{3}{8} \langle u, w_3^{(k)} \rangle - \frac{1}{2} \langle u, w_3^{(k+1)} \rangle &= \frac{3}{8} \langle u_0, w_3^{(1)} \rangle - \frac{1}{2} \langle u_0, w_3^{(2)} \rangle \\ &= \frac{3}{8} \langle u_0, \frac{\eta}{4} \frac{\epsilon}{T^2} u_0 \rangle \\ &= \frac{3\eta}{32} \frac{\epsilon}{T^2} \\ &> \delta_2. \end{aligned} \tag{18}$$

We derive that,

$$\nabla \ell_4(w_3)^{(k)} = \begin{cases} \frac{3}{8} u_0 & k = 1 \\ -\frac{1}{2} u_0 & k = 2 \\ 0 & 3 \leq k \leq T \\ 0 & k = 0. \end{cases}$$

Combining all together, we get that,

$$\nabla \widehat{F}(w_3)^{(k)} = \begin{cases} \frac{3}{8} u_0 - \frac{1}{4} \frac{\epsilon}{T^2} u_0 & k = 1 \\ -\frac{1}{2} u_0 & k = 2 \\ 0 & 3 \leq k \leq T \\ 0 & k = 0, \end{cases}$$

and

$$w_4^{(k)} = \begin{cases} \frac{\eta}{n} \sum_{i=1}^n \phi(V_i, j_i) & k = 0 \\ -\frac{3\eta}{8} u_0 + \frac{\eta}{2} \frac{\epsilon}{T^2} u_0 & k = 1 \\ \frac{\eta}{2} u_0 & k = 2 \\ 0 & 3 \leq s \leq T, \end{cases}$$

where $u_0 \in U \setminus \bigcup_{i=1}^n V_i$. ∎

**Lemma 17** *Under the conditions of Theorem 5, if $\mathcal{E}$ occurs and $w_t$ is the iterate Unprojected GD on $\widehat{F}$, with step size $\eta \leq \frac{1}{\sqrt{T}}$ and $w_1 = 0$, then, for $t = 5$ it holds that,*

$$w_5^{(k)} = \begin{cases} \frac{1}{n} \sum_{i=1}^n \phi(V_i, j_i) & k = 0, \\ -\frac{3}{8} \eta u_0 + \frac{3\eta}{4} \frac{\epsilon}{T^2} u_0 & k = 1 \\ \frac{1}{8} \eta u_0 & k = 2 \\ \frac{1}{2} \eta u_0 & k = 3 \\ 0 & 4 \leq s \leq T \end{cases}$$

*where $u_0 \in U \setminus \bigcup_{i=1}^n V_i$.*

**Proof** We begin with $\ell_1, \ell_2, \ell_3$. Note that, by Theorem 16, for every $2 \leq k \leq T$, $w_4^{(k)} = c\eta u_0$ for $c \leq \frac{1}{2}$, thus, by Theorem 11, for every $i$, $\nabla \ell_1(w_4, V_i) = 0$. For $\ell_2$, we know that, for every $i$,

$$\nabla \ell_2(w_4, (V_i, j_i))^{(k)} = \begin{cases} -\phi(V_i, j_i) & k = 0 \\ 0 & \text{otherwise.} \end{cases}$$

For $\ell_3$, by Theorem 12, using Theorem 16, where we showed that $w_4^{(1)} = c\eta u_0$ for $|c| \leq 1$ and $w_4^{(0)} = \frac{\eta}{n} \sum_{i=1}^n \phi(V_i, j_i)$, we get that,

$$\nabla \ell_3(w_4)^{(k)} = \begin{cases} \frac{1}{n} \sum_{i=1}^n \phi(V_i, j_i) & k = 0 \\ -\frac{1}{4} \frac{\epsilon}{T^2} u_0 & k = 1 \\ 0 & \text{otherwise.} \end{cases}$$

It is left to calculate $\nabla \ell_4(w_4)$. We show that the maximum is attained uniquely at $k = 2$ and $u = u_0$. First,

$$\frac{3}{8} \langle u, \frac{\eta}{2} \frac{\epsilon}{T^2} u_0 \rangle = \frac{3\eta}{16} \frac{\epsilon}{T^2} \langle u, u_0 \rangle \leq \frac{3\eta}{16T^2},$$

thus, since $T \geq 4$,

$$\frac{3}{8} \langle u, w_4^{(1)} \rangle - \frac{1}{2} \langle u, w_4^{(2)} \rangle = \frac{3}{8} \langle u, -\frac{3\eta}{8} u_0 + \frac{\eta}{2} \frac{\epsilon}{T^2} u_0 \rangle - \frac{1}{2} \langle u, \frac{\eta}{2} u_0 \rangle$$
$$\leq \frac{9\eta}{512} + \frac{\eta}{32} + \frac{9\eta}{256} = \frac{43\eta}{512}$$

$$< \frac{3\eta}{16}.$$

For $k = 2$ and $u = u_0$,

$$\frac{3}{8}\langle u, w_4^{(2)}\rangle - \frac{1}{2}\langle u, w_4^{(3)}\rangle = \frac{3}{8}\langle u_0, \frac{\eta}{2}u_0\rangle - \frac{1}{2}\langle u_0, 0\rangle = \frac{3\eta}{16}(> \delta_2).$$

For $k = 2$ and $u \neq u_{t-2}$,

$$\frac{3}{8}\langle u, w_4^{(2)}\rangle - \frac{1}{2}\langle u', w_3^{(3)}\rangle = \frac{3}{8}\langle u, \frac{\eta}{2}u_0\rangle - \frac{1}{2}\langle u, 0\rangle \leq \frac{3\eta}{128}.$$

For every $3 \leq k \leq T - 1$,

$$\frac{3}{8}\langle u, w_4^{(k)}\rangle - \frac{1}{2}\langle u', w_4^{(k+1)}\rangle = 0.$$

We derive that,

$$\nabla\ell_4(w_4)^{(k)} = \begin{cases} \frac{3}{8}u_0 & k = 2 \\ -\frac{1}{2}u_0 & k = 3 \\ 0 & 3 \leq k \leq T \\ 0 & k = 0. \end{cases}$$

Combining all together, we get that,

$$\nabla\widehat{F}(w_4)^{(k)} = \begin{cases} -\frac{1}{4}\frac{\epsilon}{T^2}u_0 & k = 1 \\ \frac{3}{8}u_0 & k = 2 \\ -\frac{1}{2}u_0 & k = 3 \\ 0 & 4 \leq k \leq T \\ 0 & k = 0 \end{cases}$$

and

$$w_5^{(k)} = \begin{cases} -\frac{3}{8}\eta u_0 + \frac{3\eta}{4}\frac{\epsilon}{T^2}u_0 & k = 1 \\ \frac{1}{8}\eta u_0 & k = 2 \\ \frac{1}{2}\eta u_0 & k = 3 \\ 0 & 4 \leq s \leq T \\ \frac{1}{n}\sum_{i=1}^{n}\phi(V_i, j_i) & k = 0, \end{cases}$$

where $u_0 \in U \setminus \bigcup_{i=1}^{n} V_i$.

∎

**Lemma 18** *Assume the conditions of Theorem 5, and consider the iterates of unprojected GD on $\widehat{F}$, with step size $\eta \leq 1/\sqrt{T}$ initialized at $w_1 = 0$. Under the event $\mathcal{E}$, we have for all $t \in [T]$ that*

$$\|w_t\| \leq 1.$$

**Proof** If $\mathcal{E}$ holds, by Theorems 8 and 14 to 16, we know that for every $t \geq 2$, $\|w_t{}^{(1)}\| \leq \frac{\eta}{2}$, $\|w_t{}^{(t-1)}\| \leq \frac{\eta}{2}$ and for every $k \in \{2, \ldots, t-2\}$, $\|w_t{}^{(t-1)}\| \leq \frac{\eta}{8}$. As a result,

$$
\begin{aligned}
\|w_t\|^2 &\leq \sum_{i=1}^{d} w_t[i]^2 \\
&\leq \sum_{k=0}^{T} \|w_t{}^{(k)}\|^2 \\
&< 2 \cdot \left(\frac{\eta}{2}\right)^2 + (T-3)\left(\frac{\eta}{8}\right)^2 + \left\|\frac{\eta}{n}\sum_{i=1}^{n}\phi(V_i, j_i)\right\|^2 \\
&\leq \frac{\eta^2}{2} + \frac{\eta^2(T-3)}{64} + \eta^2 \\
&\leq \frac{1}{64} + \frac{3}{2T} \qquad\qquad\qquad\qquad (\eta \leq \tfrac{1}{\sqrt{T}}) \\
&\leq 1 \qquad\qquad\qquad\qquad\qquad\quad (T \geq 2)
\end{aligned}
$$

$\blacksquare$

### A.3. Full proof of Theorem 5

**Proof** [of Theorem 5] By Theorem 7, with probability of at least $\frac{1}{6}$, $\mathcal{E}$ occurs and by Theorem 8, it holds for every $2 \leq k \leq T-3$ that,

$$
\begin{aligned}
w_{T,m}{}^{(k)} = \frac{1}{m}\sum_{i=1}^{m} w_{T-i+1}{}^{(k)} &= \begin{cases} \frac{\eta}{8}u_0 & k \leq T-m-2 \\ \frac{1}{m}\left(\frac{\eta}{2} + \frac{\eta}{8}(T-k-2)\right)u_0 & k \geq T-m-1 \end{cases} \qquad (19) \\
&= \begin{cases} \frac{\eta}{8}u_0 & k \leq T-m-2 \\ \frac{\eta(T-k+2)}{8m}u_0 & k \geq T-m-1 \end{cases}
\end{aligned}
$$

Then, we denote $\alpha_V \in \mathbb{R}^{T-4}$ the vector which its $k$th entry is $\max\left(\frac{3\eta}{32}, \max_{u \in V}\langle u, w_{T,m}{}^{(k+1)}\rangle\right)$. By the fact that every vector $u \in U$ is in $V$ with probability $\frac{1}{2}$, the following holds,

$$
\begin{aligned}
\mathbb{E}_V\sqrt{\sum_{k=2}^{T}\max\left(\frac{3\eta}{32}, \max_{u\in V}\langle u, w_{T,m}{}^{(k)}\rangle\right)^2} &\geq \mathbb{E}_V\sqrt{\sum_{k=2}^{T-3}\max\left(\frac{3\eta}{32}, \max_{u\in V}\langle u, w_{T,m}{}^{(k)}\rangle\right)^2} \\
&= \mathbb{E}_V\sqrt{\sum_{k=1}^{T-4}\max\left(\frac{3\eta}{32}, \max_{u\in V}\langle u, w_{T,m}{}^{(k+1)}\rangle\right)^2} \\
&= \mathbb{E}_V\|\alpha_V\| \\
&\geq \|\mathbb{E}_V\alpha_V\| \\
&= \sqrt{\sum_{k=2}^{T-3}\left(\mathbb{E}_V\max\left(\frac{3\eta}{32}, \max_{u\in V}\langle u, w_{T,m}{}^{(k)}\rangle\right)\right)^2}
\end{aligned}
$$

Then, by Eq. (19),

$$\mathbb{E}_V \sqrt{\sum_{k=2}^{T} \max\left(\frac{3\eta}{32}, \max_{u\in V}\langle u, w_{T,m}^{(k)}\rangle\right)^2}$$

$$\geq \sqrt{\sum_{k=2}^{T-m-2}\left(\mathbb{E}_V \max\left(\frac{3\eta}{32}, \max_{u\in V}\langle u, w_{T,m}^{(k)}\rangle\right)\right)^2 + \sum_{k=T-m-1}^{T-3}\left(\mathbb{E}_V \max\left(\frac{3\eta}{32}, \max_{u\in V}\langle u, w_{T,m}^{(k)}\rangle\right)\right)^2}$$

$$\geq \sqrt{\sum_{k=2}^{T-m-2}\left(\mathbb{E}_V \max\left(\frac{3\eta}{32}, \max_{u\in V}\langle u, \frac{\eta}{8}u_0\rangle\right)\right)^2 + \sum_{k=T-m-1}^{T-3}\left(\mathbb{E}_V \max\left(\frac{3\eta}{32}, \max_{u\in V}\langle u, \frac{\eta(T-k+2)}{8m}u_0\rangle\right)\right)^2}$$

$$= \frac{\eta}{8}\sqrt{\sum_{k=2}^{T-m-2}\left(\mathbb{E}_V \max\left(\frac{3}{4}, \max_{u\in V}\langle u, u_0\rangle\right)\right)^2 + \sum_{k=T-m-1}^{T-3}\left(\mathbb{E}_V \max\left(\frac{3}{4}, \frac{T-k+2}{m}\max_{u\in V}\langle u, u_0\rangle\right)\right)^2}$$

$$\geq \frac{\eta}{8}\sqrt{\sum_{k=2}^{T-m-2}\left(\mathbb{E}_V \max\left(\frac{3}{4}, \max_{u\in V}\langle u, u_0\rangle\right)\right)^2 + \sum_{k=T-m-1}^{T-3}\left(\mathbb{E}_V \max\left(\frac{3}{4}, \frac{T-k+2}{T}\max_{u\in V}\langle u, u_0\rangle\right)\right)^2}$$

$$= \frac{\eta}{8}\sqrt{\sum_{k=2}^{T-m-2}\left(\mathbb{E}_V \max\left(\frac{3}{4}, \max_{u\in V}\langle u, u_0\rangle\right)\right)^2 + \sum_{k=1}^{m-1}\left(\mathbb{E}_V \max\left(\frac{3}{4}, \frac{k+4}{T}\max_{u\in V}\langle u, u_0\rangle\right)\right)^2}$$

Now, treating each of the term separately, with probability $\frac{1}{2}$ on $V$, $\max_{u\in V}\langle u, u_0\rangle \leq \frac{1}{8}$ (otherwise it is 1), thus,

$$\mathbb{E}_V \max\left(\frac{3}{4}, \max_{u\in V}\langle u, u_0\rangle\right) = \frac{1}{2}\cdot\frac{3}{4} + \frac{1}{2}\cdot 1 = \frac{7}{8}$$

Moreover, if $k \leq \frac{3T}{4} - 4$

$$\mathbb{E}_V \max\left(\frac{3}{4}, \frac{k+4}{T}\max_{u\in V}\langle u, u_0\rangle\right) = \frac{3}{4},$$

otherwise,

$$\mathbb{E}_V \max\left(\frac{3}{4}, \frac{k+4}{T}\max_{u\in V}\langle u, u_0\rangle\right) \geq \frac{1}{2}\max\left(\frac{3}{4}, \frac{k+4}{T}\right) + \frac{1}{2}\cdot\frac{3}{4}$$

$$\geq \frac{3}{8} + \frac{k+4}{2T}$$

Then, we get, if $m \geq T - 3$, (note that it implies $l - 1 \geq \frac{3T}{4} - 4$),

$$\mathbb{E}_V \sqrt{\sum_{k=2}^{T}\max\left(\frac{3\eta}{32}, \max_{u\in V}\langle u, w_{T,m}^{(k)}\rangle\right)^2}$$

$$\geq \frac{\eta}{8}\sqrt{\sum_{k=1}^{m-1}\left(\mathbb{E}_V \max\left(\frac{3}{4}, \frac{k+4}{T}\max_{u\in V}\langle u, u_0\rangle\right)\right)^2}$$

$$\geq \frac{\eta}{8}\sqrt{\sum_{k:1\leq k\leq \frac{3T}{4}-4}\frac{9}{16} + \sum_{k:\frac{3T}{4}-4<k\leq m-1}\left(\frac{3}{8}+\frac{k+4}{2T}\right)^2}$$

$$\geq \frac{\eta}{8}\sqrt{\sum_{k:1\leq k\leq \frac{3T}{4}-4}\frac{9}{16} + \sum_{k:\frac{3T}{4}<k\leq T}\left(\frac{3}{8}+\frac{k}{2T}\right)^2}$$

$$\geq \frac{\eta}{8}\sqrt{\frac{27T-144}{64} + \sum_{k:\frac{3T}{4}<k\leq T}\left(\frac{9}{64}+\frac{3k}{8T}\right)}$$

$$\geq \frac{\eta}{8}\sqrt{\frac{27T-144}{64} + \left(\frac{9T}{256}+\frac{3}{8T}\sum_{k:\frac{3T}{4}<k\leq T}k\right)}$$

$$\geq \frac{\eta}{8}\sqrt{\frac{27T-144}{64} + \left(\frac{9T}{256}+\frac{3T^2}{16T}-\frac{3(\frac{3T}{4}+1)^2}{16T}\right)} \qquad (\tfrac{i^2}{2}\leq \sum_{i=1}^n i^2 \leq \tfrac{(i+1)^2}{2})$$

$$= \frac{\eta}{8}\sqrt{\frac{27T-144}{64} + \left(\frac{9T}{256}+\frac{3T}{16}-\frac{27T}{256}-\frac{3}{16T}-\frac{9}{32}\right)}$$

$$= \frac{\eta}{8}\sqrt{\frac{148T}{256}-\frac{45}{32}-\frac{3}{16T}}$$

$$\geq \frac{\eta}{8}\sqrt{\frac{147T}{256}} \qquad (T\geq 512 \implies \tfrac{45}{32}+\tfrac{3}{16T}\leq \tfrac{T}{256})$$

$$\geq \frac{3\eta}{32}\cdot \frac{101\sqrt{T}}{100}.$$

Otherwise, if $m < T - 4$, by similar arguments,

$$\mathbb{E}_V\sqrt{\sum_{k=2}^T \max\left(\frac{3\eta}{32},\max_{u\in V}\langle u,w_{T,m}{}^{(k)}\rangle\right)^2}$$

$$\geq \frac{\eta}{8}\sqrt{\sum_{k=2}^{T-m-2}\left(\mathbb{E}_V\max\left(\frac{3}{4},\max_{u\in V}\langle u,u_0\rangle\right)\right)^2 + \sum_{k=T-m-1}^{T-3}\left(\mathbb{E}_V\max\left(\frac{3}{4},\frac{T-k+2}{T}\max_{u\in V}\langle u,u_0\rangle\right)\right)^2}$$

$$\geq \frac{\eta}{8}\sqrt{\sum_{k=2}^{T-m-2}\left(\frac{7}{8}\right)^2 + \sum_{k:1\leq k\leq \frac{3T}{4}-4}\frac{9}{16} + \sum_{k:\frac{T}{2}<k\leq m+3}\left(\frac{3}{8}+\frac{k}{2T}\right)^2}$$

$$= \frac{\eta}{8}\sqrt{\sum_{k=m+4}^{T}\left(\frac{7}{8}\right)^2 + \sum_{k:1\leq k\leq \frac{3T}{4}-4}\frac{9}{16} + \sum_{k:\frac{T}{2}<k\leq m+3}\left(\frac{3}{8}+\frac{k}{2T}\right)^2}$$

$$\geq \frac{\eta}{8}\sqrt{\sum_{k:1\leq k\leq \frac{3T}{4}-4}\frac{9}{16} + \sum_{k:\frac{3T}{4}<k\leq T}\left(\frac{3}{8}+\frac{k}{2T}\right)^2} \qquad (\tfrac{3}{8}+\tfrac{k}{2T}\leq \tfrac{7}{8})$$

$$\geq \frac{3\eta}{32} \cdot \frac{101\sqrt{T}}{100}.$$

Moreover, we notice that for every $t$, $\ell_2(w_t) \geq -\|w_t^{(0)}\| \geq -\eta$, $\ell_3(w_t) \geq \delta_1$ and $\ell_4(w_t) \geq \delta_2$, thus, it holds that,

$$F(w_{T,l}) \geq \frac{303\eta}{3200}\sqrt{T} + \delta_1 + \delta_2 - \eta \geq \eta\left(\frac{303}{3200}\sqrt{T} - 1\right)$$

and

$$F(w_*) \leq \frac{3\eta}{32}\sqrt{T} + \eta$$

Then, with probability of at least $\frac{1}{6}$,

$$\begin{aligned}
F(w_{T,l}) - F(w_*) &\geq \eta(\frac{303}{3200}\sqrt{T} - 2 - \frac{3}{32}\sqrt{T}) \\
&\geq \eta(\frac{303}{3200}\sqrt{T} - \frac{302}{3200}\sqrt{T}) \qquad\qquad (T \geq 3200^2 \implies 2 \leq \frac{2}{3200}\sqrt{T}) \\
&= \frac{\eta}{3200}\sqrt{T}.
\end{aligned}$$

∎

## A.4. Deferred Proofs

**Proof** [of Theorem 3] Let $r = 2^{\frac{d'}{178}}$. For every $1 \leq i \leq r$ and $1 \leq j \leq d'$ we define the random variable $u_i^j$ to be $\frac{1}{\sqrt{d'}}$ with probability $\frac{1}{2}$ and $-\frac{1}{\sqrt{d'}}$ with probability $\frac{1}{2}$. Then, for every $1 \leq i \leq r$, we define the vector $u_i = (u_i^1, \cdots, u_i^{d'})$ and look at the set $U = \{u_1, u_2, \ldots u_r\}$. This set will satisfy the required property with positive probability. First, for every $i \neq k$, $\langle u_i, u_k \rangle$ are sums of $d$ random variables that take values in $\{-\frac{1}{d'}, \frac{1}{d'}\}$ with $\mathbb{E}\langle u_i, u_k \rangle = 0$. Then by Hoeffding's inequality,

$$Pr(|\langle u_i, u_k \rangle| \geq \frac{1}{8}) \leq 2e^{\frac{-2\left(\frac{1}{8}\right)^2}{d' \cdot \frac{4}{d'^2}}} = 2e^{-\frac{d'}{128}}$$

Then, by union bound on the $\binom{r}{2}$ pairs of vectors in $U$,

$$Pr(\exists i, k \ |\langle u_i, u_k \rangle| \geq \frac{1}{8}) \leq 2e^{-\frac{d'}{128}} \cdot \binom{r}{2} < 2e^{-\frac{d'}{128}} \cdot \frac{1}{2}r^2 \leq 1.$$

∎

**Proof** [of Theorem 4] First, we consider an arbitrary enumeration of $P(U) = \{V^1, \ldots V^N\}$, with $N := |P(U)|$, and define $g : P(U) \to \mathbb{R}^2$, $g(V^i) = \left(\sin\left(\frac{\pi i}{N}\right), \cos\left(\frac{\pi i}{N}\right)\right)$. Now, we refer to a vector $a \in \mathbb{R}^{2n^2}$ as a concatenation of $n^2$ vectors in $\mathbb{R}^2$, $a^{(1)}, \ldots, a^{(n^2)}$. Then, we define $\delta = 1 - \cos\left(\frac{\pi}{2N}\right)$, $\epsilon = \frac{\delta}{n^2}$ and

$$\phi(V, j)^{(j')} = \begin{cases} g(V) & j' = j \\ 0 & \text{otherwise} \end{cases}$$

As a result, for every $V^i, j$ it holds that

$$\|\phi(V^i, j)\| = \|g(V^i)\| = \sqrt{\sin\left(\frac{\pi i}{2N}\right)^2 + \cos\left(\frac{\pi i}{2N}\right)^2} = 1$$

Moreover, if $j_1 \neq j_2$,

$$\langle \phi(V^i, j_1), \phi(V^i, j_2) \rangle = 0,$$

and if $i > k$,

$$
\begin{aligned}
\langle \phi(V^i, j), \phi(V^k, j) \rangle &= \langle g(V^i), g(V^k) \rangle \\
&= \sin\left(\frac{\pi i}{2N}\right)\sin\left(\frac{\pi k}{2N}\right) + \cos\left(\frac{\pi i}{2N}\right)\cos\left(\frac{\pi k}{2N}\right) \\
&= \cos\left(\frac{\pi(i-k)}{2N}\right) \\
&\leq \cos\left(\frac{\pi}{2N}\right) \qquad \text{(cos is monotonic decreasing in } [0, \pi/2]) \\
&= 1 - \delta
\end{aligned}
$$

We notice that $0 < \delta < 1$. Now, we consider an arbitrary enumeration of $U = \{v_1, ... v_{|U|}\}$, and define the following set $\Psi \subseteq \mathbb{R}^{2n^2}$ and the following two mappings $\sigma : R^{2n^2} \to P(U), \alpha : R^{2n^2} \to U$,

$$\Psi = \{\frac{1}{n}\sum_{i=1}^{n}\phi(V_i, j_i) : \forall i \ V_i \subseteq U, \ j_i \in [n^2] \text{ and } i \neq l \implies j_i \neq j_l\}$$

Note that, for every $\psi \in \Psi$,

$$\|\psi\| = \|\frac{1}{n}\sum_{i=1}^{n}\phi(V_i, j_i)\| \leq \frac{1}{n}\sum_{i=1}^{n}\|\phi(V_i, j_i)\| = 1.$$

Then, for every $a \in \mathbb{R}^{2n^2}$ and $j \in [n^2]$, we denote the index $q(a, j) \in [|P(U)|]$ as

$$q(a, j) = \arg\max_{r}\langle g(V_r), a^{(j)} \rangle,$$

and define the following mapping $\sigma : \mathbb{R}^{2n^2} \to P(U)$,

$$\sigma(a) = \bigcup_{j=1, a^{(j)} \neq 0}^{n^2} V_{q(a,j)}.$$

Moreover, for every $a \in \mathbb{R}^{2n^2}$, we denote the index $p(a) \in [|U|]$ as

$$p(a) = \begin{cases} -1 & \sigma(a) = U, \\ \arg\min_i\{i : v_i \in U \setminus \sigma(a)\} & \sigma(a) \neq U, \end{cases}$$

and define the following mapping $\alpha : \mathbb{R}^{2n^2} \to U$,

$$\alpha(a) = \begin{cases} v_{|U|} & \sigma(a) = U \\ v_{p(a)} & \sigma(a) \neq U \end{cases} .$$

Now, Let $V_1, \ldots, V_n \subseteq U$ and $j_1, \ldots j_n$ that are sampled uniformly from $[n^2]$, We prove the last part of the lemma under the condition that $j_i \neq j_k$ for $i \neq k$. $\psi^* = \frac{1}{n} \sum_{i=1}^n \phi(V_i, j_i)$ holds

$$\langle \psi^*, \frac{1}{n} \sum_{i=1}^n \phi(V_i, j_i) \rangle = \langle \frac{1}{n} \sum_{i=1}^n \phi(V_i, j_i), \frac{1}{n} \sum_{i=1}^n \phi(V_i, j_i) \rangle$$

$$= \frac{1}{n^2} \sum_{i=1}^n \langle \phi(V_i, j_i), \phi(V_i, j_i) \rangle$$

$$= \frac{1}{n}$$

$$> \frac{7}{8n}$$

Let $\psi \in \Psi, \psi = \frac{1}{n} \sum_{l=1}^n \phi(V_l', j_l')$ such that $\psi \neq \psi^*$, then by the definition of $\Psi$ there are at most $n$ pairs $i, l$ such that $\langle \phi(V_i, j_i), \phi(V_l', j_l') \rangle \neq 0$. Thus, there exists a pair $(V_r', j_r')$ such that $(V_r', j_r') \notin \{(V_i, j_i) : i \in [n]\}$, and for every $i$, $\langle \phi(V_i, j_i), \phi(V_l', j_l') \rangle \leq 1 - \delta$. As a result,

$$\langle \psi, \frac{1}{n} \sum_{l=1}^n \phi(V_i, j_i) \rangle = \langle \frac{1}{n} \sum_{i=1}^n \phi(V_l', j_l'), \frac{1}{n} \sum_{i=1}^n \phi(V_i, j_i) \rangle$$

$$= \frac{1}{n^2} \sum_{i=1}^n \sum_{l=1}^n \langle \phi(V_i, j_i), \phi(V_l', j_l') \rangle$$

$$\leq \frac{1}{n^2} \left( 1 - \delta + \sum_{i=1, i \neq r}^n 1 \right)$$

$$\leq \frac{1}{n^2} (1 - \delta + n - 1)$$

$$= \frac{1}{n} - \frac{\delta}{n^2}$$

$$= \langle \psi^*, \frac{1}{n} \sum_{i=1}^n \phi(V_i, j_i) \rangle - \epsilon$$

Furthermore, since all $j_i$ are distinct, for every $i \in [n]$ it holds that, $\left( \frac{1}{n} \sum_{i'=1}^n \phi(V_{i'}, j_{i'}) \right)^{(j_i)} = \frac{1}{n} g(V_i)$, thus,

$$q \left( \frac{1}{n} \sum_{i'=1}^n \phi(V_{i'}, j_{i'}), j_i \right) = \arg \max_r \left\langle g(V_r), \left( \frac{1}{n} \sum_{i'=1}^n \phi(V_{i'}, j_{i'}) \right)^{(j_i)} \right\rangle$$

$$= \arg \max_r \langle g(V_r), \frac{1}{n} g(V_i) \rangle$$

$$= i,$$

and we get,

$$\sigma(\psi^*) = \sigma\left(\frac{1}{n}\sum_{i=1}^{n}\phi(V_i, j_i)\right)$$

$$= \bigcup_{j=1,\,\frac{1}{n}\sum_{i=1}^{n}\phi(V_l,j_i)^{(j)}\neq 0}^{n^2} V_{q\left(\frac{1}{n}\sum_{i=1}^{n}\phi(V_i,j_i),j\right)}$$

$$= \bigcup_{i=1}^{n} V_{q\left(\frac{1}{n}\sum_{i=1}^{n}\phi(V_i,j_i),j_i\right)} \qquad \text{(The indices that are non-zero are } \{j_i\}_{i=1}^{n}\text{)}$$

$$= \bigcup_{i=1}^{n} V_i$$

Finally, assuming that $\bigcup_{i=1}^{n} V_i \neq U$,

$$\alpha(\psi^*) = v_{p(a)} \in U \setminus \bigcup_{i=1}^{n} V_i.$$

$\blacksquare$

**Proof** [of Theorem 6] We prove that $\ell_1, \ell_2$ and $\ell_4$ are convex and 1-Lipschitz and $\ell_3$ is convex and 2-Lipschitz. First, by Theorems 3 and 4 for every $u \in U$ and $V \in P(U)$, $j \in [n^2]$, it holds that $\|u\| = 1$ and $\|\phi(V, j)\| = 1$. Hence, $\ell_2$ is a 1-Lipschitz linear function, and $\ell_4$ is a maximum over 1-Lipschitz linear functions, thus, also both functions are convex and 1-Lipschitz. Further, every $\psi \in \Psi$ satisfies $\|\psi\| \leq 1$ by Theorem 3 (ii), thus, taking into account that $\beta < 1$, we have $\ell_3$ is a maximum over 2-Lipschitz linear functions and therefore convex and 2-Lipschitz.

For $\ell_1$, for every set $V \subseteq U$, let $\alpha_V(w) \in \mathbb{R}^{T-1}$ to be the vector which its $k$'th coordinate is $\alpha_V(w)^{(k)} = \max\left(\frac{3\eta}{32}, \max_{u \in V}\langle u, w^{(k+1)}\rangle\right)$ and prove convexity and 1-Lipshitzness. For establishing convexity, observe

$$\sqrt{\sum_{k=2}^{T}\max\left(\frac{3\eta}{32}, \max_{u \in V}\langle u, (\lambda x + (1-\lambda)y)^{(k)}\rangle\right)^2}$$

$$= \sqrt{\sum_{k=2}^{T}\max\left(\frac{3\eta}{32}, \max_{u \in V}\left(\lambda\langle u, x^{(k)}\rangle + (1-\lambda)\langle u, y^{(k)}\rangle\right)\right)^2}$$

$$\leq \sqrt{\sum_{k=2}^{T}\max\left(\frac{3\eta}{32}, \max_{u \in V}\left(\lambda\langle u, x^{(k)}\rangle\right) + \max_{u \in V}\left((1-\lambda)\langle u, y^{(k)}\rangle\right)\right)^2}$$

$$\text{(convexity of max \& monotonicity of square root)}$$

$$\leq \sqrt{\sum_{k=2}^{T}\left(\lambda\max\left(\frac{3\eta}{32}, \max_{u \in V}\left(\langle u, x^{(k)}\rangle\right)\right) + (1-\lambda)\max\left(\frac{3\eta}{32}, \max_{u \in V}\langle u, y^{(k)}\rangle\right)\right)^2}$$

$$= \|\lambda\alpha_V(x) + (1-\lambda)\alpha_V(y)\|_2$$

$$\leq \lambda \|\alpha_V(x)\|_2 + (1 - \lambda)\alpha_V(y)\|_2 \qquad\qquad \text{(convexity of } \ell_2 \text{ norm)}$$

$$= \lambda \sqrt{\sum_{k=2}^{T} \max\left(\frac{3\eta}{32}, \max_{u \in V}\langle u, x^{(k)}\rangle\right)^2} + (1 - \lambda)\sqrt{\sum_{k=2}^{T} \max\left(\frac{3\eta}{32}, \max_{u \in V}\langle u, y^{(k)}\rangle\right)^2}.$$

For 1-Lipschitzness, for every $w \in \mathbb{R}^d$ and sub-gradient $g(w, V) \in \partial\ell_1(w, V)$, there exists a sub gradient $g_h(w, V) \in \partial\left(\sum_{k=2}^{T} \max\left(\frac{3\eta}{32}, \max_{u \in V}\langle u, w^{(k)}\rangle\right)^2\right)$ such that

$$\|g(w, V)\| = \frac{\|g_h(w, V)\|}{2\sqrt{\sum_{k=2}^{T} \max\left(\frac{3\eta}{32}, \max_{u \in V}\langle u, w^{(k)}\rangle\right)^2}} = \frac{\|g_h(w, V)\|}{2\sqrt{\sum_{k=2}^{T} \alpha_V(w)^{(k)2}}}.$$

Further, note that $g_h(w, V)$ is of the form

$$g_h(w, V) = 2\sum_{k=2}^{T} r_{k,V}(w)\alpha_V(w)^{(k)},$$

where

$$r_{k,V}(w) \in \partial\left(w \mapsto \alpha_V(w)^{(k)}\right)(w),$$

hence

$$r_{k,V}(w)^{(j)} \in \begin{cases} \{0\} & j \neq k, \\ \text{Conv}\left(\{0\} \cup U\right) & j = k, \end{cases}$$

where $\text{Conv}(S)$ denotes the convex hull of a set $S$. This implies that $\|r_{k,V}(w)\| \leq 1$, and $\langle r_{k,V}(w), r_{k',V}(w)\rangle = 0$ for $k \neq k'$. Therefore,

$$\|g_h(w, V)\| = \left\|2\sum_{k=2}^{T} r_{k,V}(w)\alpha_V(w)^{(k)}\right\|$$

$$= 2\sqrt{\left\|\sum_{k=2}^{T} r_{k,V}(w)\alpha_V(w)^{(k)}\right\|^2}$$

$$= 2\sqrt{\sum_{k=2}^{T} \alpha_V(w)^{(k)2}\|r_{k,V}(w)\|^2}$$

$$\leq 2\sqrt{\sum_{k=2}^{T} \alpha_V(w)^{(k)2}}.$$

This now implies that $\|g(w, V)\| \leq 1$, and completes the proof. $\blacksquare$

**Proof** [of Theorem 7] By the fact that every $V_i$ and $j_i$ are independent, it is enough to show that

$$\Pr\left(\bigcup_{i=1}^{n} V_i \neq U\right) \geq \frac{1}{2},$$

and,

$$\Pr\left(\text{for every } k \neq l,\ j_k \neq j_l\right) \geq \frac{1}{3}.$$

For the former, for every $u \in U$, since $V_i$ are sampled independently and every vector $u \in U$ is in every $V_i$ with probability $\frac{1}{2}$,

$$\Pr\left(u \in \bigcup_{i=1}^{n} V_i\right) = 1 - \Pr\left(u \notin \bigcup_{i=1}^{n} V_i\right) = 1 - 2^{-n},$$

thus, since by Theorem 3, $|U| \geq 2^{\frac{d'}{178}} = 2^n$, it holds that,

$$\Pr\left(\bigcup_{i=1}^{n} V_i = U\right) = \Pr\left(\forall u \in U\ u \in \bigcup_{i=1}^{n} V_i\right)$$
$$= (1 - 2^{-n})^{|U|}$$
$$\leq (1 - 2^{-n})^{2^n}$$
$$\leq \frac{1}{e}$$
$$< \frac{1}{2}.$$

We conclude,

$$\Pr\left(\bigcup_{i=1}^{n} V_i \neq U\right) \geq \frac{1}{2}.$$

For the latter, since all $j_i$s are sampled independently, for a single pair $k \neq l$, it holds that

$$\Pr(j_k \neq j_l) = 1 - \frac{1}{n^2}$$

As a result,

$$\Pr\left(\text{for every } k \neq l,\ j_k \neq j_l\right) = \left(1 - \frac{1}{n^2}\right)^{\frac{n(n-1)}{2}} \geq \left(1 - \frac{1}{n^2}\right)^{\frac{n^2}{2}} \geq \frac{1}{\sqrt{2e}} \geq \frac{1}{e}.$$

∎

**Proof** [of Theorem 9] First, differentiability can be derived immediately from Theorem 31. Second, for 5-Lipschitzness, for every $(V, j) \in Z$, we define $\tilde{f}_{V,j} : \mathbb{R}^d \to \mathbb{R}$ as $\tilde{f}_{V,j}(w) := \tilde{f}(w, (V, j))$. By the 5-Lipschitzness of $f$ with respect to its first argument and Jensen Inequality, for every $x, y \in \mathbb{R}^d$, it holds that

$$|\tilde{f}_{V,j}(x) - \tilde{f}_{V,j}(y)| = \left|\mathbb{E}_{v \in \delta B}\left(f_{V,j}(y + v)\right) - \mathbb{E}_{v \in \delta B}\left(f_{V,j}(w + v)\right)\right|$$
$$= \left|\mathbb{E}_{v \in \delta B}\left(f_{V,j}(x + v) - f_{V,j}(y + v)\right)\right|$$
$$\leq \mathbb{E}_{v \in \delta B}\left|\left(f_{V,j}(x + v) - f_{V,j}(y + v)\right)\right|$$
$$\leq 5|x - y|.$$

Third, for convexity, by the convexity of $f$ for every $x, y \in \mathbb{R}^d$ and $\alpha \in [0, 1]$,

$$
\begin{aligned}
\tilde{f}_{V,j}\left(\alpha x + (1 - \alpha)y\right) &= \mathbb{E}_{v \in \delta B}\left(f_{V,j}(\alpha x + (1 - \alpha)y + v)\right) \\
&= \mathbb{E}_{v \in \delta B}\left(f_{V,j}(\alpha(x + v) + (1 - \alpha)(y + v))\right) \\
&\leq \mathbb{E}_{v \in \delta B}\left(\alpha f_{V,j}(x + v) + (1 - \alpha)f_{V,j}(y + v))\right) \\
&= \alpha \mathbb{E}_{v \in \delta B}\left(f_{V,j}(x + v)\right) + (1 - \alpha)\left(\mathbb{E}_{v \in \delta B} f_{V,j}(y + v)\right) \\
&= \alpha \tilde{f}_{V,j}(x) + (1 - \alpha)\tilde{f}_{V,j}(y).
\end{aligned}
$$

∎

**Proof** [of Theorem 10] We assume that $\mathcal{E}$ (Eq. (13)) holds and show Theorem 10 under this event. We prove the claim by induction on $t$. For $t = 1$, it is trivial. Now, we assume that $w_t = \tilde{w}_t$.

For $\ell_1$, in every $t$, by the proofs of Theorems 14 to 17 and Theorem 8, it can be observed that for every $i \in [n]$, $k \geq 2$, $\max_t \max_{u \in V_i}\langle u, w_t^{(k)}\rangle \leq \frac{\eta}{16}$, thus, in every iteration the term that gets the maximal value is $\frac{3\eta}{32}$. Then, by Theorem 33 and the hypothesis of the induction, for every $i$,

$$
\nabla \tilde{\ell}_1(\tilde{w}_t, V_i) = \nabla \tilde{\ell}_1(w_t, V_i) = 0 = \nabla \ell_1(w_t, V_i).
$$

For $\tilde{\ell}_2$ and every $w \in \mathbb{R}^d$, $V \subseteq U$ and $j \in [n^2]$, by linearity of expectation,

$$
\begin{aligned}
\tilde{\ell}_2(w, (V, j)) &= \mathbb{E}_{v \in \delta B}\left[\langle w^{(0)} + v^{(0)}, -\phi(V, j)\rangle\right] \\
&= \ell_2(w, (V, j)) + \langle \mathbb{E}_{v \in \delta B} v^{(0)}, -\phi(V, j)\rangle \\
&= \ell_2(w, (V, j))
\end{aligned}
$$

Then, we derive that for every $w$ and $i$, $\nabla \tilde{\ell}_2(w, (V_i, j_i)) = \nabla \ell_2(w, (V_i, j_i))$.

For $\tilde{\ell}_3$, when $t = 1$ the term that gets the maximal value is $\delta_1$. Moreover, it can be observed that for every $\psi \in \Psi$,

$$
\max_{\psi \in \Psi}\left(\langle w_1^{(0)}, \psi\rangle - \frac{1}{4}\frac{\epsilon}{T^2}\langle \alpha(\psi), w_1^{(1)}\rangle\right) = 0.
$$

Then, we can apply Theorem 32, and get by the hypothesis of the induction,

$$
\nabla \tilde{\ell}_3(\tilde{w}_1) = \nabla \tilde{\ell}_3(w_1) = 0 = \nabla \ell_3(w_1).
$$

If $t \geq 2$, by Theorems 15 to 17 and Theorem 8, we have,

$$
\begin{aligned}
\ell_3(w_t) &= \max\left(\delta_2, \max_{\psi \in \Psi}\left(\langle w_t^{(0)}, \psi\rangle - \frac{1}{4}\frac{\epsilon}{T^2}\langle \alpha(\psi), w_t^{(1)}\rangle\right)\right) \\
&= \max_{\psi \in \Psi}\left(\langle w_t^{(0)}, \psi\rangle - \frac{1}{4}\frac{\epsilon}{T^2}\langle \alpha(\psi), w_t^{(1)}\rangle\right)
\end{aligned}
$$

and further by Theorem 12 (i), the maximal value of $\langle w^{(0)}, \psi\rangle - \frac{1}{4}\frac{\epsilon}{T^2}\langle \alpha(\psi), w^{(1)}\rangle$ is attained at $\psi = \psi^*$, and the difference from the second maximal possible value of this term is at least $\frac{\eta\epsilon}{4}$. In addition, by Theorem 12 (ii) we have that this maximum is also larger than $\delta_1$ by at least $\frac{\eta}{16n}$. Hence, since $\delta = \frac{\eta\beta}{32} = \frac{\eta\epsilon}{128T^2} < \frac{1}{2}\frac{\eta\epsilon}{4}$ we can apply Theorem 34 and get by the hypothesis of the induction that,

$$
\nabla \tilde{\ell}_3(\tilde{w}_t)^{(k)} = \nabla \tilde{\ell}_3(w_t)^{(k)}
$$

$$= \begin{cases} \frac{1}{n} \sum_{i=1}^{n} \phi(V_i, j_i) & k = 0 \\ -\frac{\epsilon}{4T^2} \alpha(\frac{1}{n} \sum_{i=1}^{n} \phi(V_i, j_i)) & k = 1 \\ 0 & \text{otherwise} \end{cases}$$

$$= \nabla \ell_3(w_t)^{(k)}.$$

For $\tilde{\ell}_4$, for $t \in \{1, 2\}$, $w_t^{(k)} = 0$ for $k > 0$ by Theorem 14, hence the term that gets the maximal value is $\delta_2$. Moreover, it can be observed that for every such $t$, and every $k \in [T-1]$ and $u \in U$,

$$\frac{3}{8} \langle u, w_t^{(k)} \rangle - \frac{1}{2} \langle u, w_t^{(k+1)} \rangle = 0.$$

Then, we can apply Theorem 32, and get by the hypothesis of the induction,

$$\nabla \tilde{\ell}_4(\tilde{w}_t) = \nabla \tilde{\ell}_4(w_t) = 0 = \nabla \ell_4(w_t).$$

For $t = 3$, it can be observed by Theorem 15 that,

$$\begin{aligned} \ell_4(w_t) &= \max \left( \delta_2, \max_{k \in [T-1], u \in U} \left( \frac{3}{8} \langle u, w_t^{(k)} \rangle - \frac{1}{2} \langle u, w_t^{(k+1)} \rangle \right) \right) \\ &= \max_{k \in [T-1], u \in U} \left( \frac{3}{8} \langle u, w_t^{(k)} \rangle - \frac{1}{2} \langle u, w_t^{(k+1)} \rangle \right) \\ &= \frac{3}{8} \langle u, w_t^{(1)} \rangle \\ &= \frac{3}{8} \frac{\eta \epsilon}{4T^2} \langle u_0, u_0 \rangle \\ &= \frac{3\eta \epsilon}{32T^2}. \end{aligned}$$

The second maximal possible value of this term is $\delta_2 = \frac{3\eta \epsilon}{64T^2}$ (by Eqs. (17) and (18)), then, by the fact that $\delta < \frac{3\eta \epsilon}{32T^2} - \frac{3\eta \epsilon}{64T^2} = \frac{3\eta \epsilon}{64T^2}$, we can again apply Theorem 34 and get by the hypothesis of the induction that

$$\begin{aligned} \nabla \tilde{\ell}_4(\tilde{w}_t)^{(k)} &= \nabla \tilde{\ell}_4(w_t)^{(k)} \\ &= \begin{cases} \frac{3}{8} u_0 & k = 1 \\ -\frac{1}{2} u_0 & k = 2 \\ 0 & \text{otherwise} \end{cases} \\ &= \nabla \ell_4(w_t)^{(k)}. \end{aligned}$$

For $t \geq 4$, it can be observed by Theorem 8 and the proof of Theorem 17 that,

$$\begin{aligned} \ell_4(w_t) &= \max \left( \delta_2, \max_{k \in [T-1], u \in U} \left( \frac{3}{8} \langle u, w_t^{(k)} \rangle - \frac{1}{2} \langle u, w_t^{(k+1)} \rangle \right) \right) \\ &= \max_{k \in [T-1], u \in U} \left( \frac{3}{8} \langle u, w_t^{(k)} \rangle - \frac{1}{2} \langle u, w_t^{(k+1)} \rangle \right) \end{aligned}$$

Moreover, the maximal value is $\frac{3\eta}{16}$ and is attained in $k_0 = t - 2, u = u_0 = \alpha(\psi^*)$. The second maximal possible value of this term is smaller then $\frac{5\eta}{64}$, then we can apply again Theorem 34 and get by the hypothesis of the induction that

$$\nabla \tilde{\ell}_4(\tilde{w}_t)^{(k)} = \nabla \tilde{\ell}_4(w_t)^{(k)}$$
$$= \begin{cases} \frac{3}{8} u_0 & k = t - 2 \\ -\frac{1}{2} u_0 & k = t - 1 \\ 0 & \text{otherwise} \end{cases}$$
$$= \nabla \ell_4(w_t)^{(k)}.$$

In conclusion, we proved that $\nabla \widehat{F}(w_t) = \nabla \widehat{\tilde{F}}(\tilde{w}_t)$, thus, by the hypothesis of the induction,

$$w_{t+1} = w_t - \nabla \widehat{F}(w_t) = \tilde{w}_t - \nabla \widehat{\tilde{F}}(\tilde{w}_t) = \tilde{w}_{t+1}$$

$\blacksquare$

## Appendix B. Underfitting of SGD: Proof of Theorem 2

In this section we provide a formal proof of our main result for SGD given in Theorem 2. Here, our goal is to establish *underfitting*: namely, to show that the algorithm may converge to a solution with an excessively large empirical risk despite successfully converging on the population risk.

As in $GD$, we construct a hard loss function, which is defined in a $d$-dimensional Euclidean space such that $d$ is polynomial in the number of examples $n$. For the first step of the construction, we use Theorem 3 (see Appendix A), which shows for every dimension $d'$ an existence of a set of approximately orthogonal vectors in $U \in \mathbb{R}^{d'}$ with size exponential in $d'$. We define the set $U$ to be $U := U_{d'}$ for $d' = 712n \log n$, the sample space to be $Z^{\text{SGD}} := \{V : V \subseteq U\}$, and the distribution $\mathcal{D}^{\text{SGD}}$ to be such that every $u \in U$ is included in $V \subseteq U$ independently with probability $\delta = \frac{1}{4n^2}$. Henceforth, we consider an arbitrary enumeration of the elements of $U$:

$$U = \{v_1, \dots, v_{|U|}\}. \tag{20}$$

As before, we refer to every vector $w \in \mathbb{R}^d$ as a concatenation of vectors, $w = (w^{(0)}, \dots, w^{(n)})$, where for $1 \le k \le n$, $w^{(k)} \in \mathbb{R}^{d'}$ and $w^{(0)} \in \mathbb{R}^{2n^2}$. In this construction, $w^{(0)}$ is also a concatenation of $n$ vectors $w^{(0,1)}, \dots, w^{(0,n)}$ such that each for every $r \in [n]$, $w^{(0,r)} \in \mathbb{R}^{2n}$.

Our approach is, as in $GD$, in every iteration $t$, to encode the set $V_t$, sampled from $\mathcal{D}^{\text{SGD}}$ into the iterate $w_{t+1}^{(0)}$. For this, we construct an encoder, $\phi : P(U) \times [n] \to \mathbb{R}^{2n}$, a decoder $\alpha : \mathbb{R}^{2n} \to U$, a real number $\epsilon > 0$ and $n$ sets denoted as $\Psi_1, \dots, \Psi_n$. Here, the idea is that the set $\psi_k$ may represent all of the possible training sets with $k$ examples, $\{V_1 \dots, V_k\}$, and in every iteration $t$, it is possible to get the vector $\psi_{t-1}^* \in \Psi_{t-1}$ — which is identified with the actual sets $V_1, \dots, V_{t-1}$ that were sampled before this iteration — as a maximizer of a linear function with margin $\epsilon$. Then, we aim to output a vector $u_t \in \bigcap_{i=t}^n V_i$. The construction of such $\phi, \alpha, \Psi_1, \dots, \Psi_n$ is provided the following.

**Lemma 19** *Let $n$, a set $U \in \mathbb{R}^d$. Let $P(U)$ be the power set of $U$. Then, there exist sets $\{\Psi_1, \dots \Psi_n\} \subseteq \mathbb{R}^{2n}$, a number $0 < \epsilon < \frac{1}{n}$ and two mappings $\phi : P(U) \times [n] \to \mathbb{R}^{2n}$, $\alpha : \mathbb{R}^{2n} \to U$ such that,*

1. *For every $j \in [n]$ and $V \subseteq U$, $\|\phi(V,j)\| \le 1$.*

2. *For every $k$, $\psi \in \Psi_k$, $\|\alpha(\psi)\| \le 1$, $\|\psi\| \le 1$.*

3. *Let $V_1, \ldots, V_k \subseteq U$. Then, for every $k$, $\psi_k^* = \frac{1}{n}\sum_{i=1}^k \phi(V_i, i)$:*

   - *For every $\psi \in \Psi_k$, $\psi \ne \psi_k^*$:*

   $$\langle \psi_k^*, \frac{1}{n}\sum_{i=1}^k \phi(V_i, i)\rangle \ge \langle \psi, \frac{1}{n}\sum_{i=1}^k \phi(V_i, i)\rangle + \epsilon;$$

   - *If $\bigcap_{i=1}^k V_i \ne \emptyset$ and $m = \arg\min\{i : v_i \in U \cap \bigcap_{i=1}^k V_i\}$, then $\alpha(\psi^*) = v_m \in U$.*

The proofs of the above and of other lemmas in this section are deferred to Appendix B.4. Proceeding, we define our loss function $f^{\text{SGD}}$, composed of three terms $\ell_1^{\text{SGD}}$, $\ell_2^{\text{SGD}}$, $\ell_3^{\text{SGD}}$:

$$f^{\text{SGD}}(w, V) := \underbrace{\sqrt{\sum_{k=2}^T \max\left(\frac{3\eta}{32}, \max_{u \in V}\langle u, w^{(k)}\rangle\right)^2}}_{\ell_1^{\text{SGD}}(w,V):=} \tag{21}$$

$$+ \max\left(\delta_1, \max_{k \in [n-1], u \in U, \psi \in \Psi_k}\left(\frac{3}{8}\langle u, w^{(k)}\rangle - \frac{1}{2}\langle \alpha(\psi), w^{(k+1)}\rangle + \langle w^{(0,k)}, \frac{1}{4n}\psi\rangle\right.\right.$$

$$\left.\left. - \langle w^{(0,k+1)}, \frac{1}{4n}\psi\rangle + \langle w^{(0,k+1)}, -\frac{1}{4n^2}\phi(V, k+1)\rangle\right)\right)$$

$$\underbrace{+ \langle w^{(0,1)}, -\frac{1}{4n^2}\phi(V, 1)\rangle - \langle \frac{1}{n^3}u_1, w^{(1)}\rangle,}_{\ell_3^{\text{SGD}}(w,V):=}$$

where the second term is denoted $\ell_2^{\text{SGD}}(w, V)$ and $u_1$ is an arbitrary vector in $U$. The next theorem is the core component of the lower bound.

**Theorem 20** *Assume that $n > 2048$ and $\eta \le \frac{1}{\sqrt{T}}$. Consider the distribution $\mathcal{D}^{\text{SGD}}$ to be such that every $u \in U$ is included in $V \subseteq U$ independently with probability $\delta = \frac{1}{4n^2}$, and the loss function $f^{\text{SGD}}$ with $d = 712n^2 \log n + 2n^2$, $\varepsilon = \frac{1}{n^2}(1 - \cos(\frac{\pi}{2|P(U)|}))$ and $\delta_1 = \frac{\eta}{8n^3}$. Then $f^{\text{SGD}}$ is convex and 4-Lipschitz, and for Unprojected SGD (cf. Eq. (2) with $W = \mathbb{R}^d$) initialized at $w_1 = 0$ with step size $\eta$, we have, with probability at least $\frac{1}{2}$ over the choice of the training sample:*

(i) *The iterates of SGD remain within the unit ball, namely $w_t \in \mathbb{B}^d$ for all $t = 1, \ldots, T$;*

(ii) *For all $m = 1, \ldots, T$, the m-suffix averaged iterate has:*

$$\widehat{F}^{\text{SGD}}(w_{T,m}) - \widehat{F}^{\text{SGD}}(\widehat{w}_*) = \Omega(\eta\sqrt{T}).$$

The proof of Theorem 20 is given in Appendix B.3, and builds upon the lemmas presented next.

**Lemma 21** *For every $V \in Z$, the loss function $f^{\text{SGD}}(w, V)$ specified by Theorem 20 is convex and 4-Lipschitz over $\mathbb{R}^d$ with respect to its first argument.*

As in GD, we provide a key lemma that characterizes the trajectory of SGD under a certain "good event". Consider a random training set sample $S = \{V_i\}_{i=1}^n$, and define:

$$\mathcal{E}' = \{\forall t \leq T \ P_t \neq \emptyset \text{ and } J_t \in S_t\}, \tag{22}$$

$$\text{where } P_t := \bigcap_{i=1}^{t-1} V_i, \tag{23}$$

$$S_t := \bigcap_{i=t}^{t=n} \overline{V_i}, \tag{24}$$

$$\text{and for } P_t \neq \emptyset : r_t := \arg\min\{r : v_r \in U \cap P_t\} \tag{25}$$

$$J_t := v_{r_t} \in U \tag{26}$$

Our next lemma shows that $\mathcal{E}'$ occurs with a constant probability.

**Lemma 22** *Under the conditions of Theorem 20, it holds that* $\Pr(\mathcal{E}') \geq \frac{1}{2}$.

Under this event, the dynamics of $SGD$ is characterized as follows,

**Lemma 23** *Assume the conditions of Theorem 20, and consider the iterates of* unprojected $SGD$, *with step size* $\eta \leq \frac{1}{\sqrt{T}}$ *initialized at* $w_1 = 0$. *Conditioned on the event* $\mathcal{E}'$, *we have for* $t \geq 4$,

$$w_t{}^{(k)} = \begin{cases} -\frac{3}{8}\eta u_1 + (t-1)\frac{\eta}{n^3} u_1 & k = 1 \\ \frac{1}{8}\eta u_k & 2 \leq k \leq t-2 \\ \frac{1}{2}\eta u_{t-1} & k = t-1 \\ 0 & t \leq k \leq n, \end{cases}$$

*and*

$$w_t{}^{(0,k)} = \begin{cases} \frac{\eta}{4n^2} \sum_{i=2}^{t-1} \phi(V_i, 1) & k = 1 \\ \frac{\eta}{4n^2} \sum_{i=1}^{t-1} \phi(V_i, i) & k = t-1 \\ 0 & k \notin \{1, t-1\}. \end{cases}$$

*where* $u_1 \in U$, *and for* $k > 1$: $u_k \in \bigcap_{i=1}^{k-1} V_i \cap \bigcap_{i=k}^{n} \overline{V_i}$.

At this point, we are ready to prove Theorem 20. The proof is similar to that of Theorem 5, using Theorem 23 instead of Theorem 8, and is given in Appendix B.3. All that remains for the proof of Theorem 2 given next, is to apply smoothing to make the learning problem from Theorem 20 differentiable, and combine its result with a standard optimization lower bound (Theorem 38).
**Proof** [of Theorem 2] Let

$$\tilde{f}^{\text{SGD}}(w, V) := \mathbb{E}_{v \sim \text{Unif}(\mathbb{B}^d)} \left[ f^{\text{SGD}}(w + \tilde{\delta}v, V) \right], \tag{27}$$

where

$$\tilde{\delta} := \frac{\eta\epsilon}{32n^3}. \tag{28}$$

Analogously, we denote the empirical loss and the population loss with respect to the loss function $\tilde{f}^{SGD}$ as $\widehat{\tilde{F}}^{SGD}(w) = \frac{1}{n} \sum_{i=1}^{n} \tilde{f}^{SGD}(w, V_i)$ and $\tilde{F}^{SGD}(w) = \mathbb{E}_{V \sim \mathcal{D}} \tilde{f}^{SGD}(w, V)$, respectively. We have that this loss function is convex and Lipschitz, and that the GD iterates identify with those of the original, non-smoothed loss.

**Lemma 24** *For every $V \in Z$, the loss function $\tilde{f}^{SGD}$ is differentiable, convex and 4-Lipschitz with respect to its first argument and over $\mathbb{R}^d$.*

**Lemma 25** *Let $w_t, \tilde{w}_t$ be the iterates of Unprojected SGD with step size $\eta \leq \frac{1}{\sqrt{T}}$ and $w_1 = 0$, on $\widehat{F}^{SGD}$ and $\widehat{\tilde{F}}^{SGD}$ respectively. Then, if $\mathcal{E}'$ occurs, for every $t \in [T]$ it holds that $w_t = \tilde{w}_t$.*

Now let $\overline{w_{n,m}}$ be the $m$-suffix average of $SGD$ on $f^{SGD}$ and let $\overline{\widehat{w}_*} = \arg\min_w \widehat{F}^{SGD}(w)$. By Theorem 25, we know that, with a probability $\frac{1}{2}$, $w_{n,m} = w_{n,m}^{SGD}$. Then, by Theorem 20 and the fact that $\|f^{SGD} - \tilde{f}^{SGD}\|_\infty \leq 4\tilde{\delta}$ (see Theorem 35),

$$\frac{\eta}{64000} \sqrt{n} \leq \widehat{F}^{SGD}(\overline{w_{n,m}}) - \widehat{F}^{SGD}(\overline{\widehat{w}_*})$$

$$= \widehat{F}^{SGD}(w_{n,m}) - \widehat{F}^{SGD}(\overline{\widehat{w}_*})$$

$$\leq \widehat{\tilde{F}}^{SGD}(w_{n,m}) + 4\tilde{\delta} - \widehat{\tilde{F}}^{SGD}(\overline{\widehat{w}_*}) + 4\tilde{\delta}$$

$$\leq \widehat{\tilde{F}}^{SGD}(w_{n,m}) + 4\tilde{\delta} - \widehat{\tilde{F}}^{SGD}(\widehat{w}_*) + 4\tilde{\delta},$$

hence,

$$\widehat{\tilde{F}}^{SGD}(w_{n,m}) - \widehat{\tilde{F}}^{SGD}(\widehat{w}_*) \geq \frac{\eta}{64000} \sqrt{n} - \frac{\eta \epsilon}{4n^3}$$

$$\geq \frac{\eta}{64000} \sqrt{n} - \frac{\eta}{4n^3}$$

$$\geq \frac{\eta}{128000} \sqrt{n} \qquad (n \geq 40)$$

$$=: C_1 \eta \sqrt{n}.$$

In addition, by Theorem 38, we know that for Unprojected $SGD$ and $d_2 = \max(25\eta^2 T^2, 1)$, there exist a constant $C_2$ and a deterministic loss function $\tilde{f}^{OPT} : \mathbb{R}^{d_2} \to \mathbb{R}$ such that

$$\tilde{f}^{OPT}(w_{T,m}) - \tilde{f}^{OPT}(\widehat{w}_*) \geq C_2 \min\left(1, \frac{1}{\eta T}\right)$$

Now, let $C = \frac{1}{2} \min(C_1, C_2)$. If $\eta \geq T^{-\frac{3}{4}}$, then, $\eta \sqrt{T} \geq \min(1, \frac{1}{\eta T})$, and we get,

$$\widehat{\tilde{F}}^{SGD}(w_{T,m}) - \widehat{\tilde{F}}^{SGD}(\widehat{w}_*) \geq C\left(\eta \sqrt{T} + \min\left(1, \frac{1}{\eta T}\right)\right) \geq C\left(\min\left(1, \eta \sqrt{T} + \frac{1}{\eta T}\right)\right).$$

Otherwise, we get that,

$$\tilde{f}^{OPT}(w_{T,m}) - \tilde{f}^{OPT}(\widehat{w}_*) \geq C\left(\eta \sqrt{T} + \min\left(1, \frac{1}{\eta T}\right)\right) \geq C\left(\min\left(1, \eta \sqrt{T} + \frac{1}{\eta T}\right)\right).$$

In both cases, by Theorems 23 and 38, $w_t \in \mathbb{B}^d$ for every $t \in [T]$, hence the result is applicable also for Projected SGD, and the proof is complete. ■

### B.1. SGD Dynamics: Proof of Theorem 23

For proving this key lemma, we analyze how the terms $\ell_1^{SGD}, \ell_2^{SGD}$ affects the dynamics of SGD under the event $\mathcal{E}'$. First, we show that the gradients of $\ell_1^{SGD}$ do not affect the dynamics of $SGD$, as the gradient of this term at any iterate $w_t$ is zero.

**Lemma 26** *Assume the conditions of Theorem 20 and the event $\mathcal{E}'$. Let $w \in \mathbb{R}^d$ and $t$ be such that for every $2 \leq k \leq t-1$, $w^{(k)} = c\eta u_k$ for $0 \leq c \leq \frac{1}{2}$ and every such $u_k$ holds $u_k \in \bigcap_{i=1}^{k-1} V_i \cap \bigcap_{i=k}^n \overline{V_i}$, and for every $t \leq k \leq T$, $w^{(k)} = 0$. Then $\ell_1^{SGD}(\cdot, V_t)$ is differentiable at $w$ and $\nabla \ell_1^{SGD}(w, V_t) = 0$.*

**Proof** First, by the fact that for every $t \leq k \leq T$, $w^{(k)} = 0$, for every such $k$,

$$\max_{u \in V_t} \langle u, w^{(k)} \rangle = 0 < \frac{3\eta}{32},$$

For $2 \leq k \leq t-1$, $w^{(k)} = c\eta u_k$, where $0 \leq c \leq \frac{1}{2}$ and every $u_k \in \bigcap_{i=k}^T \overline{V_i} \subseteq \overline{V_t}$, thus,

$$\max_{u \in V_t} \langle u, w^{(k)} \rangle \leq \frac{\eta}{2} \cdot \frac{1}{8} < \frac{3\eta}{32}.$$

We derive that $\nabla \ell_1^{SGD}(w_t, V_t) = 0$. ∎

Now, we analyze the gradient of $\ell_2^{SGD}$. The role of this component is to decode the next "bad solution" $\alpha\left(\frac{1}{n}\sum_{i=1}^{t-1}\phi(V_i, i)\right)$ from the sets $V_1, \ldots, V_{t-1}$, and make a progress in this direction in some subspace $W^{(t-1)}$. In the following lemma, we show that the gradient of $\ell_2^{SGD}$, serves this goal.

**Lemma 27** *Assume the conditions of Theorem 20 and the event $\mathcal{E}'$. For every $k$, let $\psi_k^* = \frac{1}{n}\sum_{t=1}^k \phi(V_t, t)$. Moreover, let $m \geq 3$ and $w \in \mathbb{R}^d$ such that $w^{(1)} = c\eta u_1$ for $-\frac{3}{8} \leq c \leq 0$ and $u_1 \in U$, for every $2 \leq k \leq m-1$, $w^{(k)} = \frac{1}{8}\eta u_k$ such that every $u_k$ holds $u_k \in \bigcap_{t=1}^{k-1} V_t \cap \bigcap_{i=k}^n \overline{V_t}$, $w^{(m)} = \frac{1}{2}\eta u_m$ where $u_m$ holds $u_m \in \bigcap_{t=1}^{m-1} V_t \cap \bigcap_{i=m}^n \overline{V_t}$ and for every $m+1 \leq k \leq T$, $w^{(k)} = 0$. Moreover, assume that $w$ holds $w^{(0,m)} = \frac{\eta}{4n}\psi_m^*$, $\|w^{(0,1)}\| \leq \eta$ and for every $k \notin \{m, 1\}$, $w^{(0,k)} = 0$. Then, for every $V \subseteq U$, $\ell_2^{SGD}$ is differentiable at $(w, V)$ and, we have for $k \neq 0$,*

$$\nabla \ell_2^{SGD}(w, V)^{(k)} = \begin{cases} \frac{3}{8}u_m & k = m \\ -\frac{1}{2}\alpha(\psi_m^*) & k = m+1 \\ 0 & k \notin \{m, m+1\} \end{cases}$$

*and,*

$$\nabla \ell_2^{SGD}(w, V)^{(0,k)} = \begin{cases} \frac{1}{4n^2}\sum_{t=1}^m \phi(V_t, i) & k = m \\ -\frac{1}{4n^2}\sum_{t=1}^m \phi(V_t, i) - \frac{1}{4n^2}\phi(V, i) & k = m+1 \\ 0 & k \notin \{m, m+1\}. \end{cases}$$

**Proof** First, we show that the maximum of $\ell_2^{SGD}(w, V)$ is attained with $k = m$ and $u = u_m$. For $k \geq m+1$, for every $u \in U$ and $\psi \in \Psi_k$,

$$\frac{3}{8}\langle u, w^{(k)} \rangle - \frac{1}{2}\langle \alpha(\psi), w^{(k+1)} \rangle + \langle w^{(0,k)}, \frac{1}{4n}\psi \rangle$$

$$- \langle w^{(0,k+1)}, \frac{1}{4n}\psi \rangle + \langle w^{(0,k+1)}, -\frac{1}{4n^2}\phi(V, k+1) \rangle = 0.$$

For $k = 1$, for every $u \in U$ and $\psi \in \Psi_1$, by Theorem 19, we know that for every $\psi, V, j$, $\|\psi\|, \|\phi(V, j)\| \leq 1$, and $\alpha(\psi) \in U$, thus,

$$\frac{3}{8}\langle u, w^{(k)} \rangle - \frac{1}{2}\langle \alpha(\psi), w^{(k+1)} \rangle + \langle w^{(0,k)}, \frac{1}{4n}\psi \rangle - \langle w^{(0,k+1)}, \frac{1}{4n}\psi \rangle + \langle w^{(0,k+1)}, -\frac{1}{4n^2}\phi(V, k+1) \rangle$$

$$= \frac{3c}{8}\langle u_1, u \rangle - \frac{\eta}{16}\langle u_2, \alpha(\psi) \rangle + \langle w^{(0,k)}, \frac{1}{4n}\psi \rangle - 0 + 0$$

$$\leq \frac{9\eta}{512} + \frac{\eta}{128} + \frac{\eta}{4n}$$

$$< \frac{\eta}{8}. \qquad\qquad (n \geq 4)$$

For $2 \leq k \leq m - 2$, for every $u \in U$ and $\psi \in \Psi_k$, by Theorem 19, we know that for every $\psi, V, j$, $\|\psi\|, \|\phi(V, j)\| \leq 1$, and $\alpha(\psi) \in U$, thus,

$$\frac{3}{8}\langle u, w^{(k)} \rangle - \frac{1}{2}\langle \alpha(\psi), w^{(k+1)} \rangle + \langle w^{(0,k)}, \frac{1}{4n}\psi \rangle - \langle w^{(0,k+1)}, \frac{1}{4n}\psi \rangle + \langle w^{(0,k+1)}, -\frac{1}{4n^2}\phi(V, k+1) \rangle$$

$$= \frac{3}{64}\langle u_k, u \rangle - \frac{\eta}{16}\langle u_{k+1}, \alpha(\psi) \rangle + 0 - 0 + 0$$

$$\leq \frac{3\eta}{64} + \frac{\eta}{16}$$

$$< \frac{\eta}{8}.$$

For $k = m - 1$, for every $u \in U$ and $\psi \in \Psi_k$, by Theorem 19, we know that for every $\psi, V, j$, $\|\psi\|, \|\phi(V, j)\| \leq 1$, and $\alpha(\psi) \in U$, thus,

$$\frac{3}{8}\langle u, w^{(k)} \rangle - \frac{1}{2}\langle \alpha(\psi), w^{(k+1)} \rangle + \langle w^{(0,k)}, \frac{1}{4n}\psi \rangle - \langle w^{(0,k+1)}, \frac{1}{4n}\psi \rangle + \langle w^{(0,k+1)}, -\frac{1}{4n^2}\phi(V, k+1) \rangle$$

$$= \frac{3}{64}\langle u_k, u \rangle - \frac{\eta}{4}\langle u_{k+1}, \alpha(\psi) \rangle + 0 - \langle w^{(0,k+1)}, \frac{1}{4n}\psi \rangle + \langle w^{(0,k+1)}, \frac{1}{4n^2}\phi(V, m) \rangle$$

$$\leq \frac{3\eta}{64} + \frac{\eta}{32} + \frac{1}{16n^2} + \frac{1}{16n^3}$$

$$< \frac{\eta}{8}. \qquad\qquad (n \geq 4)$$

For $k = m$, $u \neq u_m$ and every $\psi \in \Psi_m$, by Theorem 19, we know that for every $\psi, V, j$, $\|\psi\|, \|\phi(V, j)\| \leq 1$, and $\alpha(\psi) \in U$, thus,

$$\frac{3}{8}\langle u, w^{(k)} \rangle - \frac{1}{2}\langle \alpha(\psi), w^{(k+1)} \rangle + \langle w^{(0,k)}, \frac{1}{4n}\psi \rangle - \langle w^{(0,k+1)}, \frac{1}{4n}\psi \rangle + \langle w^{(0,k+1)}, -\frac{1}{4n^2}\phi(V, k+1) \rangle$$

$$= \frac{3}{8}\langle u, w^{(k)} \rangle + \langle \frac{1}{4n}\psi, w^{(0,k)} \rangle$$

$$= \frac{3}{8}\langle u, \frac{\eta}{2}u_m \rangle + \langle \frac{1}{4n}\psi, w^{(0,k)} \rangle$$

$$\leq \frac{3\eta}{128} + \frac{\eta}{16n^2}$$

$$< \frac{\eta}{32}. \qquad (n \geq 4)$$

For $k = m$, $u = u_m$ and every $\psi \in \Psi_m$, by Theorem 19, we know that for every $\psi, V, j$, $\|\psi\|, \|\phi(V, j)\| \leq 1$, and $\alpha(\psi) \in U$, thus,

$$\frac{3}{8}\langle u, w^{(k)} \rangle - \frac{1}{2}\langle \alpha(\psi), w^{(k+1)} \rangle + \langle w^{(0,k)}, \frac{1}{4n}\psi \rangle - \langle w^{(0,k+1)}, \frac{1}{4n}\psi \rangle + \langle w^{(0,k+1)}, -\frac{1}{4n^2}\phi(V, k+1) \rangle$$

$$= \frac{3}{8}\langle u, w_t^{(k)} \rangle + \langle \frac{1}{4n}\psi, w_t^{(0,k)} \rangle$$

$$= \frac{3}{8}\langle u, \frac{\eta}{2}u_m \rangle + \langle \frac{1}{4n}\psi, w_t^{(0,k)} \rangle$$

$$\geq \frac{3\eta}{16} - \frac{\eta}{16n^2}$$

$$> \frac{5\eta}{32} \qquad (n \geq 4)$$

$$> \delta_1.$$

Second, we show that when $k = m$ and $u = u_m$, the maximum among $\psi \in \Psi_m$ is attained uniquely in $\psi_m^* = \frac{1}{n}\sum_{t=1}^{m}\phi(V_t, t)$. For any $\psi \in \Psi_m$, with $\psi \neq \psi_m^*$, by Theorem 19, for $k = m$, $u = u_m$,

$$\frac{3}{8}\langle u, w^{(m)} \rangle - \frac{1}{2}\langle \alpha(\psi_m^*), w^{(m+1)} \rangle + \langle w^{(0,m)}, \frac{1}{4n}\psi_m^* \rangle$$

$$- \langle w^{(0,m+1)}, \frac{1}{4n}\psi_m^* \rangle + \langle w^{(0,m+1)}, -\frac{1}{4n^2}\phi(V, m+1) \rangle$$

$$= \frac{3}{8}\langle u, w^{(m)} \rangle + \langle \frac{1}{4n}\psi_m^*, w^{(0,m)} \rangle$$

$$= \frac{3\eta}{16} + \frac{\eta}{16n^2}\langle \psi_m^*, \frac{1}{n}\sum_{t=1}^{m}\phi(V_t, t) \rangle$$

$$\geq \frac{3\eta}{16} + \frac{\eta}{16n^2}\langle \psi, \frac{1}{n}\sum_{t=1}^{m}\phi(V_t, t) \rangle + \frac{\eta\epsilon}{16n^2}$$

$$= \frac{3}{8}\langle u, w^{(k)} \rangle + \langle \frac{1}{4n}\psi, w^{(0,m)} \rangle + \frac{\eta\epsilon}{16n^2}$$

$$= \frac{3}{8}\langle u, w^{(m)} \rangle - \frac{1}{2}\langle \alpha(\psi), w^{(m+1)} \rangle + \langle w^{(0,m)}, \frac{1}{4n}\psi \rangle$$

$$- \langle w^{(0,m+1)}, \frac{1}{4n}\psi \rangle + \langle w^{(0,m+1)}, -\frac{1}{4n^2}\phi(V, m+1) \rangle + \frac{\eta\epsilon}{16n^2}.$$

We derive that,

$$\nabla\ell_2^{\text{SGD}}(w, V)^{(k)} = \begin{cases} \frac{3}{8}u_m & k = m \\ -\frac{1}{2}\alpha(\psi_m^*) & k = m+1 \\ 0 & k \notin \{m, m+1\} \end{cases}$$

$$\nabla\ell_2^{\text{SGD}}(w, V)^{(0,k)} = \begin{cases} \frac{1}{4n^2}\sum_{t=1}^{m}\phi(V_t, t) & k = m \\ -\frac{1}{4n^2}\sum_{t=1}^{m}\phi(V_t, t) - \frac{1}{4n^2}\phi(V, m+1) & k = m+1 \\ 0 & k \notin \{m, m+1\}. \end{cases}$$

Now we can prove Theorem 23.

**Proof** [of Theorem 23] We assume that $\mathcal{E}'$ holds and prove the lemma by induction on $t$. We begin from the basis of the induction, $t = 4$, which is proved in Theorem 30 in Appendix B.2. Now, we assume the hypothesis of the induction, that the lemma holds for iteration $t$ and turn to show the required for iteration $t + 1$.

First, we notice that for every $2 \leq k \leq t - 1$, $w_t^{(k)} = c\eta u_k$ for $c \leq \frac{1}{2}$ and every such $u_k$ holds $u_k \in \bigcap_{i=1}^{k-1} V_i \cap \bigcap_{i=k}^{n} \overline{V_i}$, and for every $t \leq k \leq T$, $w_t^{(k)} = 0$. Then, by Theorem 26, we have that $\nabla \ell_1^{\text{SGD}}(w_t, V_t) = 0$.

Second, $\ell_3^{\text{SGD}}$ is a linear function, thus,

$$\nabla \ell_3^{\text{SGD}}(w_t, V_t)^{(s)} = \begin{cases} -\frac{1}{n^3} u_1 & s = 1 \\ -\frac{1}{4n^2} \phi(V_t, 1) & s = 0, 1 \\ 0 & \text{otherwise.} \end{cases}$$

Third, For $\ell_2^{\text{SGD}}(w_t, V_t)$, we notice for $m = t - 1 \geq 3$ it holds that $w_t^{(1)} = c\eta u_1$ for $-\frac{3}{8} \leq c \leq 0$ and $u_1 \in U$, for every $2 \leq k \leq m - 1$, $w_t^{(k)} = \frac{1}{8}\eta u_k$ such that every $u_k$ holds $u_k \in \bigcap_{t=1}^{k-1} V_t \cap \bigcap_{t=k}^{n} \overline{V_t}$, $w_t^{(m)} = \frac{1}{2}\eta u_m$ where $u_m$ holds $u_m \in \bigcap_{t=1}^{m-1} V_t \cap \bigcap_{i=m}^{n} \overline{V_t}$, and for every $m + 1 \leq k \leq T$, $w^{(k)} = 0$. Moreover, $w_t$ holds $w^{(0,m)} = \frac{\eta}{4n} \psi_m^*$, $\|w_t^{(0,1)}\| \leq \eta$ and for every $k \notin \{m, 1\}$, $w_t^{(0,k)} = 0$. Then, by Theorem 27, we get that, we have for $k \neq 0$,

$$\nabla \ell_2^{\text{SGD}}(w_t, V_t)^{(k)} = \begin{cases} \frac{3}{8} u_{t-1} & k = t - 1 \\ -\frac{1}{2}\alpha(\psi_{t-1}^*) & k = t \\ 0 & k \notin \{t - 1, t\} \end{cases}$$

and,

$$\nabla \ell_2^{\text{SGD}}(w_t, V_t)^{(0,k)} = \begin{cases} \frac{1}{4n^2} \sum_{i=1}^{t-1} \phi(V_i, i) & k = m \\ -\frac{1}{4n^2} \sum_{i=1}^{t} \phi(V_i, i) & k = m + 1 \\ 0 & k \notin \{m, m + 1\}. \end{cases}$$

Now, by Theorem 19, for $j = \arg\min_i \{i : v_i \in \bigcap_{i=1}^{t-1} V_i\}$, we get that

$$\alpha(\psi_{t-1}^*) = v_j \in \bigcap_{i=1}^{t-1} V_i.$$

We notice that $\bigcap_{i=1}^{t-1} V_i = P_t$ and thus $\alpha(\psi_{t-1}^*) = J_t$. Then, by $\mathcal{E}'$, $\alpha(\psi_{t-1}^*)$ also holds $\alpha(\psi_{t-1}^*) \in S_t$. Combining the above together, we get, for $u_t = \alpha(\psi_{t-1}^*) \in P_t \cap S_t$,

$$\nabla f(w_t, V_t)^{(k)} = \begin{cases} -\frac{1}{n^3} u_1 & k = 1 \\ \frac{3}{8} u_{t-1} & k = t - 1 \\ -\frac{1}{2} u_t & k = t \\ 0 & k \notin \{1, t - 1, t\}, \end{cases}$$

and,

$$\nabla f(w_t, V_t)^{(0,k)} = \begin{cases} -\frac{1}{4n^2}\phi(V_3, 1) & k = 1 \\ \frac{1}{4n^2}\sum_{i=1}^{t-1}\phi(V_i, i) & k = t - 1 \\ -\frac{1}{4n^2}\sum_{i=1}^{t}\phi(V_i, i) & k = t \\ 0 & k \notin \{1, t - 1, t\}, \end{cases}$$

and the lemma follows. ∎

## B.2. SGD Dynamics: The first iterates

**Lemma 28** *Under the conditions of Theorem 20, if $\mathcal{E}'$ occurs and $w_t$ is the iterate of Unprojected SGD with step size $\eta \leq \frac{1}{\sqrt{n}}$ and $w_1 = 0$,*

$$w_2^{(k)} = \begin{cases} \frac{\eta}{n^3}u_1 & k = 1 \\ 0 & k \geq 2 \end{cases},$$

*and,*

$$w_2^{(0,k)} = \begin{cases} \frac{\eta}{4n^2}\phi(V_1, 1) & k = 1 \\ 0 & k \neq 1. \end{cases}$$

**Proof** $w_1 = 0$, thus, for every $k$,

$$\max_{u \in V_1}\langle u, w_1^{(k)}\rangle = 0 < \frac{3\eta}{32},$$

and we derive that $\nabla\ell_1^{\text{SGD}}(w_1, V_1) = 0$. By the same argument, $\nabla\ell_2^{\text{SGD}}(w_1, V_1) = 0$ (where the maximum is attained uniquely in $\delta_2$). Moreover, $\ell_3^{\text{SGD}}$ is a linear function, then, we get that,

$$\nabla\ell_3^{\text{SGD}}(w_1, V_1)^{(k)} = \begin{cases} -\frac{1}{n^3}u_1 & k = 1 \\ 0 & k \geq 2 \end{cases},$$

and,

$$\nabla\ell_3^{\text{SGD}}(w_1, V_1)^{(0,k)} = \begin{cases} -\frac{1}{4n^2}\phi(V_1, 1) & k = 1 \\ 0 & k \neq 1, \end{cases}$$

and the lemma follows. ∎

**Lemma 29** *Under the conditions of Theorem 20, if $\mathcal{E}'$ occurs and $w_t$ is the iterate of Unprojected SGD with step size $\eta \leq \frac{1}{\sqrt{n}}$ and $w_1 = 0$,*

$$w_3^{(k)} = \begin{cases} \frac{2\eta}{n^3}u_1 - \frac{3\eta}{8}u_1 & k = 1 \\ \frac{\eta}{2}u_2 & k = 2 \\ 0 & 3 \leq k \leq n \end{cases}$$

$$w_3^{(0,k)} = \begin{cases} \frac{\eta}{4n^2}\phi(V_2,1) & k = 1 \\ \frac{\eta}{4n^2}\phi(V_1,1) + \frac{\eta}{4n^2}\phi(V_2,2) & k = 2 \\ 0 & k \geq 3. \end{cases}$$

*where $u_1 \in U$, and $u_2$ holds $u_2 \in P_2 \cap S_2$.*

**Proof**

First, by the fact that for every $2 \leq k \leq T$, $w_2^{(k)} = 0$, for every such $k$,

$$\max_{u \in V_2}\langle u, w_2^{(k)}\rangle = 0 < \frac{3\eta}{32},$$

and we derive that $\nabla\ell_1^{\text{SGD}}(w_2, V_2) = 0$.

Moreover, $\ell_3^{\text{SGD}}$ is a linear function, thus,

$$\ell_3^{\text{SGD}}(w_2, V_2)^{(k)} = \begin{cases} \frac{\eta}{n^3}u_1 & k = 1 \\ 0 & k \geq 2 \end{cases},$$

and,

$$\ell_3^{\text{SGD}}(w_2, V_2)^{(k)} = \begin{cases} \frac{\eta}{4n^2}\phi(V_2,1) & k = 1 \\ 0 & k \neq 1. \end{cases}$$

For $\ell_2^{\text{SGD}}(w_2, V_2)$, we get by the fact that for every $k \geq 1$, $w_2^{(k+1)} = w_2^{(0,k+1)} = 0$,

$$\ell_2^{\text{SGD}}(w_2, V_2) = \max\left(\delta_2, \max_{k \in [n-1], u \in U, \psi \in \Psi_k}\left(\frac{3}{8}\langle u, w_2^{(k)}\rangle + \langle \frac{1}{4n}\psi, w_2^{(0,k)}\rangle\right)\right)$$

As a first step, we show that the the maximum is attained with $k = 1$ and $u = u_1$, For $k \neq 1$, for every $u \in U$ and $\psi \in \Psi_k$,

$$\frac{3}{8}\langle u, w_2^{(k)}\rangle + \langle \frac{1}{4n}\psi, w_2^{(0,k)}\rangle = 0.$$

For $k = 1$, $u \neq u_1$ and every $\psi \in \Psi_1$, by the fact that $\|\psi\|, \|\phi(V_1,1)\| \leq 1$,

$$\frac{3}{8}\langle u, w_2^{(k)}\rangle + \langle \frac{1}{4n}\psi, w_2^{(0,k)}\rangle \leq \frac{3\eta}{64n^3} + \frac{\eta}{16n^3} = \frac{7\eta}{64n^3} < \frac{3\eta}{16n^3}.$$

For $k = 1$, $u = u_1$ and every $\psi \in \Psi_1$, by the fact that $\|\psi\|, \|\phi(V_1,1)\| \leq 1$,

$$\frac{3}{8}\langle u, w_2^{(k)}\rangle + \langle \frac{1}{4n^2}\psi, w_2^{(0,k)}\rangle \geq \frac{3\eta}{8n^3} - \frac{\eta}{16n^3} > \frac{3\eta}{16n^3} > \delta_1.$$

As a second step we show that the maximum among $\psi \in \Psi_1$ is attained uniquely in $\psi_1^* = \frac{1}{n}\phi(V_1,1)$. For any $\psi \in \Psi_1$, with $\psi \neq \psi_1^*$. By Theorem 19, for $k = 1$, $u = u_1$,

$$\frac{3}{8}\langle u, w_2^{(k)}\rangle + \langle \frac{1}{4n}\psi_1^*, w_2^{(0,k)}\rangle = \frac{3\eta}{8n^3} + \langle \frac{1}{4n}\psi_1^*, \frac{\eta}{4n^2}\phi(V_1,1)\rangle$$

$$= \frac{3\eta}{8n^3} + \frac{\eta}{16n^2}\langle \psi_1^*, \frac{1}{n}\phi(V_1,1)\rangle$$

$$\geq \frac{3\eta}{8n^3} + \frac{\eta}{16n^2}\langle\psi, \frac{1}{n}\phi(V_1, 1)\rangle + \frac{\eta\epsilon}{16n^2}$$

$$= \frac{3}{8}\langle u, w_2{}^{(k)}\rangle + \langle\frac{1}{4n}\psi, w_2{}^{(0,k)}\rangle + \frac{\eta\epsilon}{16n^2}$$

We got that the maximum is uniquely attained at $k = 1, u = u_1, \psi = \frac{1}{n}\phi(V_1, 1)$. Now, by Theorem 19, for $j = \arg\min_i\{i : v_i \in V_1\}$, we get that

$$\alpha(\psi) = v_j \in V_1.$$

We notice that $V_1 = P_2$ and thus $\alpha(\psi) = J_2$. Then, by $\mathcal{E}'$, $\alpha(\psi)$ also holds $\alpha(\psi) \in S_2$. Combining the above together, we get, for $u_2 = \alpha(\psi) \in P_2 \cap S_2$,

$$\nabla f(w_2, V_2)^{(k)} = \begin{cases} \frac{3}{8}u_1 - \frac{1}{n^3}u_1 & k = 1 \\ -\frac{1}{2}u_2 & k = 2 \\ 0 & k \geq 3 \end{cases}$$

and,

$$\nabla f(w_2, V_2)^{(0,k)} = \begin{cases} \frac{1}{4n^2}\phi(V_1, 1) - \frac{1}{4n^2}\phi(V_2, 1) & k = 1 \\ -\frac{1}{4n^2}\phi(V_1, 1) - \frac{1}{4n^2}\phi(V_2, 2) & k = 2 \\ 0 & k \geq 3, \end{cases}$$

and the lemma follows. ■

**Lemma 30** *Under the conditions of Theorem 20, if $\mathcal{E}'$ occurs and $w_t$ is the iterate of Unprojected SGD with step size $\eta \leq \frac{1}{\sqrt{n}}$ and $w_1 = 0$,*

$$w_4{}^{(k)} = \begin{cases} \frac{3\eta}{n^3}u_1 - \frac{3\eta}{8}u_1 & k = 1 \\ \frac{\eta}{8}u_2 & k = 2 \\ \frac{\eta}{2}u_3 & k = 3 \\ 0 & k \geq 4 \end{cases},$$

*and,*

$$w_4{}^{(0,k)} = \begin{cases} \frac{\eta}{4n^2}\phi(V_2, 1) + \frac{\eta}{4n^2}\phi(V_3, 1) & k = 1 \\ \frac{\eta}{4n^2}\phi(V_1, 1) + \frac{\eta}{4n^2}\phi(V_2, 2) + \frac{\eta}{4n^2}\phi(V_3, 3) & k = 3 \\ 0 & k \notin \{1, 3\} \end{cases}.$$

**Proof** *First, we notice that by Theorem 29, it holds that $w_3{}^{(2)} = c\eta u_2$ for $c \leq \frac{1}{2}$ and $u_2$ holds $u_2 \in V_1 \cap \bigcap_{i=2}^n \overline{V_i}$, and for every $3 \leq k \leq T$, $w_t{}^{(k)} = 0$. Then, by Theorem 26, we have that $\nabla\ell_1^{SGD}(w_3, V_3) = 0$. Moreover, $\ell_3^{SGD}$ is a linear function, thus,*

$$\ell_3^{SGD}(w_3, V_3)^{(k)} = \begin{cases} \frac{\eta}{n^3}u_1 & k = 1 \\ 0 & k \geq 2 \end{cases},$$

*and,*

$$\ell_3^{SGD}(w_3, V_3)^{(k)} = \begin{cases} \frac{\eta}{4n^2}\phi(V_3, 1) & k = 1 \\ 0 & k \neq 1. \end{cases}$$

*For $\ell_2^{SGD}(w_3, V_3)$, we first show that the the maximum is attained with $k = 2$ and $u = u_2$. For $k \geq 3$, for every $u \in U$ and $\psi \in \Psi_k$,*

$$\frac{3}{8}\langle u, w_3^{(k)}\rangle - \frac{1}{2}\langle \alpha(\psi), w_3^{(k+1)}\rangle + \langle w_3^{(0,k)}, \frac{1}{4n}\psi\rangle$$
$$- \langle w_3^{(0,k+1)}, \frac{1}{4n}\psi\rangle + \langle w_3^{(0,k+1)}, -\frac{1}{4n^2}\phi(V, k+1)\rangle = 0.$$

*For $k = 1$, for every $u \in U$ and $\psi \in \Psi_1$, by the fact that for every $\psi, V, j, \|\psi\|, \|\phi(V, j)\| \leq 1$,*

$$\frac{3}{8}\langle u, w_3^{(k)}\rangle - \frac{1}{2}\langle \alpha(\psi), w_3^{(k+1)}\rangle + \langle w^{(0,k)}, \frac{1}{4n}\psi\rangle$$
$$- \langle w_3^{(0,k+1)}, \frac{1}{4n}\psi\rangle + \langle w_3^{(0,k+1)}, -\frac{1}{4n^2}\phi(V, k+1)\rangle$$
$$= \frac{3}{8}(\frac{2\eta}{n^3} - \frac{3\eta}{8})\langle u_1, u\rangle - \frac{\eta}{4}\langle u_2, \alpha(\psi)\rangle + \langle \frac{1}{4n^2}\phi(V_2, 1), \psi\rangle - \langle \frac{\eta}{4n^2}\phi(V_1, 1) + \frac{\eta}{4n^2}\phi(V_2, 2), \psi\rangle$$
$$+ \langle \frac{\eta}{4n^2}\phi(V_1, 1) + \frac{\eta}{4n^2}\phi(V_2, 2), \frac{1}{4n^2}\phi(V_3, 2)\rangle$$
$$\leq \frac{9\eta}{512} + \frac{\eta}{32} + \frac{\eta}{4n^2} + \frac{\eta}{2n^2} + \frac{\eta}{8n^4}$$
$$< \frac{29\eta}{256} \qquad\qquad (n \geq 4)$$
$$< \frac{\eta}{8}.$$

*For $k = 2$, $u \neq u_2$ and every $\psi \in \Psi_2$, , by the fact that for every $\psi, V, j, \|\psi\|, \|\phi(V, j)\| \leq 1$,*

$$\frac{3}{8}\langle u, w_3^{(k)}\rangle - \frac{1}{2}\langle \alpha(\psi), w_3^{(k+1)}\rangle + \langle w_3^{(0,k)}, \frac{1}{4n}\psi\rangle$$
$$- \langle w_3^{(0,k+1)}, \frac{1}{4n}\psi\rangle + \langle w_3^{(0,k+1)}, -\frac{1}{4n^2}\phi(V, k+1)\rangle$$
$$= \frac{3}{8}\langle u, w_3^{(k)}\rangle + \langle \frac{1}{4n}\psi, w_3^{(0,k)}\rangle$$
$$= \frac{3}{8}\langle u, \frac{\eta}{2}u_2\rangle + \langle \frac{1}{4n}\psi, \frac{\eta}{4n^2}\phi(V_1, 1) + \frac{\eta}{4n^2}\phi(V_2, 2)\rangle$$
$$\leq \frac{3\eta}{128} + \frac{\eta}{8n^3}$$
$$< \frac{\eta}{32}. \qquad\qquad (n \geq 4)$$

*For $k = 2$, $u = u_2$ and every $\psi \in \Psi_2$, by the fact that $\|\psi\|, \|\phi(V_1, 1)\| \leq 1$,*

$$\frac{3}{8}\langle u, w_3^{(k)}\rangle - \frac{1}{2}\langle \alpha(\psi), w_3^{(k+1)}\rangle + \langle w_3^{(0,k)}, \frac{1}{4n}\psi\rangle$$

$$- \langle w_3^{(0,k+1)}, \frac{1}{4n}\psi \rangle + \langle w_3^{(0,k+1)}, -\frac{1}{4n^2}\phi(V, k+1) \rangle$$

$$= \frac{3}{8}\langle u, w_3^{(k)} \rangle + \langle \frac{1}{4n}\psi, w_3^{(0,k)} \rangle$$

$$= \frac{3}{8}\langle u, \frac{\eta}{2}u_2 \rangle + \langle \frac{1}{4n}\psi, \frac{\eta}{4n^2}\phi(V_1, 1) + \frac{\eta}{4n^2}\phi(V_2, 2) \rangle$$

$$\geq \frac{3\eta}{16} - \frac{\eta}{8n^3}$$

$$> \frac{5\eta}{32} \qquad\qquad (n \geq 4)$$

$$> \delta_1.$$

*Second, we show that the maximum among $\psi \in \Psi_2$ is attained uniquely in $\psi_2^* = \frac{1}{n}\phi(V_1, 1) + \frac{1}{n}\phi(V_2, 2)$. For any $\psi \in \Psi_2$, with $\psi \neq \psi_2^*$, by Theorem 19, for $k = 2, u = u_2$,*

$$\frac{3}{8}\langle u, w_3^{(k)} \rangle - \frac{1}{2}\langle \alpha(\psi_2^*), w_3^{(k+1)} \rangle + \langle w_3^{(0,k)}, \frac{1}{4n}\psi_2^* \rangle$$

$$- \langle w_3^{(0,k+1)}, \frac{1}{4n}\psi_2^* \rangle + \langle w_3^{(0,k+1)}, -\frac{1}{4n^2}\phi(V, k+1) \rangle$$

$$= \frac{3}{8}\langle u, w_3^{(k)} \rangle + \langle \frac{1}{4n}\psi_2^*, w_3^{(0,k)} \rangle$$

$$= \frac{3\eta}{16} + \langle \frac{1}{4n}\psi_2^*, \frac{\eta}{4n^2}\phi(V_1, 1) + \frac{\eta}{4n^2}\phi(V_2, 2) \rangle$$

$$= \frac{3\eta}{16} + \frac{\eta}{16n^2}\langle \psi_2^*, \frac{1}{n}\phi(V_1, 1) + \frac{1}{n}\phi(V_2, 2) \rangle$$

$$\geq \frac{3\eta}{16} + \frac{\eta}{16n^2}\langle \psi, \frac{1}{n}\phi(V_1, 1) + \frac{1}{n}\phi(V_2, 2) \rangle + \frac{\eta\epsilon}{16n^2}$$

$$= \frac{3}{8}\langle u, w_3^{(k)} \rangle + \langle \frac{1}{4n}\psi, w_3^{(0,k)} \rangle + \frac{\eta\epsilon}{16n^2}$$

$$= \frac{3}{8}\langle u, w_3^{(k)} \rangle - \frac{1}{2}\langle \alpha(\psi), w_3^{(k+1)} \rangle + \langle w_3^{(0,k)}, \frac{1}{4n}\psi \rangle$$

$$- \langle w_3^{(0,k+1)}, \frac{1}{4n}\psi \rangle + \langle w_3^{(0,k+1)}, -\frac{1}{4n^2}\phi(V, k+1) \rangle + \frac{\eta\epsilon}{16n^2}$$

*We got that the maximum is uniquely attained at $k = 2, u = u_2, \psi = \psi_2^*$. Now, by Theorem 19, for $j = \arg\min_i\{i : v_i \in V_1 \cap V_2\}$, we get that*

$$\alpha(\psi) = v_j \in V_1 \cap V_2.$$

We notice that $V_1 \cap V_2 = P_3$ and thus $\alpha(\psi) = J_3$. Then, by $\mathcal{E}'$, $\alpha(\psi)$ also holds $\alpha(\psi) \in S_3$. Combining the above together, we get, for $u_1 \in U$, $u_2 \in P_2 \cap S_2$ and $u_3 = \alpha(\psi_2^*) \in P_3 \cap S_3$,

$$\nabla f(w_3, V_3) = \begin{cases} -\frac{1}{n^3} u_1 & s = 1 \\ \frac{3}{8} u_2 & s = 2 \\ -\frac{1}{2} u_3 & s = 3 \\ 0 & 4 \leq s \leq n \\ -\frac{1}{4n^2} \phi(V_3, 1) & s = 0, 1 \\ \frac{1}{4n^2} \phi(V_1, 1) + \frac{1}{4n^2} \phi(V_2, 2) & s = 0, 2 \\ -\frac{1}{4n^2} \phi(V_1, 1) - \frac{1}{4n^2} \phi(V_2, 2) - \frac{1}{4n^2} \phi(V_3, 3) & s = 0, 3 \\ 0 & s = 0, k \text{ for } k \geq 3, \end{cases}$$

and the lemma follows. ∎

### B.3. Proof of Theorem 20

**Proof** [of Theorem 20] We show that the theorem holds if the event $\mathcal{E}'$ occurs. First, we prove that for every $t$, $\|w_t\| \leq 1$. By Theorem 23,

$$
\begin{aligned}
\|w_t\| &\leq \sqrt{\sum_{i=1}^{d} w_t[i]^2} \\
&\leq \sqrt{\sum_{k=1}^{n} \|w_t^{(k)}\|^2 + \sum_{l=1}^{n} \|w_t^{(0,l)}\|^2} \\
&< \sqrt{2 \cdot \left(\frac{\eta}{2}\right)^2 + (n-2)\left(\frac{\eta}{8}\right)^2 + 2 \cdot \left(\frac{\eta}{4n}\right)^2} \\
&\leq \sqrt{\left(\frac{\eta^2}{2}\right) + \frac{\eta^2(n-2)}{64} + 2\eta^2} \\
&\leq \sqrt{\frac{1}{64} + \frac{5}{2n}} & (\eta \leq \tfrac{1}{\sqrt{n}}) \\
&\leq 1 & (n \geq 4)
\end{aligned}
$$

Now, denote $\alpha_V \in \mathbb{R}^{n-3}$ the vector which its $k$th entry is $\max\left(\frac{\eta}{16}, \max_{u \in V_i}\langle u, (n-k+2)\frac{\eta}{8} u_{k+1}\rangle\right)$. For $\overline{w}_n = w_{n,n}$, and any $2 \leq s \leq n-2$,

$$\overline{w}_n^{(s)} = \frac{\eta}{2n} u_s + (n-s-1)\frac{\eta}{8n} u_s = (n-s+3)\frac{\eta}{8n} u_s.$$

Then,

$$\frac{1}{n} \sum_{i=1}^{n} \sqrt{\sum_{k=2}^{n} \max\left(\frac{3\eta}{32}, \max_{u \in V_i}\langle u, \overline{w}_n^{(k)}\rangle\right)^2} \geq \frac{1}{n} \sum_{i=1}^{n} \sqrt{\sum_{k=2}^{n-2} \max\left(\frac{3\eta}{32}, \max_{u \in V_i}\langle u, \overline{w}_n^{(k)}\rangle\right)^2}$$

$$= \frac{1}{n} \sum_{i=1}^{n} \sqrt{\sum_{k=2}^{n-2} \max\left(\frac{3\eta}{32}, \max_{u \in V_i}\langle u, (n-k+3)\frac{\eta}{8n}u_k\rangle\right)^2}$$

$$= \frac{1}{n} \sum_{i=1}^{n} \sqrt{\sum_{k=1}^{n-3} \max\left(\frac{3\eta}{32}, \max_{u \in V_i}\langle u, (n-k+2)\frac{\eta}{8n}u_{k+1}\rangle\right)^2}$$

$$= \frac{1}{n} \sum_{i=1}^{n} \sqrt{\sum_{k=1}^{n-3} \max\left(\frac{3\eta}{32}, \max_{u \in V_i}\langle u, (n-k+2)\frac{\eta}{8n}u_{k+1}\rangle\right)^2}$$

$$= \frac{1}{n} \sum_{i=1}^{n} \|\alpha_{V_i}\|$$

$$\geq \|\frac{1}{n} \sum_{i=1}^{n} \alpha_{V_i}\|$$

$$= \sqrt{\sum_{k=2}^{n-2}\left(\frac{1}{n} \sum_{i=1}^{n} \max\left(\frac{3\eta}{32}, \max_{u \in V_i}\langle u, (n-k+3)\frac{\eta}{8n}u_k\rangle\right)\right)^2}$$

$$= \frac{\eta}{8}\sqrt{\sum_{k=2}^{n-2}\left(\frac{1}{n} \sum_{i=1}^{n} \max\left(\frac{3}{4}, \max_{u \in V_i}\langle u, \frac{n-k+3}{n}u_k\rangle\right)\right)^2}$$

$$= \frac{\eta}{8}\sqrt{\sum_{k=2}^{n-2}\left(\frac{1}{n} \sum_{i=1}^{n} \max\left(\frac{3}{4}, \max_{u \in V_i}\langle u, \frac{n-k+2}{n}u_k\rangle\right)\right)^2}$$

*Now, by the fact that if $\mathcal{E}'$ holds, by Theorem 23, for $2 \leq k \leq n-2$, $u_k \in P_k = \bigcap_{i=1}^{k-1} V_k$,*

$$\frac{1}{n} \sum_{i=1}^{n} \sqrt{\sum_{k=2}^{n} \max\left(\frac{3\eta}{32}, \max_{u \in V_i}\langle u, \overline{w}_n^{(k)}\rangle\right)^2}$$

$$\geq \frac{\eta}{8}\sqrt{\sum_{k=2}^{n-2}\left(\frac{1}{n} \sum_{i=1}^{k-1} \max\left(\frac{3}{4}, \max_{u \in V_i}\langle u, \frac{n-k+2}{n}u_k\rangle\right) + \frac{1}{n} \sum_{i=k}^{n} \max\left(\frac{3}{4}, \max_{u \in V_i}\langle u, \frac{n-k+2}{n}u_k\rangle\right)\right)^2}$$

$$\geq \frac{\eta}{8}\sqrt{\sum_{k=2}^{n-2}\left(\frac{3(n-k+1)}{4n} + \frac{k-1}{n}\max\left(\frac{3}{4}, \frac{n-k+2}{n}\right)\right)^2}$$

$$\geq \frac{\eta}{8}\sqrt{\sum_{2 \leq k \leq \frac{n}{4}-2}\left(\frac{3(n-k+1)}{4n} + \frac{(k-1)(n-k+1)}{n^2}\right)^2 + \sum_{\frac{n}{4}-3 < k \leq n-2}\left(\frac{3(n-k+1)}{4n} + \frac{3(k-1)}{4n}\right)^2}$$

$$= \frac{\eta}{8}\sqrt{\sum_{2 \leq k \leq \frac{n}{4}-2}\left(\frac{(n-k+1)(3n+4(k-1))}{4n^2}\right)^2 + \frac{27n}{64}}$$

$$= \frac{\eta}{8}\sqrt{\sum_{1 \leq k \leq \frac{n}{4}-3}\left(\frac{(n-k)(3n+4k)}{4n^2}\right)^2 + \frac{27n}{64}}$$

$$\geq \frac{\eta}{8} \sqrt{\sum_{1 \leq k \leq \frac{n}{4} - 3} \left(\frac{3}{4} + \frac{k}{4n} - \frac{k^2}{n^2}\right)^2 + \frac{27n}{64}}$$

*Now, the fact that for $\frac{n}{8} \leq k \leq \frac{n}{4}$, $\frac{k}{4n} \leq \frac{k^2}{n^2}$ and for $k \leq \frac{n}{8}$, $\frac{k}{8n} \leq \frac{k^2}{n^2}$,*

$$\frac{1}{n} \sum_{i=1}^{n} \sqrt{\sum_{k=2}^{n} \max\left(\frac{3\eta}{32}, \max_{u \in V_i}\langle u, \overline{w}_n{}^{(k)}\rangle\right)^2}$$

$$\geq \frac{\eta}{8} \sqrt{\sum_{1 \leq k \leq \frac{n}{8}} \left(\frac{3}{4} + \frac{k}{8n}\right)^2 + \frac{9n}{128} - \frac{27}{16} + \frac{27n}{64}}$$

$$\geq \frac{\eta}{8} \sqrt{\frac{9n}{128} + \frac{3}{64n} \sum_{k=1}^{\lfloor \frac{n}{8} \rfloor} k + \frac{9n}{128} - \frac{27}{16} + \frac{27n}{64}}$$

$$\geq \frac{\eta}{8} \sqrt{\frac{1}{2}\left(\frac{n}{8} - 1\right)^2 - \frac{27}{16} + \frac{36n}{64}}$$

$$\geq \frac{\eta}{8} \sqrt{\frac{n}{512} - \frac{27}{16} + \frac{36n}{64}} \qquad (n \geq 16)$$

$$\geq \frac{\eta}{8} \sqrt{\frac{577n}{1024}} \qquad (n \geq 2048)$$

$$\geq \frac{3\eta}{32} \cdot \frac{2001}{2000}$$

*Now, for $m < n$ and $2 \leq k \leq n - 2$,*

$$w_{n,m}{}^{(k)} = \begin{cases} \frac{\eta}{8} u_k & k \leq n - m - 1 \\ \frac{1}{m}\left(\frac{\eta}{2} u_k + (n - k - 1)\frac{\eta}{8} u_s\right) & k \geq n - m \end{cases}$$

$$= \begin{cases} \frac{\eta}{8} u_k & k \leq n - m - 1 \\ \frac{\eta(n-k+3)}{8m} u_s & k \geq n - m. \end{cases}$$

*Then, by similar arguments, it holds that,*

$$\frac{1}{n} \sum_{i=1}^{n} \sqrt{\sum_{k=2}^{n} \max\left(\frac{3\eta}{32}, \max_{u \in V_i}\langle u, w_{n,m}{}^{(k)}\rangle\right)^2} \geq \frac{1}{n} \sum_{i=1}^{n} \sqrt{\sum_{k=2}^{n-2} \max\left(\frac{3\eta}{32}, \max_{u \in V_i}\langle u, w_{n,m}{}^{(k)}\rangle\right)^2}$$

$$\geq \sqrt{\sum_{k=2}^{n-2} \left(\frac{1}{n} \sum_{i=1}^{n} \max\left(\frac{3\eta}{32}, \max_{u \in V_i}\langle u, w_{n,m}{}^{(k)}\rangle\right)\right)^2}$$

$$= \frac{\eta}{8} \sqrt{\sum_{k=2}^{n-m-1} \left(\frac{1}{n} \sum_{i=1}^{n} \max\left(\frac{3}{4}, \max_{u \in V_i}\langle u, u_k\rangle\right)\right)^2 + \sum_{k=n-m}^{n-2} \left(\frac{1}{n} \sum_{i=1}^{n} \max\left(\frac{3}{4}, \max_{u \in V_i}\langle u, \frac{n-k+3}{m} u_k\rangle\right)\right)^2}$$

$$\geq \frac{\eta}{8} \sqrt{\sum_{k=2}^{n-m-1} \left(\frac{1}{n} \sum_{i=1}^{n} \max\left(\frac{3}{4}, \max_{u \in V_i}\langle u, u_k\rangle\right)\right)^2 + \sum_{k=n-m}^{n-2} \left(\frac{1}{n} \sum_{i=1}^{n} \max\left(\frac{3}{4}, \max_{u \in V_i}\langle u, \frac{n-k+2}{n} u_k\rangle\right)\right)^2}$$

$$\geq \frac{\eta}{8} \sqrt{\sum_{k=2}^{n-2} \left( \frac{1}{n} \sum_{i=1}^{n} \max \left( \frac{3}{4}, \max_{u \in V_i} \langle u, \frac{n-k+2}{n} u_k \rangle \right) \right)^2} \qquad (k \geq 2 \implies \frac{n-k+2}{n} \leq 1)$$

$$\geq \frac{3\eta}{32} \cdot \frac{2001}{2000} \qquad \qquad \text{(calculation for } w_{n,n})$$

*As a result, we notice that for every $t$, $\ell_2^{SGD}(w_t) \geq -\frac{1}{4n^2} - \frac{1}{n^3}$ and $\ell_2(w_t) \geq \delta_1$ thus, it holds that,*

$$\begin{aligned} \widehat{F}(w_{n,m}) &\geq \frac{3\eta\sqrt{n}}{32} \cdot \frac{2001}{2000} - \frac{1}{4n^2} - \frac{1}{n^3} + \delta_1 \\ &\geq \frac{3\eta\sqrt{n}}{32} \cdot \frac{2001}{2000} - \frac{\eta}{2n^2} \\ &\geq \frac{3\eta\sqrt{n}}{32} \cdot \frac{2001}{2000} - \frac{\eta\sqrt{n}}{80000} \qquad (n \geq 256) \\ &\geq \frac{3\eta\sqrt{n}}{32} \cdot \left( \frac{2001}{2000} - \frac{1}{4000} \right) \\ &\geq \frac{3\eta\sqrt{n}}{32} \cdot \frac{4001}{4000} \end{aligned}$$

*and*

$$\widehat{F}(\widehat{w}_*) \leq \widehat{F}(0) \leq \frac{3\eta}{32} \sqrt{n}$$

*Then, if $\mathcal{E}'$ holds*

$$\begin{aligned} \widehat{F}(w_{n,m}) - \widehat{F}(\widehat{w}_*) &\geq \frac{3\eta\sqrt{n}}{32} \cdot \frac{2001}{2000} - \frac{3\eta}{32} \sqrt{n} \\ &= \frac{\eta\sqrt{n}}{64000} \end{aligned}$$

∎

## B.4. Deferred Proofs

**Proof** [of Theorem 19] The construction is similar to Theorem 4. First, we consider an arbitrary enumeration of $P(U) = \{V^1, ... V^{|P(U)|}\}$ and define $g : P(U) \to \mathbb{R}^2, g(V^i) = \left( \sin \left( \frac{\pi i}{2|P(U)|} \right), \cos \left( \frac{\pi i}{2|P(U)|} \right) \right)$. Here, we refer to a vector $a \in \mathbb{R}^{2n}$ as a concatenation of $n$ vectors in $\mathbb{R}^2$, $a^{(1)}, ..., a^{(n)}$. Then, we define $\delta = 1 - \cos \left( \frac{\pi}{2|P(U)|} \right)$, $\epsilon = \frac{\delta}{n^2}$ and

$$\phi(V, j)^{(i)} = \begin{cases} g(V) & i = j \\ 0 & \text{otherwise} \end{cases}$$

As a result, for every $V_i, j$ it holds that

$$\|\phi(V^i, j)\| = \|g(V^i)\| = \sqrt{\sin \left( \frac{\pi i}{2|P(U)|} \right)^2 + \cos \left( \frac{\pi i}{2|P(U)|} \right)^2} = 1$$

Moreover, if $j_1 \neq j_2$,

$$\langle \phi(V^i, j_1), \phi(V^i, j_2) \rangle = 0,$$

and if $i > k$,

$$
\begin{aligned}
\langle \phi(V^i, j), \phi(V^k, j) \rangle &= \langle g(V^i), g(V^k) \rangle \\
&= \sin\left(\frac{\pi i}{2|P(U)|}\right) \sin\left(\frac{\pi k}{2|P(U)|}\right) + \cos\left(\frac{\pi i}{2|P(U)|}\right) \cos\left(\frac{\pi k}{2|P(U)|}\right) \\
&= \cos\left(\frac{\pi(i-k)}{2|P(U)|}\right) \\
&\leq \cos\left(\frac{\pi}{2|P(U)|}\right) \qquad \text{(cos is monotonic decreasing in } [0, \pi/2]) \\
&= 1 - \delta
\end{aligned}
$$

We notice that $0 < \delta < 1$. Now, we consider an arbitrary enumeration of $U = \{v_1, \ldots v_{|U|}\}$, and define the following sets $\Psi_1, \ldots \Psi_n \subseteq \mathbb{R}^{2n}$ and the following two mappings $\sigma : R^{2n} \to P(U), \alpha : R^{2n} \to U$,

$$\Psi_k = \{\frac{1}{n} \sum_{i=1}^{k} \phi(V_i, i) : \forall i \; V_i \subseteq U\}$$

Note that, for every $\psi \in \Psi$,

$$\|\psi\| = \|\frac{1}{n} \sum_{i=1}^{k} \phi(V_i, j_i)\| \leq \frac{1}{n} \sum_{i=1}^{k} \|\phi(V_i, j_i)\| \leq 1.$$

Then, for every $a \in \mathbb{R}^{2n}$ and $j \in [n]$, we denote the index $q(a, j) \in [|P(U)|]$ as

$$q(a, j) = \arg\max_{r} \langle g(V_r), a^{(j)} \rangle,$$

and define the following mapping $\sigma : \mathbb{R}^{2n} \to P(U)$,

$$\sigma(a) = \bigcap_{j=1, a^{(j)} \neq 0}^{n} V_{q(a,j)}.$$

Moreover, for every $a \in \mathbb{R}^{2n}$, we denote the index $p(a) \in [|U|]$ as

$$p(a) = \arg\min_{i} \{i : v_i \in \sigma(a)\},$$

and define the following mapping $\alpha : \mathbb{R}^{2n^2} \to U$,

$$\alpha(a) = \begin{cases} v_{|U|} & \sigma(a) = \emptyset \\ v_{p(a)} & \sigma(a) \neq \emptyset \end{cases}.$$

Note that for every $a \in \mathbb{R}^{2n}$, $\alpha(a) \in U$, thus, $\|\alpha(a)\| \leq 1$.

Now, Let $V_1, \ldots, V_n \subseteq U$, $k \in [n]$ and $\psi_k^* = \frac{1}{n} \sum_{i=1}^{k} \phi(V_i, i)$. Then,

$$\langle \psi^*, \frac{1}{n} \sum_{i=1}^{k} \phi(V_i, i) \rangle = \langle \frac{1}{n} \sum_{i=1}^{k} \phi(V_i, i), \frac{1}{n} \sum_{i=1}^{k} \phi(V_i, i) \rangle$$

$$= \frac{1}{n^2} \sum_{i=1}^{k} \langle \phi(V_i, i), \phi(V_i, i) \rangle$$

$$= \frac{k}{n^2}$$

For $\psi = \frac{1}{n} \sum_{i=1}^{k} \phi(V_i', i)$ such that $\psi \neq \psi^*$, there exists a index $r$ such that $V_r' \neq V_r$ ,thus,

$$\langle \psi, \frac{1}{n} \sum_{l=1}^{k} \phi(V_i, i) \rangle = \langle \frac{1}{n} \sum_{i=1}^{k} \phi(V_i', i), \frac{1}{n} \sum_{i=1}^{k} \phi(V_i, i) \rangle$$

$$= \frac{1}{n^2} \sum_{i=1}^{k} \langle \phi(V_i, i), \phi(V_i', i) \rangle$$

$$\leq \frac{1}{n^2} \left( 1 - \delta + \sum_{i=1, i \neq r}^{k} 1 \right)$$

$$\leq \frac{1}{n^2} (1 - \delta + k - 1)$$

$$= \frac{k}{n^2} - \frac{\delta}{n^2}$$

$$= \langle \psi_k^*, \frac{1}{n} \sum_{i=1}^{n} \phi(V_i, j_i) \rangle - \epsilon$$

Furthermore, it holds that, $\frac{1}{n} \sum_{i=1}^{n} \phi(V_i, i)^{(i)} = \frac{1}{n} g(V_i)$, thus,

$$q \left( \frac{1}{n} \sum_{i=1}^{k} \phi(V_i, i), i \right) = \arg\max_{r} \langle g(V_r), \frac{1}{n} \sum_{i=1}^{k} \phi(V_i, i)^{(i)} \rangle$$

$$= \arg\max_{r} \langle g(V_r), \frac{1}{n} g(V_i) \rangle$$

$$= i,$$

thus, we get,

$$\sigma(\psi^*) = \sigma \left( \frac{1}{n} \sum_{i=1}^{k} \phi(V_i, i) \right)$$

$$= \bigcap_{j=1, \frac{1}{n} \sum_{i=1}^{k} \phi(V_i, i)^{(j)} \neq 0}^{n} V_{q(\frac{1}{n} \sum_{i=1}^{k} \phi(V_i, i)^{(j)}, j)}$$

$$= \bigcap_{j=1}^{k} V_{q(\frac{1}{n} \sum_{i=1}^{k} \phi(V_i, i), j)} \qquad \text{(The indices that are non-zero are } j = 1, \ldots, k)$$

$$= \bigcap_{i=1}^{k} V_i$$

Then, assuming that $\bigcap_{i=1}^{k} V_i \neq \emptyset$, and let and $m = \arg\min_i \{i : v_i \in \bigcap_{i=1}^{k} V_i\}$, $p(a) = m$ and,

$$\alpha(\psi^*) = v_m \in \bigcap_{i=1}^{k} V_i.$$

∎

**Proof** [of Theorem 21] First, $\ell_1^{\text{SGD}}$ is convex and 1-Lipschitz by the fact that $\ell_1^{\text{SGD}} = \ell_1$ and Theorem 6. Moreover, by Theorem 19, $\ell_2^{\text{SGD}}$ is a maximum over 1-Lipschitz linear functions, thus, $\ell_2^{\text{SGD}}$ is convex and 1-Lipschitz. Finally, $\ell_3^{\text{SGD}}$ is a summation of two 1-Lipschitz linear functions, thus, $\ell_3^{\text{SGD}}$ is convex and 2-Lipschitz. Combining all together, we get the lemma. ∎

**Proof** [of Theorem 22] First, by union bound,

$$\Pr\left(\forall t \in [n]\ P_t \neq \emptyset \text{ and } J_t \in S_t\right) = 1 - \Pr\left(\exists t\ P_t = \emptyset \text{ or } J_t \notin S_t\right)$$

$$\geq 1 - \sum_{t=1}^{n} \Pr\left(P_t = \emptyset \text{ or } (P_t \neq \emptyset \text{ and } J_t \notin S_t)\right)$$

$$\geq 1 - \sum_{t=1}^{n} \Pr\left(P_t = \emptyset\right) - \sum_{t=1}^{n} \Pr\left(P_t \neq \emptyset \text{ and } J_t \notin S_t\right)$$

$$= 1 - \sum_{t=1}^{n} \Pr\left(P_t = \emptyset\right) - \sum_{t=1}^{n} \Pr\left(P_t \neq \emptyset\right) \Pr\left(J_t \notin S_t | P_t \neq \emptyset\right).$$

Now, for every $v \in U$,

$$\Pr(v \notin \bigcap_{i=1}^{t-1} V_i) = 1 - \Pr(v \in \bigcap_{i=1}^{t-1} V_i) = 1 - \delta^{t-1} \leq 1 - \delta^n,$$

and similarly,

$$\Pr\left(v \notin S_t\right) = 1 - (1-\delta)^{n-t+1} \leq 1 - (1-\delta)^n.$$

Then,

$$\Pr\left(P_t = \emptyset\right) = \Pr\left(\bigcap_{i=1}^{t} V_i = \emptyset\right)$$

$$= \Pr(\forall v \in U; v \notin \bigcap_{i=1}^{t} V_i)$$

$$\leq (1 - \delta^n)^{|U|}.$$

Moreover, by the fact that for every $t$, $P_t$ is independent of $V_{t+1}, \dots V_n$,

$$\Pr\left(P_t \neq \emptyset\right) \Pr\left(J_t \notin S_t | P_t \neq \emptyset\right) \leq 1 - (1-\delta)^n.$$

Combining all of the above, we get that,

$$\Pr(\forall t \in [n] \ P_t \neq \emptyset \text{ and } J_t \in S_t)$$

$$= 1 - \sum_{t=1}^{n} \Pr\left(P_t = \emptyset\right) - \sum_{t=1}^{n} \Pr\left(P_t \neq \emptyset\right) \Pr\left(J_t \notin S_t | P_t \neq \emptyset\right)$$

$$\geq 1 - n(1 - \delta^n)^{|U|} - n\left(1 - (1 - \delta)^n\right).$$

Further, for $\delta = \frac{1}{4n^2}$, by the fact that $|U| \geq 2^{\frac{d'}{178}} = 2^{4n \log(n)} = n^{4n}$,

$$|U|\delta^n \geq n^{4n} n^{-2n} 4^{-n} \geq n^{2n} 4^{-n} \geq \log(4n).$$

Therefore,

$$\Pr\left(\forall t \in [n] \ P_t \neq \emptyset \text{ and } J_t \in S_t\right) \geq 1 - n(1 - \delta^n)^{|U|} - n\left(1 - (1 - \delta)^n\right)$$

$$\geq 1 - ne^{-|U|\delta^n} - n\left(1 - (1 - n\delta)\right)$$

$$\geq 1 - ne^{-\log(4n)} - n^2\delta$$

$$\geq 1 - \frac{1}{4} - \frac{1}{4}$$

$$= \frac{1}{2}.$$

∎

**Proof** [of Theorem 24] First, differentiability can be derived immediately from Theorem 31. Second, for 4-Lipschitzness, for every $V \in Z$, we define $\tilde{f}_V^{\text{SGD}} : \mathbb{R}^d \to \mathbb{R}$ as $\tilde{f}_V^{\text{SGD}}(w) := \tilde{f}^{\text{SGD}}(w, V)$. By the 5-Lipschitzness of $f^{\text{SGD}}$ with respect to its first argument and Jensen Inequality, for every $x, y \in \mathbb{R}^d$, it holds that

$$|\tilde{f}_V^{\text{SGD}}(x) - \tilde{f}_V^{\text{SGD}}(y)| = \left|\mathbb{E}_{v \in \delta B}\left(f_V^{\text{SGD}}(y + v)\right) - \mathbb{E}_{v \in \delta B}\left(f_V^{\text{SGD}}(w + v)\right)\right|$$

$$= \left|\mathbb{E}_{v \in \delta B}\left(f_V^{\text{SGD}}(x + v) - f_V^{\text{SGD}}(y + v)\right)\right|$$

$$\leq \mathbb{E}_{v \in \delta B}\left|\left(f_V^{\text{SGD}}(x + v) - f_V^{\text{SGD}}(y + v)\right)\right|$$

$$\leq 4|x - y|.$$

Third, for convexity, by the convexity of $f^{\text{SGD}}$ for every $x, y \in \mathbb{R}^d$ and $\alpha \in [0, 1]$,

$$\tilde{f}_V^{\text{SGD}}\left(\alpha x + (1 - \alpha)y\right) = \mathbb{E}_{v \in \delta B}\left(f_V^{\text{SGD}}(\alpha x + (1 - \alpha)y + v)\right)$$

$$= \mathbb{E}_{v \in \delta B}\left(f_V^{\text{SGD}}(\alpha(x + v) + (1 - \alpha)(y + v))\right)$$

$$\leq \mathbb{E}_{v \in \delta B}\left(\alpha f_V^{\text{SGD}}(x + v) + (1 - \alpha)f_V(y + v)\right)$$

$$= \alpha \mathbb{E}_{v \in \delta B}\left(f_V^{\text{SGD}}(x + v)\right) + (1 - \alpha)\left(\mathbb{E}_{v \in \delta B} f_V^{\text{SGD}}(y + v)\right)$$

$$= \alpha \tilde{f}_V^{\text{SGD}}(x) + (1 - \alpha)\tilde{f}_{V,j}^{\text{SGD}}(y).$$

**Proof** [of Theorem 25] We assume that $\mathcal{E}'$ (Eq. (22)) holds and prove Theorem 25 under this event. We prove the claim by induction on $t$. For $t = 1$, it is trivial. Now, we assume that $w_t = \tilde{w}_t$. First, for $\tilde{\ell}_3^{\text{SGD}}$ and every $w$ and $V$, by linearity of expectation,

$$\tilde{\ell}_3^{\text{SGD}}(w, V) = \mathbb{E}_{v \in \delta B} \left( \langle w^{(0,1)} + v^{(0,1)}, -\frac{1}{4n^2} \phi(V, 1) \rangle - \langle \frac{1}{n^3} u_1, w^{(1)} + v^{(1)} \rangle \right)$$

$$= \ell_3^{\text{SGD}}(w, V) + \left( \langle \mathbb{E}_{v \in \delta B} v^{(0,1)}, -\frac{1}{4n^2} \phi(V, 1) \rangle - \langle \frac{1}{n^3} u_1, \mathbb{E}_{v \in \delta B} v^{(1)} \rangle \right)$$

$$= \ell_3^{\text{SGD}}(w, V)$$

Then, we derive that for every $w$, $\nabla \tilde{\ell}_3^{\text{SGD}}(w, V) = \nabla \ell_3^{\text{SGD}}(w, V)$. Now, for $r \in \{1, 2\}$ we show that in each term $\tilde{\ell}_r^{\text{SGD}}(w_t, V_t)$, the argument that gives the maximum value is the same as $\ell_r^{\text{SGD}}(w_t, V_t)$.

For $\tilde{\ell}_1^{\text{SGD}}(w_t, V_t)$, in every $t$, by the proofs of Theorem 28, Theorem 29 and Theorem 23, the maximal value is $\frac{3\eta}{32}$. Moreover, it can be observed that for every $k \geq 2$, $\max_t \max_{u \in V_t} \langle u, w_t^{\text{SGD}(k)} \rangle \leq \frac{\eta}{16}$. Then, by Theorem 33, and the hypothesis of the induction,

$$\nabla \tilde{\ell}_1^{\text{SGD}}(\tilde{w}_t, V_t) = \nabla \tilde{\ell}_1^{\text{SGD}}(w_t, V_t) = 0 = \nabla \ell_1^{\text{SGD}}(w_t, V_t).$$

Now, for $\tilde{\ell}_2^{\text{SGD}}$, for $t = 1$, $\nabla \ell_2^{\text{SGD}}(w_1, V_1) = 0$ and the maximum is attained uniquely in $\delta_1 = \frac{\eta}{8n^3}$ (the second maximal value is zero). Then, we can apply Theorem 32 and by the hypothesis of the induction, it follows that,

$$\nabla \tilde{\ell}_2^{\text{SGD}}(\tilde{w}_1, V_1) = \nabla \tilde{\ell}_2^{\text{SGD}}(w_1, V_1) = 0 = \nabla \ell_2^{\text{SGD}}(w_1, V_1).$$

For every $t \geq 2$, the maximum is attained uniquely in the linear term of $k = t - 1, u = u_{t-1}$ and $\psi = \psi_{t-1}^*$ such that the difference between the maximal value the second largest value is larger than $\frac{\eta \epsilon}{16n^2}$. Then, we can apply Theorem 34 and by the hypothesis of the induction, it follows that,

$$\nabla \tilde{\ell}_2^{\text{SGD}}(\tilde{w}_t, V_t) = \nabla \tilde{\ell}_2^{\text{SGD}}(w_t, V_t) = \nabla \ell_t^{\text{SGD}}(w_t, V_t).$$

In conclusion, we proved that $\nabla \tilde{f}^{\text{SGD}}(\tilde{w}_t, V_t) = \nabla f^{\text{SGD}}(w_t, V_t)$, thus, by the hypothesis of the induction,

$$w_{t+1} = w_t - \nabla f^{\text{SGD}}(w_t, V_t) = \tilde{w}_t - \nabla \tilde{f}^{\text{SGD}}(\tilde{w}_t, V_t) = \tilde{w}_{t+1}.$$

$\blacksquare$

# Appendix C. Differentiability: Auxiliary Lemmas

**Lemma 31** *(Lemma 1 in Flaxman et al. (2005)) Let $d$ and $\delta > 0$, $\mathbb{B}$ be the $d$-dimensional unit ball and $\mathbb{S}$ be the $d$-dimensional unit sphere. Moreover, let $\mathcal{D}_{\mathbb{B}}$ and $\mathcal{D}_{\mathbb{S}}$ be the uniform distributions on $\mathbb{B}, \mathbb{S}$ respectively. If $\tilde{f}(x) = \mathbb{E}_{v \sim \mathcal{D}_{\mathbb{B}}} [f(x + \delta v)]$, then,*

$$\nabla \tilde{f}(x) = \frac{d}{\delta} \mathbb{E}_{a \sim \mathcal{D}_{\mathbb{S}}} [f(x + \delta a) a]$$

**Lemma 32** *Let $d$. Let $\mathbb{B}$ be the $d$-dimensional unit ball and $\mathcal{D}_{\mathbb{B}}$ the uniform distributions on $\mathbb{B}$. Let $\zeta_1 > \zeta_2 > 0$, a function $g : \mathbb{R} \to \mathbb{R}$ and $a_1, \ldots a_l \in \mathbb{B}$. Moreover, let $h : \mathbb{B} \to R$, $h(x) = g\left(\max(\zeta_1, \max_{1 \le r \le l} \langle a_r, x \rangle\right)$ and $x_0 \in \mathbb{B}$ such that $\max_{1 \le r \le l} \langle a_r, x_0 \rangle \le \zeta_2$. We define $\tilde{h}(x) := \mathbb{E}_{v \sim \mathcal{D}_{\mathbb{B}}} [h(x + \delta v)]$. Then, for any $0 < \delta < \zeta_1 - \zeta_2$,*

$$\nabla \tilde{h}(x_0) = 0,$$
$$\tilde{h}(x_0) = g(\zeta_1).$$

**Proof** First, for every $r$ and $v \in \mathbb{B}$, by Cauchy-Schwartz Inequality,

$$\langle a_r, x_0 + \delta v \rangle = \langle a_r, x_0 \rangle + \langle a_r, \delta v \rangle \le \zeta_2 + \delta < \zeta_1$$

Then,

$$\max(\zeta_1, \max_{1 \le r \le l} \langle a_r, x_0 + \delta v \rangle) = \zeta_1,$$

and

$$h(x_0 + \delta v) = g(\max(\zeta_1, \max_{1 \le r \le l} \langle a_r, x_0 + \delta v \rangle)) = g(\zeta_1).$$

As a result,

$$\tilde{h}(x_0) = \mathbb{E}_{v \sim \mathcal{D}_{\mathbb{B}}} [h(x_0 + \delta v)] = g(\zeta_1)$$

and by Theorem 31,

$$
\begin{aligned}
\nabla \tilde{h}(x_0) &= \frac{d}{\delta} \mathbb{E}_{v \sim \mathcal{D}_{\mathbb{S}}} [h(x_0 + \delta v) v] \\
&= \frac{d}{\delta} \mathbb{E}_{v \sim \mathcal{D}_{\mathbb{S}}} \left[ g \left( \max(\zeta_1, \max_{1 \le r \le l} \langle a_r, x_0 + \delta v \rangle) \right) v \right] \\
&= \frac{d}{\delta} \mathbb{E}_{v \sim \mathcal{D}_{\mathbb{S}}} [g(\zeta_1) v] \\
&= \frac{d}{\delta} g(\zeta_1) \mathbb{E}_{v \sim \mathcal{D}_{\mathbb{S}}} [v] \\
&= 0
\end{aligned}
$$

∎

**Lemma 33** *Let $d$ and $K$. Let $\mathbb{B}$ be the $dK$-dimensional unit ball and $\mathcal{D}_{\mathbb{B}}$ the uniform distributions on $\mathbb{B}$. Let $\zeta_1 > \zeta_2 > 0$ and $a_1, \ldots a_l \in \mathbb{B}^d$. Moreover, let $g : \mathbb{B}^d \to R$, $g(x) = \max(\zeta_1, \max_{1 \le r \le l} \langle a_r, x \rangle)$ and $h : \mathbb{B} \to \mathbb{R}$, $h(x) = \sqrt{\sum_{k=1}^{K} g(x^{(k)})^2}$. Let $x_0 \in \mathbb{B}$ such that for every $k$, $\max_{1 \le r \le l} \langle a_r, x_0^{(k)} \rangle \le \zeta_2$. We define $\tilde{h}(x) := \mathbb{E}_{v \sim \mathcal{D}_{\mathbb{B}}} [h(x + \delta v)]$. Then, for any $0 < \delta < \zeta_1 - \zeta_2$,*

$$\nabla \tilde{h}(x_0) = 0,$$
$$\tilde{h}(x_0) = \zeta_1 \sqrt{K}.$$

**Proof** First, for every $k, r$ and $u \in \mathbb{B}^d$, by Cauchy-Schwartz Inequality,

$$\langle a_r, x_0^{(k)} + \delta u \rangle = \langle a_r, x_0^{(k)} \rangle + \langle a_r, \delta u \rangle \leq \zeta_2 + \delta < \zeta_1$$

Then,

$$g(x_0^{(k)} + \delta u) = \max(\zeta_1, \max_{1 \leq r \leq l} \langle a_r, x_0^{(k)} + \delta u \rangle) = \zeta_1,$$

and for every $v \in \mathbb{B}$,

$$h(x_0 + \delta v) = \sqrt{\sum_{k=1}^{K} g(\max(\zeta_1, \max_{1 \leq r \leq l} \langle a_r, x_0^{(k)} + \delta v^{(k)} \rangle)} = \zeta_1 \sqrt{K}.$$

As a result,

$$\tilde{h}(x_0) = \mathbb{E}_{v \sim \mathcal{D}_\mathbb{B}} [h(x_0 + \delta v)] = \zeta_1 \sqrt{K}.$$

Now, by Theorem 31,

$$
\begin{aligned}
\nabla \tilde{h}(x_0) &= \frac{d}{\delta} \mathbb{E}_{v \sim \mathcal{D}_\mathbb{S}} [h(x_0 + \delta v)v] \\
&= \frac{d}{\delta} \mathbb{E}_{v \sim \mathcal{D}_\mathbb{S}} \left[ \left( \sqrt{\sum_{k=1}^{K} \left( \max(\zeta_1, \max_{1 \leq r \leq l} \langle a_r, x_0^{(k)} + \delta v^{(k)} \rangle) \right)^2} \right) \cdot v \right] \\
&= \frac{d}{\delta} \mathbb{E}_{v \sim \mathcal{D}_\mathbb{S}} \left[ \zeta_1 \sqrt{K} v \right] \\
&= \frac{d}{\delta} \zeta_1 \sqrt{K} \mathbb{E}_{v \sim \mathcal{D}_\mathbb{S}} [v] \\
&= 0.
\end{aligned}
$$

∎

**Lemma 34** *Let $d$. Let $\mathbb{B} = \mathbb{B}_G^d$ be the d-dimensional ball of radius $G$ and $\mathcal{D}_\mathbb{B}$ the uniform distributions on $\mathbb{B}$. Let $\zeta_1 > \zeta_2, \zeta_3 > 0$ and vectors $a_1, \dots a_l \in \mathbb{B}$. Moreover, let $h : \mathbb{B} \to \mathbb{R}$, $h(x) = \max(\zeta_3, \max_{1 \leq r \leq l} \langle a_r, x \rangle)$ and $x_0 \in \mathbb{B}, r_0 \in [l]$ such that $\langle a_{r_0}, x_0 \rangle = \zeta_1$ and $\max_{1 \leq r \leq l, r \neq r_0} \langle a_r, x_0 \rangle \leq \zeta_2$. We define $\tilde{h}(x) := \mathbb{E}_{v \sim \mathcal{D}_\mathbb{B}} [h(x + \delta v)]$. Then, for any $0 < \delta < \frac{1}{2G} (\zeta_1 - \max(\zeta_2, \zeta_3))$,*

$$
\begin{aligned}
\tilde{h}(x_0) &= \langle a_{r_0}, x_0 \rangle \\
\nabla \tilde{h}(x_0) &= a_{r_0}
\end{aligned}
$$

**Proof** First, by Cauchy-Schwartz Inequality,

$$
\begin{aligned}
\max \left( \zeta_3, \max_{r \neq r_0} \langle a_r, x_0 + \delta v \rangle \right) &\leq \max \left( \zeta_3, \max_{r \neq r_0} \langle a_r, x_0 \rangle + \max_{r \neq r_0} \langle \delta v, a_r \rangle \right) \\
&\leq \max(\zeta_3, \zeta_2 + G\delta) \\
&\leq \max(\zeta_3, \zeta_2) + G\delta
\end{aligned}
$$

$$< \frac{1}{2}(\zeta_1 + \max(\zeta_3, \zeta_2)),$$

and for $r_0$,

$$\langle a_{r_0}, x_0 + \delta v \rangle = \langle a_{r_0}, x_0 \rangle + \langle \delta v, a_{r_0} \rangle \geq \zeta_1 - G\delta > \frac{1}{2}(\zeta_1 + \max(\zeta_2, \zeta_3)).$$

We derive that for every $v \in \mathbb{B}$,

$$h(x_0 + \delta v) = \max\left(\zeta_3, \max_{1 \leq r \leq l} \langle a_r, x + \delta v \rangle\right) = \langle a_{r_0}, x + \delta v \rangle.$$

and that the maximum is attained in $r_0$. Then,

$$\begin{aligned}
\tilde{h}(x_0) &= \mathbb{E}_{v \sim \mathcal{D}_\mathbb{B}}[h(x_0 + \delta v)] \\
&= \mathbb{E}_{v \sim \mathcal{D}_\mathbb{B}}[\langle a_{r_0}, x_0 + \delta v \rangle] \\
&= \langle a_{r_0}, x_0 + \delta \mathbb{E}_{v \sim \mathcal{D}_\mathbb{B}} v \rangle \\
&= \langle a_{r_0}, x_0 \rangle
\end{aligned}$$

and by Theorem 31,

$$\begin{aligned}
\nabla \tilde{h}(x_0) &= \frac{d}{\delta} \mathbb{E}_{v \sim \mathcal{D}_\mathbb{S}}\left[\max\left(\zeta_3, \left(\max_{1 \leq r \leq l} \langle a_r, x_0 + \delta v \rangle\right)\right) v\right] \\
&= \frac{d}{\delta} \mathbb{E}_{v \sim \mathcal{D}_\mathbb{S}}\left[(\langle a_{r_0}, x_0 + \delta v \rangle) v\right] \\
&= \langle a_{r_0}, x_0 \rangle \frac{d}{\delta} \mathbb{E}_{v \sim \mathcal{D}_\mathbb{S}}[v] + \frac{d}{\delta} \mathbb{E}_{v \sim \mathcal{D}_\mathbb{S}}[\langle a_{r_0}, \delta v \rangle v] \\
&= 0 + d \mathbb{E}_{v \sim \mathcal{D}_\mathbb{S}}[vv^T] a_{r_0} \\
&= d \mathbb{E}_{v \sim \mathcal{D}_\mathbb{S}}[vv^T] a_{r_0} \\
&= d \frac{1}{d} I a_{r_0} \\
&= a_{r_0}.
\end{aligned}$$

$\blacksquare$

**Lemma 35** *Let $d$ and $\delta > 0$. Let $f : \mathbb{R}^d \to \mathbb{R}$ be a $G$-Lipschitz function. Let $\mathbb{B}$ be the $d$-dimensional unit ball. Moreover, let $\mathcal{D}_\mathbb{B}$ be the uniform distributions on $\mathbb{B}$. If $\tilde{f}(x) = \mathbb{E}_{v \sim \mathcal{D}_\mathbb{B}}[f(x + \delta v)]$, then for every $x$,*

$$|\tilde{f}(x) - f(x)| \leq G\delta$$

**Proof** By the fact that $f$ is $G$-Lipschitz,

$$\begin{aligned}
|\tilde{f}(x) - f(x)| &= |\mathbb{E}_{v \sim \mathcal{D}_\mathbb{B}}[f(x + \delta v)] - f(x)| \\
&\leq |\mathbb{E}_{v \sim \mathcal{D}_\mathbb{B}}[f(x)] + G\delta \mathbb{E}_{v \sim \mathcal{D}_\mathbb{B}} + [\|v\|] - f(x)| \\
&= G\delta \mathbb{E}_{v \sim \mathcal{D}_\mathbb{B}}[\|v\|] \\
&\leq G\delta
\end{aligned}$$

$\blacksquare$

# Appendix D. Lower bound of $\Omega\left(\min\left(1, \frac{1}{\eta T}\right)\right)$

In this section, we prove the $\Omega\left(\min\left(1, \frac{1}{\eta T}\right)\right)$ lower bound. Since our hard construction for getting this bound involves a deterministic loss function, GD is equivalent to SGD. For clarity, we refer in our proof to the performance of GD, however, the same result is applicable also for SGD with $T = n$ iterations.

## D.1. Construction of a non-differentiable loss function.

For $d = \max(25\eta^2 T^2, 1)$, we define the hard loss function $f^{\text{OPT}} : \mathbb{R}^d \to \mathbb{R}$, as follows,

$$f^{\text{OPT}}(w) = \max\left(0, \max_{i \in [d]}\{\frac{1}{\sqrt{d}} - w[i] - \frac{\eta i}{4d}\}\right). \tag{29}$$

For this loss function, we prove the following lemma,

**Lemma 36** *Assume $n, T > 0, \eta \leq \frac{1}{5\sqrt{T}}$. Consider the loss function $f^{\text{OPT}}$ that is defined in Eq. (29) for $d = \max(25\eta^2 T^2, 1)$. Then, for Unprojected GD (cf. Eq. (1) with $W = \mathbb{R}^d$) on $f^{\text{OPT}}$, initialized at $w_1 = 0$ with step size $\eta$, we have,*

   *(i) The iterates of GD remain within the unit ball, namely $w_t \in \mathbb{B}^d$ for all $t = 1, \ldots, T$;*

   *(ii) For all $m = 1, \ldots, T$, the m-suffix averaged iterate has:*

$$f^{\text{OPT}}(w_{T,m}) - f^{\text{OPT}}(w_*) = \Omega\left(\min\left(1, \frac{1}{\eta T}\right)\right).$$

**Algorithm's dynamics** We start by proving a lemma that characterizes the dynamics of the algorithm.

**Lemma 37** *Assume the conditions of Theorem 36, and consider the iterate of Unprojected GD on $f^{\text{OPT}}$, initialized at $w_1 = 0$ with step size $\eta \leq \frac{1}{5\sqrt{T}}$ Let $w_t$ be the iterate of Then, it holds that,*

   *(i) For every $i \in [d]$ and for every $t \in [T]$,*

$$w_t[i] \leq \frac{1}{2\sqrt{d}}$$

   *(ii) For every $t \in [T]$, there exists an index $j_t \in [d]$ such that $k \neq j_t$,*

$$\frac{1}{\sqrt{d}} - w_t[j_t] - \frac{\eta j}{4d} > \frac{1}{\sqrt{d}} - w_t[k] - \frac{\eta k}{4d} + \frac{\eta}{8d}.$$

   *(iii) For every $t \in [T]$, $j_t$ also holds*

$$\frac{1}{\sqrt{d}} - w_t[j_t] - \frac{\eta j_t}{4d} > \frac{\eta}{8d}.$$

**Proof** We prove by induction on $t$. For $t = 1$, $w_t = 0$, thus,

$$w_1[i] = 0 \le \frac{1}{2\sqrt{d}}.$$

Moreover, the maximizer is $j_1 = 1$. Then, we notice that for both $d = 1$ and $d = 25\eta^2 T^2$, $\eta \le \frac{1}{5\sqrt{T}} \implies \eta \le \frac{1}{5\sqrt{d}}$. Then, it holds that,

$$\frac{1}{\sqrt{d}} - w_1[j_1] - \frac{\eta j_1}{4d} \ge \frac{1}{\sqrt{d}} - w_1[j_1] - \frac{\eta}{4}$$

$$\ge \frac{19}{20\sqrt{d}}$$

$$> \frac{\eta}{8d},$$

and, for every $k \ne j_1$,

$$\frac{1}{\sqrt{d}} - w_1[j_1] - \frac{\eta j_1}{4d} = \frac{1}{\sqrt{d}} - w_1[k] - \frac{\eta k}{4d} + \frac{\eta(k - j_1)}{4d}$$

$$\ge \frac{1}{\sqrt{d}} - w_1[k] - \frac{\eta k}{4d} + \frac{\eta}{4d}$$

$$> \frac{1}{\sqrt{d}} - w_1[k] - \frac{\eta k}{4d} + \frac{\eta}{8d}.$$

In the step of the induction we assume that the lemma holds for every $s \le t$ and prove it for $s = t + 1$. By the hypothesis of the induction, we know that for every iteration $s \le t$, $\|w_t\|_2 \le \frac{1}{2}$, as a result, we know that the projections does not affect the dynamics of the algorithm until the iteration $t$. Moreover, we know that for every iteration $s \le t$ there exists an index $j_s \in [d]$ such that the term that achieve the maximum value in $w_s$ is $\frac{1}{\sqrt{d}} - w_s[j_s] - \frac{\eta j}{4d}$. This maximum is attained uniquely in $j_s$ by margin that is strictly larger than $\frac{\eta}{8d}$. As a result, we derive that, for every $s \le t$, $\nabla f(w_s) = -e_{j_s}$. Now, for every index $m \in [d]$, we define,

$$n_t^m = |\{s \le t : m = \arg\max_{i \in [d]}\{\frac{1}{\sqrt{d}} - w_s[i] - \frac{\eta i}{4d}\}\}|.$$

We get that, for every $i$ it holds that,

$$w_{t+1}[i] = \eta n_t^i.$$

Then,

$$\|w_{t+1}\|_1 = \sum_i \eta n_t^i \le \eta t,$$

and ,thus, there exists a entry $k \in [d]$ with $w_{t+1}[k] \le \frac{\eta t}{d}$. Now, we prove the first part of the lemma using this observation and the step of the induction. For every $i \ne j_t$,

$$w_{t+1}[i] = w_t[i] \le \frac{1}{2\sqrt{d}}.$$

Otherwise, we know that, by the definition of $j_t$

$$\frac{1}{\sqrt{d}} - w_t[i] - \frac{\eta i}{4d} > \frac{1}{\sqrt{d}} - w_t[k] - \frac{\eta k}{4d} + \frac{\eta}{8d},$$

$$\begin{aligned}
w_t[i] &< w_t[k] + \frac{\eta(k-i)}{4d} - \frac{\eta}{8d} \\
&\leq \frac{\eta t}{d} + \frac{\eta}{4} \\
&\leq \frac{1}{25\sqrt{d}} + \frac{1}{20\sqrt{d}}
\end{aligned}$$

and,

$$\begin{aligned}
w_{t+1}[i] &\leq w_t[i] + \eta \\
&\leq \frac{1}{25\sqrt{d}} + \frac{1}{20\sqrt{d}} + \frac{1}{5\sqrt{d}} \\
&\leq \frac{1}{2\sqrt{d}},
\end{aligned}$$

where we again used the fact that $\eta \leq \frac{1}{5\sqrt{T}}$ implies $\eta \leq \frac{1}{5\sqrt{d}}$ for both $d = 1$ and $d = 25\eta^2 T^2$.

For the second part of the lemma, we define $J_t \subseteq [d]$, $J_t = \arg\min_j \{n_t^j\}$ and $j_{t+1} = \min\{j \in J_t\}$ and show that $j_{t+1}$ holds the required. We know, for every $j \neq i \in [d]$,

$$w_{t+1}[i] - w_{t+1}[j] = \eta(n_t^i - n_t^j).$$

For $k \neq j_{t+1}$ with $n_t^k > n_t^{j_{t+1}}$,

$$\begin{aligned}
\frac{1}{\sqrt{d}} - w_{t+1}[j_{t+1}] - \frac{\eta j_{t+1}}{4d} &\leq \frac{1}{\sqrt{d}} - w_{t+1}[k] - \eta - \frac{\eta j_{t+1}}{4d} \\
&= \frac{1}{\sqrt{d}} - w_{t+1}[k] - \eta - \frac{\eta k}{4d} + \frac{\eta(k - j_{t+1})}{4d} \\
&\leq \frac{1}{\sqrt{d}} - w_{t+1}[k] - \eta - \frac{\eta k}{4d} + \frac{\eta}{4} \\
&< \frac{1}{\sqrt{d}} - w_{t+1}[k] - \frac{\eta k}{4d} - \frac{\eta}{2}.
\end{aligned}$$

in contradiction to the fact that $j_{t+1}$ gets the maximal value. For $k \neq j_{t+1}$ with $n_t^k > n_t^{j_{t+1}}$, it holds that $w_{t+1}[j_{t+1}] \leq w_{t+1}[k] - \eta$, and,

$$\begin{aligned}
\frac{1}{\sqrt{d}} - w_{t+1}[j_{t+1}] - \frac{\eta j_{t+1}}{4d} &\geq \frac{1}{\sqrt{d}} - w_{t+1}[k] + \eta - \frac{\eta j_{t+1}}{4d} \\
&= \frac{1}{\sqrt{d}} - w_{t+1}[k] + \eta - \frac{\eta k}{4d} + \frac{\eta(k - j_{t+1})}{4d} \\
&\geq \frac{1}{\sqrt{d}} - w_{t+1}[k] + \eta - \frac{\eta k}{4d} - \frac{\eta}{4}
\end{aligned}$$

$$> \frac{1}{\sqrt{d}} - w_{t+1}[k] - \frac{\eta k}{4d} + \frac{\eta}{8d},$$

as required. For the third part of the lemma, we know that for every $i \in [d]$,

$$\begin{aligned}
\frac{1}{\sqrt{d}} - w_{t+1}[i] - \frac{\eta i}{4d} &\geq \frac{1}{2\sqrt{d}} - \frac{\eta}{4} \\
&= \frac{9}{20\sqrt{d}} \\
&> \frac{\eta}{8d}.
\end{aligned}$$

∎

**Proof of lower bound.** Now we can prove Theorem 36.

**Proof** [of Theorem 36] The first part of the theorem is an immediate corollary from Theorem 37. Moreover, by applying this lemma again, we know that, for every $i \in [d]$,

$$w_{T,m}[i] \leq \frac{1}{2\sqrt{d}},$$

thus,

$$\begin{aligned}
f^{\mathrm{OPT}}(w_{T,m}) - f^{\mathrm{OPT}}(w_*) &\geq \frac{1}{2\sqrt{d}} - \frac{\eta}{4} - 0 \\
&\geq \frac{1}{2\sqrt{d}} - \frac{\eta}{20\sqrt{d}} \\
&> \frac{1}{4\sqrt{d}} \\
&= \min\left(\frac{1}{4}, \frac{1}{20\eta T}\right).
\end{aligned}$$

∎

## D.2. Construction of a differentiable loss function.

In this section, we prove the lower bound for a smoothing of $f^{\mathrm{OPT}}$, defined as

$$\tilde{f}^{\mathrm{OPT}}(w) = \mathbb{E}_{v \in \mathbb{B}^d} \max\left(0, \max_{i \in [d]}\{\frac{1}{\sqrt{d}} - w[i] - \delta v[i] - \frac{\eta i}{4d}\}\right), \tag{30}$$

namely, we prove the following lemma,

**Lemma 38** *Assume $n, T > 0, \eta \leq \frac{1}{5\sqrt{T}}$. Consider the loss function $\tilde{f}^{\mathrm{OPT}}$ that is defined in Eq. (30) for $d = \max(25\eta^2 T^2, 1)$ and $\delta = \frac{\eta}{16d}$. Then, for Unprojected GD (cf. Eq. (1) with $W = \mathbb{R}^d$) on $f^{\mathrm{OPT}}$, initialized at $w_1 = 0$ with step size $\eta$, we have,*

   *(i) The iterates of GD remain within the unit ball, namely $w_t \in \mathbb{B}^d$ for all $t = 1, \ldots, T$;*

*(ii) For all $m = 1, \ldots, T$, the m-suffix averaged iterate has:*

$$\tilde{f}^{OPT}(w_{T,m}) - \tilde{f}^{OPT}(w_*) = \Omega\left(\min\left(1, \frac{1}{\eta T}\right)\right).$$

First, we prove that the smoothing of the loss function does not affect the dynamics of the algorithm, as stated in the following lemma,

**Lemma 39** *Under the conditions of Theorems 36 and 38, let $w_t, \tilde{w}_t$ be the iterates of Unprojected GD with step size $\eta \leq \frac{1}{5\sqrt{T}}$ and $w_1 = 0$, on $f^{OPT}$ and $\tilde{f}^{OPT}$ respectively. Then, for every $t \in [T]$, it holds that $w_t = \tilde{w}_t$.*

**Proof** We proof the lemma by induction on $t$. For $t = 1$, we know that $w_1 = \tilde{w}_1 = 0$. Now, we assume that $w_t = \tilde{w}_t$. By Theorem 37, we know that the maximum of the loss function is attained uniquely with the property that the difference between the maximal value and the second maximal value is larger then $\frac{\eta}{8d}$. As a result, by the facts that $f$ is 1-Lipschitz and $\delta = \frac{\eta}{16d}$, we can use Theorem 34 for $\widehat{F}(w_t)$ and get that,

$$\nabla \widehat{F}(w_t) = \nabla \widehat{\tilde{F}}(w_t) = \nabla \widehat{\tilde{F}}(\tilde{w}_t).$$

It follows by the hypothesis of the induction that,

$$w_{t+1} = w_t - \nabla \widehat{F}(w_t) = \tilde{w}_t - \nabla \widehat{\tilde{F}}(\tilde{w}_t) = \tilde{w}_{t+1}.$$

∎

Now we can prove Theorem 38.

**Proof** [of Theorem 38] Let $\overline{w_{T,m}}$ be the $m$-suffix average of $GD$ when is applied on $f^{OPT}$. Let $\overline{w_*} = \arg\min_w f^{OPT}(w)$. By Theorem 39, we know that, $w_{T,m} = \overline{w_{T,m}}$. Then, by Theorem 36 and Theorem 35,

$$\frac{1}{4\sqrt{d}} \leq f^{OPT}(\overline{w_{T,m}}) - f^{OPT}(\overline{w_*})$$

$$= f^{OPT}(w_{T,m}) - f^{OPT}(\overline{w_*})$$

$$\leq \tilde{f}^{OPT}(w_{T,m}) + \delta - \tilde{f}^{OPT}(\overline{w_*}) + \delta$$

$$\leq \tilde{f}^{OPT}(w_{T,m}) + \delta - \tilde{f}^{OPT}(w_*) + \delta,$$

and,

$$\tilde{f}^{OPT}(w_{T,m}) - \tilde{f}^{OPT}(w_*) \geq \frac{1}{4\sqrt{d}} - \frac{\eta}{8d}$$

$$\geq \frac{1}{4\sqrt{d}} - \frac{1}{8\sqrt{d}}$$

$$\geq \frac{1}{8\sqrt{d}}$$

$$\geq \min(\frac{1}{8}, \frac{1}{40\eta T}).$$

∎

