# OpenReview forum: "The Dimension Strikes Back with Gradients: Generalization of Gradient Methods in Stochastic Convex Optimization"
_algorithmiclearningtheory.org/ALT/2025/Conference — ALT 2025_

### Official Review · Reviewer_NCzh · 2024-11-08

**Rating:** 7
**Confidence:** 4

**Review:**

This work investigates the generalization properties of gradient descent (GD) and stochastic gradient descent (SGD) in high-dimensional stochastic convex optimization problems, specifically examining how the sample complexity scales with the problem dimension. The authors provide an in-depth analysis of the dimension dependence in generalization, delivering two notable lower bounds on the sample complexity for both full-batch GD and one-pass SGD.

For full-batch GD, they construct a learning scenario with problem dimension $𝑑=𝑂(𝑛^2)$, showing that even when tuned to optimize the empirical risk, GD can yield an approximate empirical risk minimizer with Ω(1) excess population risk. This result implies a lower bound of Ω(𝑑) on the number of training examples required for GD to reach a non-trivial generalization error, addressing open questions raised by previous work (Feldman, 2016; Amir, Koren, and Livni, 2021). This finding underscores that non-trivial dependence on the problem dimension is unavoidable, highlighting a significant challenge in achieving effective generalization in high-dimensional settings.

Furthermore, for one-pass SGD, the authors apply a similar technique to demonstrate a Ω(d) lower bound for the sample complexity needed to achieve a non-trivial empirical error, even though SGD achieves optimal test performance. This result provides an exponential improvement in dimension dependence over previous bounds (Koren et al., 2022) and resolves an open question from that work.

Overall, this paper advances the understanding of dimension-dependent generalization bounds for gradient methods in stochastic convex optimization, offering theoretical contributions that clarify the limitations and challenges of GD and SGD in high-dimensional settings. The results could impact how these optimization techniques are applied to large-scale learning problems, guiding future research into more dimension-aware gradient-based methods. I went through the proof to the best of my abilities within the time-limit provided and they seem correct to me.

Hence, I recommend acceptance.

**Paper Award:**

No

---

### Official Review · Reviewer_AE5G · 2024-11-10
**Solid piece of work with interesting problem, clear message, and significant result**

**Rating:** 7
**Confidence:** 3

**Review:**

(1) In the paper under review, the authors consider the problem of stochastic convex optimization with Lipschitz loss and address the questions raised by Feldman (NeurIPS, 2016), Amir, Koren, Livni (NeurIPS, 2021), and Koren, Livni, Mansour, Sherman (NeurIPS 2022) concerning lower bounds on the excess risk of gradient descent and stochastic gradient descent. In particular, if the parameter dimension $d$ is of order $\Theta(n T + n^2 + \eta^2 T^2)$, they provide an example (Theorem 1) when the excess risk of the GD estimate (with a fixed step size and $T$ iterations) is $\Omega( \min\{\eta \sqrt{T} + T/\eta, 1\} )$. When $T = n$ and $d = \Theta(n \log n + n^2 + \eta^2 n^2)$, a similar lower bound holds for the excess generalization error of SGD (see Theorem 2). This means that one needs at least $\Omega(\sqrt{d})$ samples to reach a nontrivial error.

(2) The authors note that the established lower bound $\Omega(\sqrt{d})$ is weaker than the sample complexity $\Omega(\sqrt{d})$ for a general ERM (see (Feldman, NeurIPS, 2016)). However, Theorems 1 and 2 significantly improve over Amir, Koren, Livni (NeurIPS, 2021), and Koren, Livni, Mansour, Sherman (NeurIPS 2022), respectively.

(3) The paper is well written. The authors spend much efforts to put the problem of interest into the context and describe main ideas used in the previous works. They also overview main steps in the proofs of Theorem 1 and 2. I find this important, because the appendix is quite long and I could not check it carefully during the review period. I only have a couple of minor remarks listed below.

(4) The authors mention a follow-up work (Livni, 2024) where the author proved a stronger $\Omega(d)$ lower bound on the sample complexity of GD and bridged the gap between the performance of GD and ERM. However, Livni (2024) was inspired by the ideas of the present paper. For this reason, I think that the follow-up result of Livni (2024) does not belittle merits of the present work. Besides, the question whether a similar bound holds for SGD is still open.

(5) There is a line of research studying the performance of ERMs and GD in the case of strongly convex and exp-concave losses (see, e.g., (Koren, Levy, NeurIPS, 2015), (Klochkov, Zhivotovskiy, NeurIPS, 2021), (Puchkin, Zhivotovskiy, COLT, 2023)). Is it possible to establish similar lower bounds in these scenarios?


Minor remarks.

(i) The expressions $1/\eta T$ and $1/\eta n$ are slightly confusing (see pp. 3, 4, 6, etc.). I would replace them by $T/\eta$ and $n/\eta$, respectively.

(ii) Please, mention that $\mathbb B^d$ in Theorems 1 and 2 stands for the unit ball in $\mathbb R^d$. This notation was not introduced before.

**Paper Award:**

Yes

---

### Official Review · Reviewer_s8ww · 2024-11-16
**Interesting contributions, needs some minor re-writes for exposition.**

**Rating:** 7
**Confidence:** 3

**Review:**

The paper provides new lower bounds for gradient descent as well as stochastic gradient descent through a nontrivial modification of the Feldman construction. The results are crisply stated. The contributions are meaningful and the proof techniques are non-trivial extensions of previous lower bound constructions. The memorization trick for the gradient descent so that the descent step finds the bad solution u_0 is specially interested.

Some parts of the introduction could use more background to link the classical results with the modern ones. For example, the discussion about the uniform convergence bounds and the sudden shift to the discussion about the modern regimes which the authors call “practical” is a bit too much of a jump unless the reader is already familiar with these settings. I would suggest adding more context about how the uniform convergence settings can be too restrictive to be applicable in practical machine learning. Similarly there is no discussion of the Koren 2022 result, it is mentioned directly as one of the contributions saying the authors achieve improvement in dimension dependence of the “empirical risk” (quotes mine, but the authors use italics without explaining what it is and how it is difference from the generalization error that they have been discussing in the rest of the introduction). Further down the sentence about failure of uniform convergence and lack of stability in GD vs SGD is too vague.

Section 3.1 needs a re-write. What is the optimization problem? Is it min of h over all w in U? what is V in eq 6? How is the probability of any u \in U being = V_i for any i equal to 0.5? I had to read the Feldman paper to understand the construction.

Do the proposed constructions, especially for the SGD setup, extend of any first order method?

Disclaimer: I have not gone through all the proofs given the short review time.

**Paper Award:**

No

---

### Author Rebuttal · Authors · 2024-11-21

We thank all reviewers for their thoughtful comments and positive feedback. Below we have addressed any concerns and / or questions.

# Reviewer 1 (s8ww)

**“Some parts of the introduction could use more background to link the classical results with the modern ones” / “Section 3.1 needs a re-write”;** We thank you again for your thorough review and helpful comments - we will revise these sections and take your notes into account.

**Regarding the extension of our work to any first-order method:**
This is a great question. For the case of lower bounding the generalization gap as in the SGD setup, the result cannot be extended to any first order method since SGD with-replacement (Koren et al. 2022, Section 4) exhibits a generalization gap that decreases at the optimal (dimension independent) rate - see Corollary 2 in Koren et al. 2022.
That being said, it would be interesting to understand which first order methods our construction does extend to - this is unclear to us at the moment and requires additional future work.

# Reviewer 2 (AE5G)

**(5) Performance of ERMs and GD in the strongly convex / exp-concave setting:**
This is indeed a very interesting question.
For the case of strongly convex functions, previous works (e.g., Amir, Koren, Livni ‘21) establish lower bounds of $\Omega(1/\lambda \sqrt n)$ for the canonical GD setup in the  $\lambda$-strongly convex setting (via a construction with exponential dimension); so, better than the general convex case but weaker than the desired $1/\lambda n$ rate. As to whether a similar result can be achieved with a poly dimension construction - we do not have a definitive answer, but suspect it should be possible with additional technical work.
With regards to the exp-concave setting: First, any lower bound established for the class of strongly convex functions will essentially carry over to the exp-concave setting. Second, for proving stronger lower bounds, we believe some of our techniques would prove valuable in this case, but as in the strongly convex setting, would also require additional technical contributions that merit future independent work.

**Minor remarks:** Thanks for pointing these out! We will amend as necessary (in fact, the expressions you mentioned should be $1/(\eta T)$ and $1/(\eta n)$) .

# Reviewer 3 (NCzh)

Thank you again for your support of our paper, we will gladly address any remaining questions during the discussion period.

---

### Meta-Review · Area_Chair_s9jv · 2024-12-12

**Recommendation:** Accept
**Confidence:** 4

**Metareview:**

The paper presents the generalization gap of gradient descent (GD) and stochastic gradient descent (SGD), with specific focus on dimension dependence. It introduces new lower bound $\Omega(\sqrt{d})$ number of samples reach a non-trivial test error.  The main proof ideas are based Feldman 2016 construction of a bad ERM and adding non-smooth component of the loss. The contribution of the paper are also well acknowledged by all reviewers.

Also acknowledged by reviewers, the proofs are non-trivial and insightful for the lower bound establishment. The paper is clearly written and well organized. The authors should revise the paper according to the reviewer's suggestion and responses.

**Paper Award:**

No